# Online Detection of LLM-Generated Texts via Sequential Hypothesis Testing by Betting

Can Chen [1]   Jun-Kun Wang [1,2]

## Abstract

Developing algorithms to differentiate between machine-generated texts and human-written texts has garnered substantial attention in recent years. Existing methods in this direction typically concern an offline setting where a dataset containing a mix of real and machine-generated texts is given upfront, and the task is to determine whether each sample in the dataset is from a large language model (LLM) or a human. However, in many practical scenarios, sources such as news websites, social media accounts, and online forums publish content in a streaming fashion. Therefore, in this online scenario, how to quickly and accurately determine whether the source is an LLM with strong statistical guarantees is crucial for these media or platforms to function effectively and prevent the spread of misinformation and other potential misuse of LLMs. To tackle the problem of *online* detection, we develop an algorithm based on the techniques of sequential hypothesis testing by betting that not only builds upon and complements existing offline detection techniques but also enjoys statistical guarantees, which include a controlled false positive rate and the expected time to correctly identify a source as an LLM. Experiments were conducted to demonstrate the effectiveness of our method.

## 1. Introduction

Over the past few years, there has been growing evidence that LLMs can produce content with qualities on par with human-level writing, including writing stories (Yuan et al.,

[1]Halıcıoğlu Data Science Institute, University of California San Diego, La Jolla, USA [2]Department of Electrical and Computer Engineering, University of California San Diego, La Jolla, USA. Correspondence to: Can Chen <cac024@ucsd.edu>, Jun-Kun Wang <jkw005@ucsd.edu>.

*Proceedings of the $42^{nd}$ International Conference on Machine Learning*, Vancouver, Canada. PMLR 267, 2025. Copyright 2025 by the author(s).

2022), producing educational content (Kasneci et al., 2023), and summarizing news (Zhang et al., 2024). On the other hand, concerns about potentially harmful misuses have also accumulated in recent years, such as producing fake news (Zellers et al., 2019), misinformation (Lin et al., 2021; Chen & Shu, 2023), plagiarism (Bommasani et al., 2021; Lee et al., 2023), malicious product reviews (Adelani et al., 2020), and cheating (Stokel-Walker, 2022; Susnjak & McIntosh, 2024). To tackle the relevant issues associated with the rise of LLMs, a burgeoning body of research has been dedicated to distinguishing between human-written and machine-generated texts (Jawahar et al., 2020; Lavergne et al., 2008; Hashimoto et al., 2019; Gehrmann et al., 2019; Mitchell et al., 2023; Su et al., 2023; Bao et al., 2023; Solaiman et al., 2019; Bakhtin et al., 2019; Zellers et al., 2019; Ippolito et al., 2019; Tian, 2023; Uchendu et al., 2020; Fagni et al., 2021; Adelani et al., 2020; Abdelnabi & Fritz, 2021; Zhao et al., 2023; Kirchenbauer et al., 2023; Christ et al., 2024).

While these existing methods can efficiently identify a text source in an offline setting where each text is classified independently, they are not specifically designed to handle scenarios where texts arrive sequentially, and the goal is to quickly detect if the source is an LLM with strong statistical guarantees. A common limitation of most detectors is that their *offline* detection process involves computing a score for each sample to determine how likely it is machine-generated and classify a text as a machine-generated one by determining whether the score passes a certain threshold. The threshold is a critical parameter that needs to be tuned using a validation dataset and must be determined before classifying testing data. Moreover, existing *offline* detectors are unable to control the type-I error rate (false positive rate) in an *online* setting—even for state-of-the-art zero-shot detectors such as Fast-DetectGPT (Bao et al., 2023), Binoculars (Hans et al., 2024), or other efficient models like RoBERTa-Base/Large (Liu et al., 2019), ReMoDetect (Lee et al., 2024), and Raidar (Mao et al., 2024), which do not require threshold tuning at inference time.

For example, a naive way to adapt an *offline* detector to an *online* scenario is to declare the source as an LLM as soon as any single text is predicted as such. However, this adaptation leads to a false positive rate that converges to 1

as the number of texts becomes sufficiently large—unless the detector is perfectly (100%) accurate on human-written texts. Formally, if the detector has accuracy $\delta$ on human-written texts, then under the naive reduction approach to online detection, the false positive rate over $T$ rounds will be $1 - \delta^T$, which approaches 1 as $T$ grows, even when $\delta$ is very close to 1.

Therefore, these offline detection methods fall short in the *online* scenarios, where one would like to design efficient algorithms with robust performance metrics, such as being a valid level-$\alpha$ test, having asymptotic power 1, and enjoying a bound on the expected detection time for LLMs, which can be appealing for time-sensitive applications like monitoring online platforms. For example, the American Federal Communications Commission in 2017 decided to repeal net neutrality rules according to the public opinions collected through an online platform (Selyukh, 2017; Weiss, 2019). However, it was ultimately discovered that the overwhelming majority of the total 22 million comments that support rescinding the rules were machine-generated (Kao, 2017). In 2019, Weiss (2019) used GPT-2 to overwhelm a website for collecting public comments on a medical reform waiver within only four days, where machine-generated comments eventually made up 55.3% of all the comments (more precisely, $1,001$ out of $1,810$ comments). As discussed by Fröhling & Zubiaga (2021), a GPT-J model trained on a politics message board was then deployed on the same forum. It generated posts that included objectionable content and accounted for about 10% of all activity during peak times (Kilcher, 2022). Furthermore, other online attacks mentioned by Fröhling & Zubiaga (2021) may even manipulate public discourse (Ferrara et al., 2016), flood news with fake content (Belz, 2019), or fraud by impersonating others on the Internet or via e-mail (Solaiman et al., 2019). However, to the best of our knowledge, existing bot detection methods for social media (e.g., Davis et al. (2016); Varol et al. (2017); Pozzana & Ferrara (2020); Ferrara (2023) and the references therein) might not be directly applicable to the online setting with strong statistical guarantees, and they often require training on extensive labeled datasets beforehand. This highlights the urgent need to develop algorithms with strong statistical guarantees that can quickly and effectively identify machine-generated texts in a timely manner, which has been largely overlooked in the literature.

Our goal, therefore, is to tackle the problem of online detection of LLM-generated texts. More precisely, building upon existing score functions from those "offline approaches", we aim to quickly determine whether the source of a sequence of texts observed in a streaming fashion is an LLM or a human, *with* a concrete bound on the required samples to detect LLMs and *without* the need for tuning a threshold of the affinity score that is typically used to classify data in those offline approaches. Our algorithm leverages the techniques

of sequential hypothesis testing (Shafer, 2021; Ramdas et al., 2023; Shekhar & Ramdas, 2023). Specifically, we frame the problem of online LLM detection as a sequential hypothesis testing problem, where at each round $t$, a text from an unknown source is observed, and we aim to infer whether it is generated by an LLM. We also assume that a pool of examples of human-written texts is available, and our algorithm can sample a text from this pool of examples at any time $t$. Our method constructs a null hypothesis $H_0$ (to be elaborated soon), for which correctly rejecting the null hypothesis implies that the algorithm correctly identifies the source as an LLM under a mild assumption. Furthermore, since it is desirable to quickly identify an LLM when it is present and avoid erroneously declaring the source as an LLM, we also aim to control the type-I error rate (false positive rate) while maximizing the power to reduce type-II error rate (false negative rate), and to establish an upper bound on the expected total number of rounds to declare that the source is an LLM. We emphasize that our approach is non-parametric, and hence it does not need to assume that the underlying data of human or machine-generated texts follow a certain distribution (Balsubramani & Ramdas, 2015). It also avoids the need for assuming that the sample size is fixed or to specify it before the testing starts, and hence it is in contrast with some typical hypothesis testing methods that do not enjoy strong statistical guarantees in the *anytime-valid* fashion (Garson, 2012; Good, 2013; Tartakovsky et al., 2014). The way to achieve these is based on recent developments in sequential hypothesis testing via betting (Shafer, 2021; Shekhar & Ramdas, 2023). The setting of online testing with *anytime-valid* guarantees could be particularly useful when one seeks substantial savings in both data collection and time without compromising the reliability of their statistical testing. These desiderata might be elusive for approaches that based on collecting data in batch and classifying them offline to achieve.

We evaluate the effectiveness of our method through comprehensive experiments. The code and datasets are available via this link: https://github.com/canchen-cc/online-llm-detection.

## 2. Preliminaries

We begin by providing a recap of the background on sequential hypothesis testing.

**Sequential Hypothesis Testing with Level-$\alpha$ and Asymptotic Power One.** Let us denote a forward filtration $\mathcal{F} = (\mathcal{F}_t)_{t \geq 0}$, where $\mathcal{F}_t = \sigma(Z_1, \ldots, Z_t)$ represents an increasing sequence that accumulates all the information from the observations $\{Z_i : i \geq 1\}$ up to time point $t$. A process $W := (W_t)_{t \geq 1}$, adapted to $(\mathcal{F}_t)_{t \geq 1}$, is defined as a P-martingale if it satisfies $E_P[W_t | \mathcal{F}_{t-1}] = W_{t-1}$ for all $t \geq 1$. Furthermore, $W$ is a P-supermartingale if

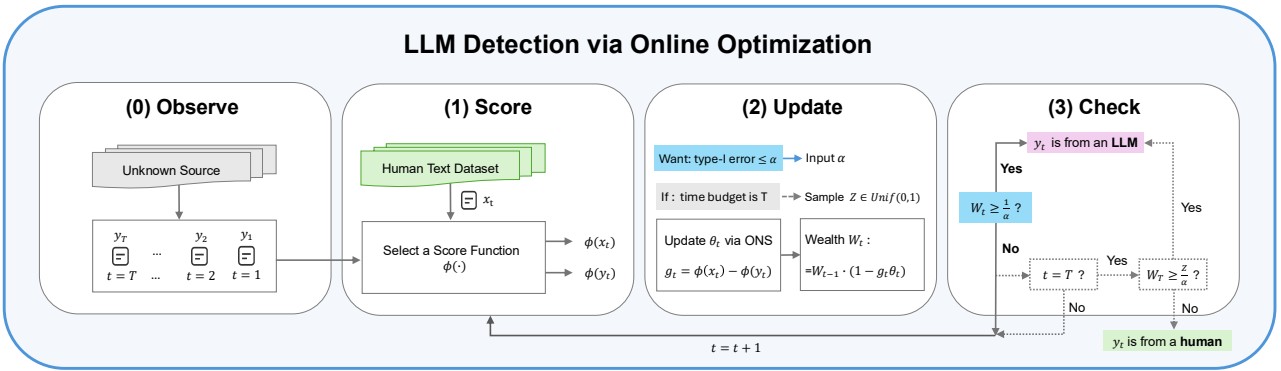

*Figure 1.* Overview of LLM Detection via Online Optimization. (0) (**Observe**) We sequentially observe text $y_t$ generated by an unknown source starting from time $t = 1$ and aim to determine whether these texts are produced by a human or an LLM. The detection process can be divided into three steps. (1) (**Score**) At each time $t$, text $x_t$ and $y_t$ are evaluated by a selected score function $\phi(\cdot)$, where the sample $x_t$ is drawn from a prepared dataset consisting of human-written text examples. (2) (**Update**) The parameter $\theta_t$ is updated via the Online Newton Step (ONS) to increase the wealth $W_t$ rapidly when $y_t$ is an LLM-generated text. A large value of $W_t$ serves as significant evidence and provides confidence to declare that the unknown source is an LLM. (3) (**Check**) Whether the wealth $W_t \geq 1/\alpha$ is checked. If this event happens, we declare the unknown source of $y_t$ as LLM. Otherwise, if the time budget $T$ is not yet exhausted or if we have an unlimited time budget, we proceed to $t + 1$ and repeat the steps. When $t = T$, if the condition $W_T \geq Z/\alpha$ holds, where $Z$ is drawn from an uniform distribution in $[0, 1]$, our algorithm will also declare the source as an LLM.

$E_P[W_t | \mathcal{F}_{t-1}] \leq W_{t-1}$ for all $t \geq 1$. In our algorithm design, we will consider a martingale $W$ and define the event $\{W_t \geq 1/\alpha\}$ as rejecting the null hypothesis $H_0$, where $\alpha > 0$ is a user-specified "significance level" parameter. We denote the stopping time $\tau := \inf\{t \geq 1 : W_t \geq \frac{1}{\alpha}\}$ accordingly.

We further recall that a hypothesis test is a level-$\alpha$ test if $\sup_{P \in H_0} P(\exists t \geq 1, W_t \geq 1/\alpha) \leq \alpha$, or alternatively, if $\sup_{P \in H_0} P(\tau < \infty) \leq \alpha$. Furthermore, a test has asymptotic power $1 - \beta$ if $\sup_{P \in H_1} P(\forall t \geq 1, W_t < 1/\alpha) \leq \beta$, or if $\sup_{P \in H_1} P(\tau = \infty) \leq \beta$, where $H_1$ represents the alternative hypothesis. A test with asymptotic power one (i.e., $\beta = 0$) means that when the alternative hypothesis $H_1$ is true, the test will eventually rejects the null hypothesis $H_0$. As shown later, our algorithm that will be introduced shortly is a provable sequential hypothesis testing method with level-$\alpha$ and asymptotic power one.

**Problem Setup.** We consider a scenario in which, at each round $t$, a text $y_t$ from an unknown source is observed, and additionally, a human-written text $x_t$ can be sampled from a dataset of human-written text samples at our disposal. The goal is to quickly and correctly determine whether the source that produces the sequence of texts $\{y_t\}_{t=1}^T$ is an LLM or a human. We assume that a score function $\phi(\cdot) : \text{Text} \to \mathbb{R}$ is available, which, given a text as input, outputs a score. The score function $\phi(\cdot)$ that we consider in this work are those proposed for detecting LLM-generated texts in offline settings, e.g., Mitchell et al. (2023); Bao et al. (2023); Su et al. (2023); Bao et al. (2023); Yang et al.

(2023). We provide more details on these score functions in the experiments section and in the exposition of the literature in Appendix A.1.

Following related works on sequential hypothesis testing via online optimization (e.g., Shekhar & Ramdas (2023); Chugg et al. (2023)), we assume that each text $y_t$ is i.i.d. from a distribution $\rho^y$, and similarly, each human-written text $x_t$ is i.i.d. from a distribution $\rho^x$. Denote the mean $\mu_x := \mathbb{E}_{\rho^x}[\phi(x)]$ and $\mu_y := \mathbb{E}_{\rho^y}[\phi(y)]$ respectively. The task of hypothesis testing that we consider can be formulated as

$$H_0 \text{ (null hypothesis)} : \quad \mu_x = \mu_y,$$
$$H_1 \text{ (alternative hypothesis)} : \quad \mu_x \neq \mu_y.$$

We note that when $H_0$ is true, this is not equivalent to saying that the texts $\{y_t\}_{t=1}^T$ are human-written, as different distributions can share the same mean. However, under the additional assumption of the existence of a good score function $\phi(\cdot)$ which produces scores for machine-generated texts with a mean $\mu_y$ different from that of human-generated texts $\mu_x$, $H_0$ is equivalent to the unknown source being human. Therefore, under this additional assumption, when the unknown source $\rho^y$ is an LLM, then rejecting the null hypothesis $H_0$ is equivalent to correctly identifying that the source is indeed an LLM. In our experiments, we found that this assumption holds empirically for the score functions we adopt. That is, the empirical mean of $\phi(y_t)$ significantly differs from that of $\phi(x_t)$ when each $y_t$ is generated by an LLM. Figure 1 illustrates the online detection process for LLM-generated texts.

**Sequential Hypothesis Testing by Betting.** Consider the scenario that an online learner engages in multiple rounds of a game with an initial wealth $W_0 = 1$. In each round $t$ of the game, the learner plays a point $\theta_t$. Then, the learner receives a fortune after committing $\theta_t$, which is $-g_t\theta_t W_{t-1}$, where $W_{t-1}$ is the learner's wealth from the previous round $t-1$, and $g_t$ can be thought of as "the coin outcome" at $t$ that the learner is trying to "bet" on (Orabona & Pál, 2016). Consequently, the dynamic of the wealth of the learner evolves as:

$$W_t = W_{t-1} \cdot (1 - g_t\theta_t) = W_0 \cdot \Pi_{s=1}^t (1 - g_s\theta_s). \quad (1)$$

To connect the learner's game with sequential hypothesis testing, one of the key techniques that will be used in the algorithm design and analysis is Ville's inequality (Ville, 1939), which states that if $(W_t)_{t\geq 1}$ is a nonnegative supermartingale, then one has $P(\exists t : W_t \geq 1/\alpha) \leq \alpha\mathbb{E}[W_0]$. The idea is that if we can guarantee that the learner's wealth $W_t$ remains nonnegative from $W_0 = 1$, then Ville's inequality can be used to control the type-I error at level $\alpha$ *simultaneously at all time steps* $t$. To see this, let $g_t = \phi(x_t) - \phi(y_t)$. Then, when $P \in H_0$ (i.e., the null hypothesis $\mu_x = \mu_y$ holds), the wealth $(W_t)_{t\geq 1}$ is a P-supermartingale, because

$$\mathbb{E}_P[W_t|F_{t-1}] = \mathbb{E}_P[W_{t-1}(1 - \theta_t g_t)|F_{t-1}]$$
$$= \mathbb{E}_P[W_{t-1} \cdot (1 - \theta_t(\phi(x_t) - \phi(y_t)))|F_{t-1}] = W_{t-1}.$$

Hence, if the learner's wealth $W_t$ can remain nonnegative given the initial wealth $W_0 = 1$, we can apply Ville's inequality to get a provable level-$\alpha$ test, since $W_t$ is a nonnegative supermartingale in this case. Another key technique is *randomized* Ville's inequality (Ramdas & Manole, 2023) for a nonnegative supermartingale $(W_t)_{t\geq 1}$, which states that $P(\exists t \leq T : W_t \geq 1/\alpha \text{ or } W_T \geq Z/\alpha) \leq \alpha$, where $T$ is any $\mathcal{F}$-stopping time and $Z$ is randomly drawn from the uniform distribution in $[0, 1]$. This inequality becomes particularly handy when there is a time budget $T$ in sequential hypothesis testing while maintaining a valid level-$\alpha$ test.

We now switch to discussing the control of the type-II error, which occurs when the wealth $W_t$ is not accumulated enough to reject $H_0$ when $H_1$ is true. Therefore, we need a mechanism to enable the online learner in the game quickly increase the wealth under $H_1$. Related works of sequential hypothesis testing by betting (Shekhar & Ramdas, 2023; Chugg et al., 2023) propose using a no-regret learning algorithm to achieve this. Specifically, a no-regret learner aims to obtain a sublinear regret, which is defined as $\text{Regret}_T(\theta_*) := \sum_{t=1}^T \ell_t(\theta_t) - \sum_{t=1}^T \ell_t(\theta_*)$, where $\theta_*$ is a benchmark. In our case, we will consider the loss function at $t$ to be $\ell_t(\theta) := -\ln(1 - g_t\theta)$. The high-level idea is based on the observation that the first term in the regret definition is the log of the learner's wealth, modulo a minus sign, i.e., $\ln(W_T) = \sum_{t=1}^T \ln(1 - g_t\theta_t) =$

---

**Algorithm 1** Online Detection of LLMs via Online Optimization and Betting

**Require:** a score function $\phi(\cdot) : \text{Text} \to \mathbb{R}$.
1: **Init:** $\theta_1 \leftarrow 0$, $a_0 \leftarrow 1$, wealth $W_0 \leftarrow 1$, step size $\gamma$, and significance level parameter $\alpha \in (0, 1)$.
2: *# $T$ is the time budget, which can be set to $\infty$ if there is no time constraint.*
3: **for** $t = 1, 2, \ldots, T$ **do**
4:     Observe a text $y_t$ from an unknown source and compute $\phi(y_t)$.
5:     Sample $x_t$ from a dataset of human-written texts and compute $\phi(x_t)$.
6:     Set $g_t = \phi(x_t) - \phi(y_t)$.
7:     Update wealth $W_t = W_{t-1} \cdot (1 - g_t\theta_t)$.
8:     **if** $W_t \geq 1/\alpha$ **then**
9:         Declare that the source producing the sequence of texts $y_t$ is an LLM.
10:     **end if**
11:     Get a hint $d_{t+1}$ which satisfies $d_{t+1} \geq |g_{t+1}|$.
12:     Specify the decision space $\mathcal{K}_{t+1}$ as $\mathcal{K}_{t+1} := [-\frac{1}{2d_{t+1}}, \frac{1}{2d_{t+1}}]$ to ensure $W_{t+1} \geq 0$.
13:     *# Update $\theta_{t+1} \in \mathcal{K}_{t+1}$ via ONS on the loss function $\ell_t(\theta) := -\ln(1 - g_t\theta)$.*
14:     Compute $z_t = \frac{d\ell_t(\theta_t)}{d\theta} = \frac{g_t}{1 - g_t\theta_t}$, $a_t = a_{t-1} + z_t^2$.
15:     $\theta_{t+1} = \max\left(\min\left(\theta_t - \frac{1}{\gamma}\frac{z_t}{a_t}, \frac{1}{2d_{t+1}}\right), -\frac{1}{2d_{t+1}}\right)$.
16: **end for**
17: **if** the source has not been declared as an LLM **then**
18:     Sample $Z \sim \text{Unif}(0, 1)$, declare the sequence of texts $y_t$ is from an LLM if $W_T \geq Z/\alpha$.
19: **end if**

---

$-\sum_{t=1}^T \ell_t(\theta_t)$, while the second term is that of a benchmark. Therefore, if the learner's regret can be upper-bounded, say $C$, then the learner's wealth is lower-bounded as $W_T \geq \left(\Pi_{t=1}^T(1 - g_t\theta_*)\right)\exp(-C)$. An online learning algorithm with a small regret bound can help increase the wealth quickly under $H_1$. We refer the reader to Appendix D for a rigorous argument, where we note that applying a no-regret algorithm to guarantee the learner's wealth is a neat technique that is well-known in online learning, see e.g., Chapter 9 of Orabona (2019) for more details. Following existing works (Shekhar & Ramdas, 2023; Chugg et al., 2023), we will adopt Online Newton Steps (ONS) (Hazan et al., 2007) in our algorithm.

## 3. Our Algorithm

We have covered most of the underlying algorithmic design principles of our online method for detecting LLMs, and we are now ready to introduce our algorithm, which is shown in Algorithm 1. Compared to existing works on sequential hypothesis testing via betting (e.g., Shekhar & Ramdas (2023);

Chugg et al. (2023)), which assume knowledge of a bound on the magnitude of the "coin outcome" $g_t$ in the learner's wealth dynamic (1) for all time steps before the testing begins (e.g., assuming for all $t$, $\phi(x_t)$ and $\phi(y_t)$ are always in $[0,1]$), we relax this assumption, which is motivated by the concern that a good estimate of the bound for the coin outcome might not be available before the sequential testing starts. Specifically, we consider the scenario where an upper bound on $|g_{t+1}|$ at round $t+1$, which is denoted by $d_{t+1}$, is available before updating $\theta_{t+1}$ at each round $t$. Our algorithm then plays a point in the decision space $\mathcal{K}_{t+1}$ that guarantees the learner's wealth remains a non-negative supermartingale (Step 12 in Algorithm 1). We note that if the bound of the output of the underlying score function $\phi(\cdot)$ is known *a priori*, this scenario holds naturally. Otherwise, we can estimate an upper bound for $|g_t|$ for all $t$ based on the first few time steps and execute the algorithm thereafter. One approach is to set the estimate as a conservatively large constant, e.g., twice the maximum value observed in the first few time steps. We observe that this estimate works for our algorithm with most of the score functions $\phi(\cdot)$ that we consider in the experiments. On the other hand, we note that a tighter bound $d_t$ will lead to a faster time to reject $H_0$ when the unknown source is an LLM, as indicated by the following propositions, where we note that the analysis of the test power requires the assumption that the data are i.i.d.

**Proposition 3.1.** *Algorithm 1 is a level-$\alpha$ sequential test with asymptotic power one. Furthermore, if $y_t$ is generated by an LLM, the expected time $\tau$ to declare the unknown source as an LLM is bounded by*

$$\mathbb{E}[\tau] = \mathcal{O}\left(\frac{d_*^2}{\Delta^2}\ln\left(\frac{d_*}{\alpha\Delta}\right) + \frac{d_*^4}{\Delta^4}\right),$$

*where $\Delta := |\mu_x - \mu_y|$, $d_* := \max_{t\geq 1} d_t$ with $d_t \geq |g_t|$, and $\gamma = \frac{1}{2}\min\{\frac{d_t}{G_t}, 1\}$ with $G_t := \max_{\theta\in\mathcal{K}_t}|\nabla\ell_t(\theta)|$ denoting the upper bound of the gradient $\nabla\ell_t(\theta)$.*

**Remark 1.** Under the additional assumption of the existence of a good score function $\phi(\cdot)$ that can generate scores with different means for human-written texts and LLM-generated ones, Proposition 3.1 implies that when the unknown source is declared by Algorithm 1 as an LLM, the probability of this declaration being false will be bounded by $\alpha$. Additionally, if the unknown source is indeed an LLM, then our algorithm can guarantee that it will eventually detect the LLM, since it has asymptotic power one. Moreover, Proposition 3.1 also provides a non-asymptotic result for bounding the expected time to reject the null hypothesis $H_0$, which is also the expected time to declare that the unknown source is an LLM. The bound indicates that a larger difference of the means $\Delta$ can lead to a shorter time to reject the null $H_0$.

**(Composite Hypotheses.)** As Chugg et al. (2023), we also consider the composite hypothesis, which can be formulated

as $H_0 : |\mu_x - \mu_y| \leq \epsilon$ versus $H_1 : |\mu_x - \mu_y| > \epsilon$. The hypothesis can be equivalently expressed in terms of two hypotheses,

$$H_0^A : \mu_x - \mu_y - \epsilon \leq 0 \text{ vs. } H_1^A : \mu_x - \mu_y - \epsilon > 0,$$
$$H_0^B : \mu_y - \mu_x - \epsilon \leq 0 \text{ vs. } H_1^B : \mu_y - \mu_x - \epsilon > 0.$$

Consequently, the dynamic of the wealth evolves as $W_t^A = W_{t-1}^A \cdot (1 - \theta_t(g_t - \epsilon))$ and $W_t^B = W_{t-1}^B \cdot (1 - \theta_t(-g_t - \epsilon))$ respectively, where $g_t = \phi(x_t) - \phi(y_t)$. We note that both $g_t - \epsilon$ and $-g_t - \epsilon$ are within the interval $[-d_t - \epsilon, d_t - \epsilon]$. The composite hypothesis is motivated by the fact that, in practice, even if both sequences of texts $x_t$ and $y_t$ are human-written, they may have been written by different individuals. Therefore, it might be more reasonable to allow for a small difference $\epsilon > 0$ in their means when defining the null hypothesis $H_0$.

**Proposition 3.2.** *Algorithm 3 in the appendix is a level-$\alpha$ sequential test with asymptotic power one, where the wealth $W_t^A$ for $H_0^A(H_1^A)$ and $W_t^B$ for $H_0^B(H_1^B)$ are calculated through level-$\alpha/2$ tests. Furthermore, if $y_t$ is generated by an LLM, the expected time $\tau$ to declare the unknown source as an LLM is bounded by*

$$\mathbb{E}[\tau] = \mathcal{O}\left(\frac{(d_* + \epsilon)^2}{(\Delta - \epsilon)^2}\ln\left(\frac{d_* + \epsilon}{\alpha(\Delta - \epsilon)}\right) + \frac{(d_* + \epsilon)^4}{(\Delta - \epsilon)^4}\right),$$

*where $\Delta := |\mu_x - \mu_y|$, $d_* := \max_{t\geq 1} d_t$ with $d_t \geq |g_t|$, $\gamma = \frac{1}{2}\min\{\frac{2d_t}{G_t}, 1\}$ with $G_t := \max_{\theta\in\mathcal{K}_t}|\nabla\ell_t(\theta)|$ denoting the upper bound of the gradient $\nabla\ell_t(\theta)$, and $\epsilon$ is a nonnegative constant satisfying $\epsilon \leq d_t$ for all $t$.*

**Remark 2.** We note that when the alternative hypothesis $H_1$ is true, we have $\Delta > \epsilon$. Proposition 3.2 indicates that even if there is a difference $\epsilon$ in mean scores between texts written by different humans, the probability that the source is incorrectly declared by Algorithm 3 as an LLM can be controlled below $\alpha$. Furthermore, the bound on $\mathbb{E}[\tau]$ implies that smaller $\epsilon$ and larger $\Delta$ will result in a shorter time to reject $H_0$.

## 4. Experiments

**Score Functions.** We use 10 score functions in total from the related works for the experiments. As mentioned earlier, a score function takes a text as an input and outputs a score. For example, one of the configurations of our algorithm that we try uses a score function called Likelihood, which is based on the average of the logarithmic probabilities of each token conditioned on its preceding tokens (Solaiman et al., 2019; Hashimoto et al., 2019). More precisely, for a text $x$ which consists of $n$ tokens, this score function can be formulated as $\phi(x) = \frac{1}{n}\sum_{j=1}^n \log p_\theta(x_j|x_{1:j-1})$, where $x_j$ denotes the $j$-th token of the text $x$, $x_{1:j-1}$ means the first

$j - 1$ tokens, and $p_\theta$ represents the probability computed by a language model used for scoring. The score functions that we considered in the experiments are: 1. DetectGPT: perturbation discrepancy (Mitchell et al., 2023). 2. Fast-DetectGPT: conditional probability curvature (Bao et al., 2023). 3.LRR: likelihood log-rank ratio (Su et al., 2023). 4. NPR: normalized perturbed log-rank (Su et al., 2023). 5. DNA-GPT: WScore (Yang et al., 2023). 6. Likelihood: mean log probabilities (Solaiman et al., 2019; Hashimoto et al., 2019; Gehrmann et al., 2019). 7. LogRank: averaged log-rank in descending order by probabilities (Gehrmann et al., 2019). 8. Entropy: mean token entropy of the predictive distribution (Gehrmann et al., 2019; Solaiman et al., 2019; Ippolito et al., 2019). 9. RoBERTa-base: a pre-trained classifier (Liu et al., 2019). 10. RoBERTa-large: a larger pre-trained classifier with more layers and parameters (Liu et al., 2019). The first eight score functions calculate scores based on certain statistical properties of texts, with each text's score computed via a language model. The last two score functions compute scores by using some pre-trained classifiers. For the reader's convenience, more details about the implementation of the score functions $\phi(\cdot)$ are provided in Appendix B.

**LLMs and Datasets.** Our experiments focus on the black-box setting (Bao et al., 2023), which means that if $x$ is generated by a model $q_s$, i.e., $x \sim q_s$, a different model $p_\theta$ will then be used to evaluate the metrics such as the log-probability $\log p_\theta(x)$ when calculating $\phi(x)$. The models $q_s$ and $p_\theta$ are respectively called the "source model" and "scoring model" for clarity. The black-box setting is a relevant scenario in practice because the source model used for generating the texts to be inferred is likely unknown in practice, which makes it elusive to use the same model to compute the scores. We construct a dataset that contains some real news and fake ones generated by LLMs for 2024 Olympics. Specifically, we collect 500 news about Paris 2024 Olympic Games from its official website (Olympics, 2024) and then use three source models, Gemini-1.5-Flash, Gemini-1.5-Pro (Google Cloud, 2024a), and PaLM 2 (Google Cloud, 2024b; Chowdhery et al., 2023) to generate an equal number of fake news based on the first 30 tokens of each real one respectively. Two scoring models for computing the text scores $\phi(\cdot)$ are considered, which are GPT-Neo-2.7B (Neo-2.7) (Black et al., 2021) and Gemma-2B (Google, 2024). The perturbation model that is required for the score function DetectGPT and NPR is T5-3B (Raffel et al., 2020). For Fast-DetectGPT, the sampling model is GPT-J-6B (Wang & Komatsuzaki, 2021) when scored with Neo-2.7, and Gemma-2B when the scoring model is Gemma-2B. We sample human-written text $x_t$ from a pool of 500 news articles from the XSum dataset (Narayan et al., 2018). We emphasize that we also consider existing datasets from Bao et al. (2023) for the experiments. Details can be found in

Appendix H, which also demonstrate the effectiveness of our method.

**Baselines.** Our method are compared with two baselines, which adapt the fixed-time permutation test (Good, 2013) to the scenario of the sequential hypothesis testing. Specifically, the first baseline conducts a permutation test after collecting every $k$ samples. If the result does not reject $H_0$, then it will wait and collect another $k$ samples to conduct another permutation test on this new batch of $k$ samples. This process is repeated until $H_0$ is rejected or the time $t$ runs out (i.e., when $t = T$). The significance level parameter of the permutation test is set to be the same constant $\alpha$ for each batch, which does not maintain a valid level-$\alpha$ test overall. The second baseline is similar to the first one except that the significance level parameter for the $i$-th batch is set to be $\alpha/2^i$, with $i$ starting from 1, which aims to ensure that the cumulative type-I error is bounded by $\alpha$ via the union bound. The detailed process of the baselines is described in Appendix G.

**Parameters of Our Algorithm.** All the experiments in the following consider the setting of the composite hypothesis. For the step size $\gamma$, we simply follow the related works (Cutkosky & Orabona, 2018; Chugg et al., 2023; Shekhar & Ramdas, 2023) and let $1/\gamma = 2/(2 - \ln 3)$. We consider two scenarios of sequential hypothesis testing in the experiments. The first scenario (oracle) assumes that one has prior knowledge of $d_t$ (or $d_*$) and $\epsilon$, and the performance of our algorithm in this case could be considered as an ideal outcome that it can achieve. For simulating this ideal scenario in the experiments, we let $\epsilon$ be the absolute difference between the mean scores of XSum texts and 2024 Olympic news, which are datasets of human-written texts. The second scenario considers that we do not have such knowledge a priori, and hence we have to estimate $d_t$ (or $d_*$) and specify the value of $\epsilon$ using the samples collected in the first few times steps, and then the hypothesis testing is started thereafter. In our experiments, we use the first 10 samples from each sequence of $x_t$ and $y_t$ and set $d_t$ to be a constant, which is twice the value of $\max_{s \leq 10} |\phi(x_s) - \phi(y_s)|$. For estimating $\epsilon$, we obtain scores for 20 texts sampled from the XSum dataset and randomly divide them into two groups, and set $\epsilon$ to twice the average absolute difference between the empirical means of these two groups across 1000 random shuffles.

**Experimental Results.** The experiments evaluate the performance of our method and baselines under both $H_0$ and $H_1$. As there is inherent randomness from the observed samples of the texts in the online setting, we repeat 1000 runs and report the average results over these 1000 runs. Specifically, we report the false positive rate (FPR) under $H_0$, which is the number of times the source of $y_t$ is incorrectly declared as an LLM when it is actually human, divided by the total of 1000 runs. We also report the average

time to reject the null under $H_1$ (denoted as Rejection Time $\tau$), which is the time our algorithm takes to reject $H_0$ and correctly identify the source when $y_t$ is indeed generated by an LLM. More precisely, the rejection time $\tau$ is the average time at which either $W_t^A$ or $W_t^B$ exceeds $2/\alpha$ before $T$; otherwise, $\tau$ is set to $T = 500$, regardless of whether it rejects $H_0$ at $T$, since the the time budget runs out. The parameter value $d_t$ in Scenario 1 (oracle) is shown in Table 2, and the value for $\epsilon$ can be found in Table 1 in the appendix. For the estimated $\epsilon$ and $d_t$ of each sequential testing in Scenario 2, they are displayed in Table 5 and Table 6 respectively in the appendix. Our method and the baselines require specifying the significance level parameter $\alpha$. In our experiments, we try 20 evenly spaced values of the significance level parameter $\alpha$ that ranges from $0.005$ to $0.1$ and report the performance of each one.

Figure 2 shows the performance of our algorithm with different score functions under Scenario 1 (oracle). Our algorithm consistently controls FPRs below the significance levels $\alpha$ and correctly declare the unknown source as an LLM before $T = 500$ for all score functions. This includes using the Neo-2.7 or Gemma-2B scoring models to implement eight of these score functions that require a language model. On the plots, each marker represents the average results over 1000 runs of our algorithm with a specific score function $\phi(\cdot)$ under different values of the parameter $\alpha$. The subfigures on the left in Figure 2a and 2b show False Positive Rate (under $H_0$) versus Rejection Time (under $H_1$); therefore, a curve that is closer to the bottom-left corner is more preferred. From the plot, we can see that the configurations of our algorithm with the score function being Fast-DetectGPT, DetectGPT, or Likelihood have the most competitive performance. When the unknown source is an LLM, they can detect it at time around $t = 100$ on average, and the observation is consistent under different language models used for the scoring. The subfigures on the right in Figure 2 show that the FPR is consistently bounded by the chosen value of the significance level parameter $\alpha$.

Figure 3 shows the empirical results of our algorithm under Scenario 2, where it has to use the first few samples to specify $d_t$ and $\epsilon$ before starting the algorithm. Under this scenario, our algorithm equipped with most of the score functions still performs effectively. We observe that our algorithm with 1) Fast-DetectGPT as the score function $\phi(\cdot)$ and Neo-2.7 as the language model for computing the score, and with 2) Likelihood as the score function $\phi(\cdot)$ and Gemma-2B for computing the value of $\phi(\cdot)$ have the best performance under this scenario. Compared to the first case where the oracle of $d_t$ and $\epsilon$ is available and exploited, these two configurations only result in a slight degradation of the performance under Scenario 2, and we note that our algorithm can only start updating after the first 10 time steps under this scenario. We observe that the bound

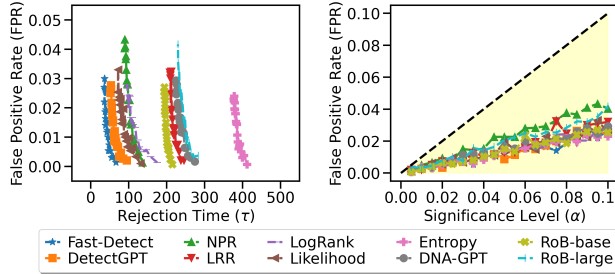

(a) Averaged results with text $x_t$ sampled from XSum and $y_t$ from 2024 Olympic news or machine-generated news, across three source models. The scoring model is Neo-2.7.

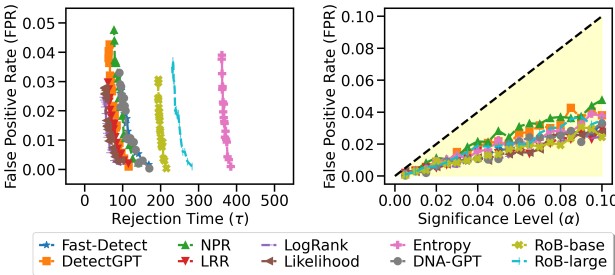

(b) Averaged results with text $x_t$ sampled from XSum and $y_t$ from 2024 Olympic news or machine-generated news, across three source models. The scoring model is Gemma-2B.

*Figure 2.* Averaged results of Scenario 1 (oracle), which shows the average of the rejection time under $H_1$ (i.e., the average time to detect LLMs) and the false positive rate under $H_0$ for 10 different score functions and 20 different values of the significance level parameter $\alpha$. Here, three source models (Gemini-1.5-Flash, Gemini-1.5-Pro and PaLM 2) are used to generate an equal number of the machine-generated texts, and two different scoring models (Neo-2.7 and Gemma-2B) are used for computing the function value of the score functions. The left subfigure in each panel (a) and (b) shows the average time to correctly declare an LLM versus the average false positive rates over 1000 runs for each $\alpha$. Thus, plots closer to the bottom-left corner are better, as they indicate correct detection of an LLM with shorter rejection time and a lower FPR. In the right subfigure of each panel, the black dashed line along with the shaded area illustrates the desired FPRs. Our algorithm under various configurations consistently has an FPR smaller than the value of the significance level parameter $\alpha$.

of $d_t$ that we estimated using the samples collected from first 10 time steps is significantly larger than the tightest bound of $d_t$ in most of the runs where we refer the reader to Table 2, 6 in the appendix for details, which explains why most of the configurations under Scenario 2 need a longer time to detect LLMs, as predicted by our propositions. We also observe that the configurations with two supervised classifiers (RoBERTa-based and RoBERTa-large) and the combinations of a couple of score functions and the scoring model Gemma-2B do not strictly control FPRs across all significance levels. This is because the estimated $d_t$ for these score functions is not large enough to ensure that the

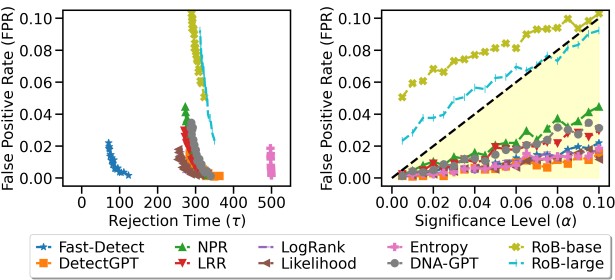

(a) Averaged results with text $x_t$ sampled from XSum and $y_t$ from 2024 Olympic news or machine-generated news, across three source models. The scoring model is Neo-2.7.

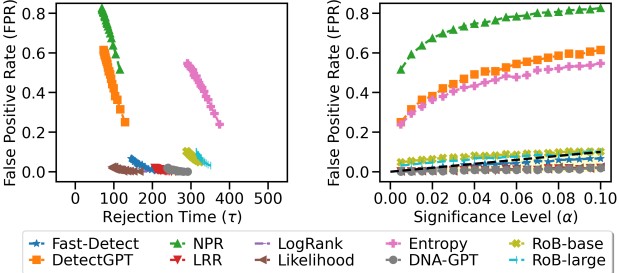

(b) Averaged results with text $x_t$ sampled from XSum and $y_t$ from 2024 Olympic news or machine-generated news, across three source models. The scoring model is Gemma-2B.

*Figure 3.* Averaged results of Scenario 2, where our algorithm has to use the first few samples to specify $d_t$ and $\epsilon$ before starting the algorithm. The plots are about the average of the rejection time under $H_1$ (i.e., the average time to detect LLMs) and the false positive rate under $H_0$ for 10 different score functions and 20 different values of the significance level parameter $\alpha$ when using two different scoring models, Neo-2.7 (a) and Gemma-2B (b).

wealth $W_t$ remains nonnegative at all time points $t$. That is, we observed $2d_t < |\phi(x_t) - \phi(y_t)| + \epsilon$ for some $t$ in the experiments, and hence the wealth $W_t$ is no longer a non-negative supermartingale, which prevents the application of Ville's inequality to guarantee a level-$\alpha$ test. Nevertheless, our algorithm, when using most score functions that utilize the scoring model Neo-2.7, can still effectively control type-I error and detect LLMs around $t = 300$.

In Appendix H, we provide more experimental results, including those using existing datasets from Bao et al. (2023) for simulating the sequential testing, where our algorithm on these datasets also performs effectively. Moreover, we found that the rejection time is influenced by the relative magnitude of $\Delta - \epsilon$ and $d_t - \epsilon$, as predicted by our propositions, and the details are provided in Appendix G. From the experimental results, when the knowledge of $d_t$ and $\epsilon$ is not available beforehand, as long as the estimated $d_t$ and $\epsilon$ guarantee a nonnegative supermartingale, and the estimated $\epsilon$ lies between the actual mean score difference of human-written texts and that of human-written vs LLM-generated texts, our algorithm can maintain a sequential valid level-$\alpha$

test and efficiently detect LLMs.

**Comparisons with Baselines.** In this part, we use the score function of Fast-DetectGPT and scoring model Neo-2.7 to get text scores, and then compare the performance of our method with two baselines that adapt the fixed-time permutation test to the sequential hypothesis setting. Batch sizes $k \in \{25, 50, 100, 250, 500\}$ are considered for the baselines. We set the estimated $\epsilon$ and $d_t$ values the same as in Scenario 2. The baselines are also implemented in a manner to conduct the composite hypothesis test.

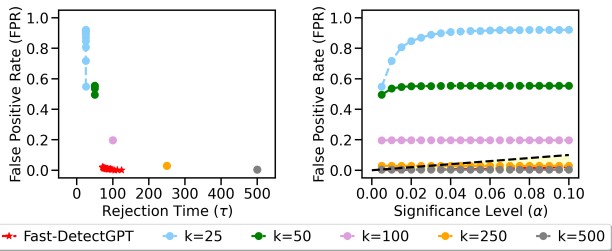

(a) Comparison between our method and the baseline that sets the value of the significance level parameter to be the same constant $\alpha$ for every batch.

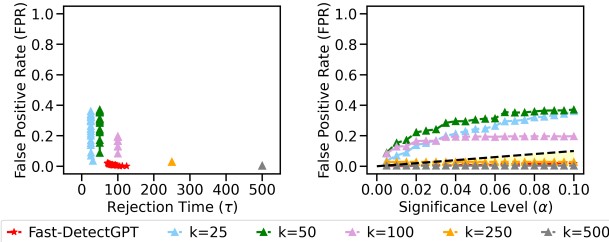

(b) Comparison between our method and the baseline that sets the value of the significance level parameter for the $i$-th batch test to be $\alpha/2^i$.

*Figure 4.* Comparison of the average results under Scenario 2, where one has to use the first 10 samples to specify $\epsilon$ and/or $d_t$ before starting the algorithm. Human-written text $x_t$ are sampled from XSum dataset, while $y_t$ is from 2024 Olympic news (under $H_0$) or machine-generated news (under $H_1$). Fake news are generated by three source models: Gemini-1.5-Flash, Gemini-1.5-Pro and PaLM 2. We report the case when the score function is Fast-DetectGPT and the scoring model is Neo-2.7 for our algorithm.

We observe a significant difference between the scores of XSum texts and machine-generated news, which causes the baselines of the permutation test to reject the null hypothesis most of the time immediately after receiving the first batch. This in turn results in the nearly vertical lines observed in the left subfigure of Figure 4a and Figure 4b, where the averaged rejection time across 1000 repeated tests closely approximates the batch size $k$ for each of the 20 significance levels $\alpha$. On the other hand, we observe that when $H_0$ is true, the baselines might not be a valid $\alpha$-test, even with a corrected significance level. This arises from the increased variability of text scores introduced by smaller batch sizes,

which results in observed absolute differences in means that may exceed the $\epsilon$ value under $H_0$. Our method can quickly detect an LLM while keeping the false positive rates (FPRs) below the specified significance levels, which is a delicate balance that can be difficult for the baselines to achieve. Without prior knowledge of the $\epsilon$ value, the baselines of permutation tests may fail to control the type-I error with small batch sizes and cannot quickly reject the null hypothesis while ensuring that FPRs remain below $\alpha$, unlike our method.

**Extensions.** We further consider (i) an experiment where the sequential texts are produced by various LLMs instead of a single LLM, and (ii) a scenario that the unknown source publishes both human-written and LLM-generated texts, where the null hypothesis becomes that all texts from the unknown source are human-written, and the alternative hypothesis posits that not all texts are human-written. We refer the reader to Appendix I for the encouraging results.

## 5. Conclusion

In this paper, we demonstrate that our algorithm, which builds on the score functions of offline detectors, can rapidly determine whether a stream of text is generated by an LLM and provides strong statistical guarantees. Specifically, it controls the type-I error rate below the significance level, ensures that the source of LLM-generated texts can eventually be identified, and guarantees an upper bound on the expected time to correctly declare the unknown source as an LLM under a mild assumption. Although the choice of detector can influence the algorithm's performance and some parameters related to text scores need to be predefined before receiving the text, our experimental results show that most of the existing detectors we use provide effective score functions, and our method performs well in most cases when using estimated values of parameters based on text scores from the first few time steps. To further enhance its efficacy, it may be worthwhile to design score functions tailored to the sequential setting, improve parameter estimations with scores from more time steps, and study the trade-offs between delaying the start of testing and obtaining more reliable estimates. Moreover, our algorithm could potentially be used as an effective tool to promptly identify and mitigate the misuse of LLMs for text generation, such as monitoring social media accounts that disseminate harmful information generated by LLMs, rapidly detecting sources of LLM-generated fake news on public websites, and identifying users who post machine-generated comments to manipulate public opinion. Exploring these applications could be a promising direction.

Finally, we note that a recent work of Chen & Wang (2025) introduces optimistic interior-point methods that enable updates over the full interior of the decision space. Incorporating such advanced betting strategies into the testing-by-

betting framework presents a promising future direction for improving online detection of machine-generated texts.

## Acknowledgements

The authors gratefully acknowledge the support provided by the Gemma Academic Program and Google Cloud Credits.

## Impact Statement

This paper presents work whose goal is to advance the field of Machine Learning. There are many potential societal consequences of our work, none which we feel must be specifically highlighted here.

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

# A. More Related Works

## A.1. Related Works of Detecting Machine-Generated Texts

Some methods distinguish between human-written and machine-generated texts by comparing their statistical properties (Jawahar et al., 2020). Lavergne et al. (2008) introduced a method, which uses relative entropy scoring to effectively identify texts from Markovian text generators. Perplexity is also a metric for detection, which quantifies the uncertainty of a model in predicting text sequences (Hashimoto et al., 2019). Gehrmann et al. (2019) developed GLTR tool, which leverages statistical features such as per-word probability, rank, and entropy, to enhance the accuracy of fake-text detection. Mitchell et al. (2023) created a novel detector called DetectGPT, which identifies a machine-generated text by noting that it will exhibit higher log-probability than samples where some words of the original text have been rewritten/perturbed. Su et al. (2023) then introduced two advanced methods utilizing two metrics: Log-Likelihood Log-Rank Ratio (LRR) and Normalized Perturbed Log Rank (NPR), respectively. Bao et al. (2023) developed Fast-DetectGPT, which replaces the perturbation step of DetectGPT with a more efficient sampling operation. Solaiman et al. (2019) employed the classic logistic regression model on TF-IDF vectors to detect texts generated by GPT-2, and noted that texts from larger GPT-2 models are more challenging to detect than those from smaller GPT-2 models. Researchers have also trained supervised models on neural network bases. Bakhtin et al. (2019) found that Energy-based models (EBMs) outperform the behavior of using the original language model log-likelihood in real and fake text discrimination. Zellers et al. (2019) developed a robust detection method named GROVER by using a linear classifier, which can effectively spot AI-generated 'neural' fake news. Ippolito et al. (2019) showed that BERT-based (Devlin, 2018) classifiers outperform humans in identifying texts characterized by statistical anomalies, such as those where only the top k high-likelihood words are generated, yet humans excel at semantic understanding. Solaiman et al. (2019) fine-tuned RoBERTa (Liu et al., 2019) on GPT-2 outputs and achieved approximately 95% accuracy in detecting texts generated by 1.5 billion parameter GPT-2. The effectiveness of RoBERTa-based detectors is further validated across different text types, including machine-generated tweets (Fagni et al., 2021), news articles (Uchendu et al., 2020), and product reviews (Adelani et al., 2020). Other supervised classifiers, such as GPTZero (Tian, 2023) and OpenAI's Classifier (OpenAI, 2023), have also proven to be strong detectors. Moreover, some research has explored watermarking methods that embed detectable patterns in LLM-generated texts for identifying, see e.g., Jalil & Mirza (2009); Kamaruddin et al. (2018); Abdelnabi & Fritz (2021); Zhao et al. (2023); Kirchenbauer et al. (2023); Christ et al. (2024). Recently, Kobak et al. (2024) introduced "excess word usage", a novel data-driven approach that identifies LLM usage in academic writing and avoids biases that could be potentially introduced by generation prompts from traditional human text datasets.

## A.2. Related Works of Sequential Hypothesis Testing by Betting

Kelly (1956) first proposed a strategy for sequential betting with initial wealth on the outcome of each coin flip $g_t$ in round $t$, which can take values of $+1$ (head) or $-1$ (tail), generated i.i.d. with the probability of heads $p \in [0, 1]$. It is shown that betting a fraction $\beta_t = 2p - 1$ on heads in each round will yield more accumulated wealth than betting any other fixed fraction of the current wealth in the long run. Orabona & Pál (2016) demonstrated the equivalence between the minimum wealth of betting and the maximum regret in one-dimensional Online Linear Optimization (OLO) algorithms, which introduces the coin-betting abstraction for the design of parameter-free algorithms. Based on this foundation, Cutkosky & Orabona (2018) developed a coin betting algorithm, which uses an exp-concave optimization approach through the Online Newton Step (ONS). Subsequently, Shekhar & Ramdas (2023) applied their betting strategy, along with the general principles of testing by betting as clarified by Shafer (2021), to nonparametric two-sample hypothesis testing. Chugg et al. (2023) then conducted sequentially audits on both classifiers and regressors within the general two-sample testing framework established by Shekhar & Ramdas (2023), which demonstrate that this method remains robust even in the face of distribution shifts. Additionally, other studies (Orabona & Jun, 2023; Waudby-Smith & Ramdas, 2024) have developed practical strategies that leverage online convex optimization methods, with which the betting fraction can be adaptively selected to provide statistical guarantees for the results. We also note that sequential hypothesis testing by betting has drawn significant attention in recent years, e.g., Vovk & Wang (2021); Shaer et al. (2023); Ramdas et al. (2023); Podkopaev & Ramdas (2023); Pandeva et al. (2024); Podkopaev et al. (2023); Teneggi & Sulam (2024); Grünwald et al. (2024); Podkopaev et al. (2024); Cho et al. (2024); Fischer & Ramdas (2024); Bar et al. (2024).

# B. Related Score Functions

The score function $\phi : \text{Text} \to \mathbb{R}$ will take a text as input and then output a real number. It is designed to maximize the ability to distinguish machine text from human text, that is, we want the score function to maximize the difference in scores between human-written text and machine-generated text.

**DetectGPT.** Three models: source model, perturbation model and scoring model are considered in the process of calculating the score $\phi(x)$ of text $x$ by the metric of DetectGPT (the normalized perturbation discrepancy) (Mitchell et al., 2023). Firstly, the original text $x$ is perturbed by a perturbation model $q_\zeta$ to generate $m$ perturbed samples $\tilde{x}^{(i)} \sim q_\zeta(\cdot|x), i \in [1, 2, \cdots, m]$, then the scoring model $p_\theta$ is used to calculate the score

$$\phi(x) = \frac{\log p_\theta(x) - \tilde{\mu}}{\tilde{\sigma}}, \tag{2}$$

where $\tilde{\mu} = \frac{1}{m}\sum_{i=1}^{m} \log p_\theta(\tilde{x}^{(i)})$, and $\tilde{\sigma}^2 = \frac{1}{m}\sum_{i=1}^{m}\left[\log p_\theta(\tilde{x}^{(i)}) - \tilde{\mu}\right]^2$. We can write $\log p_\theta(x)$ as $\sum_{j=1}^{n} \log p_\theta(x_j|x_{1:j-1})$, where $n$ denotes the number of tokens of $x$, $x_j$ denotes the $j$-th token, and $x_{1:j-1}$ means the first $j-1$ tokens. Similarly, $\log p_\theta(\tilde{x}^{(i)}) = \sum_{j=1}^{\tilde{n}^{(i)}} \log p_\theta(\tilde{x}_j^{(i)}|\tilde{x}_{1:j-1}^{(i)})$, where $\tilde{n}^{(i)}$ is the number of tokens of $i$-th perturbed sample $\tilde{x}^{(i)}$.

**Fast-DetectGPT.** Bao et al. (2023) considered three models: source model, sampling model and scoring model for the metric of Fast-DetectGPT (conditional probability curvature). The calculation is conducted by first using the sampling model $q_\zeta$ to generate alternative samples that each consist of $n$ tokens. For each token $\tilde{x}_j$, it is sampled conditionally on $x_{1:j-1}$, that is, $\tilde{x}_j \sim q_\zeta(\cdot|x_{1:j-1})$ for $j = 1, \cdots, n$. The sampled text is $\tilde{x} = (\tilde{x}_1, \cdots, \tilde{x}_n)$. Then, the scoring model $p_\theta$ is used to calculate the logarithmic conditional probability of the text, given by $\sum_{j=1}^{n} \log p_\theta(x_j|x_{1:j-1})$, and then normalize it, where $n$ denotes the number of tokens of $x$. This score function is quantified as

$$\phi(x) = \frac{\sum_{j=1}^{n} \log p_\theta(x_j|x_{1:j-1}) - \tilde{\mu}}{\tilde{\sigma}}. \tag{3}$$

There are two ways to calculate the mean value $\tilde{\mu}$ and the corresponding variance $\tilde{\sigma}^2$, one is to calculate the population mean

$$\tilde{\mu} = \mathbb{E}_{\tilde{x}}\left[\sum_{j=1}^{n} \log p_\theta(\tilde{x}_j^{(i)}|x_{1:j-1})\right] = \sum_{j=1}^{n} \mathbb{E}_{\tilde{x}}\left[\log p_\theta(\tilde{x}_j^{(i)}|x_{1:j-1})\right]$$

$$= \sum_{j=1}^{n}\sum_{i=1}^{s} q_\zeta(\tilde{x}_j^{(i)}|x_{1:j-1}) \cdot \log p_\theta(\tilde{x}_j^{(i)}|x_{1:j-1}),$$

if we denote $\sum_{i=1}^{s} q_\zeta(\tilde{x}_j^{(i)}|x_{1:j-1}) \cdot \log p_\theta(\tilde{x}_j^{(i)}|x_{1:j-1})$ as $\tilde{\mu}_j$, then the variance is

$$\tilde{\sigma}^2 = \mathbb{E}_{\tilde{x}}\left[\sum_{j=1}^{n}\left(\log p_\theta(\tilde{x}_j^{(i)}|x_{1:j-1}) - \tilde{\mu}_j\right)^2\right] = \mathbb{E}_{\tilde{x}}\left[\sum_{j=1}^{n}\left(\log^2 p_\theta(\tilde{x}_j^{(i)}|x_{1:j-1}) - \tilde{\mu}_j^2\right)\right]$$

$$= \sum_{j=1}^{n}\left(\sum_{i=1}^{s} q_\zeta(\tilde{x}_j^{(i)}|x_{1:j-1}) \cdot \log^2 p_\theta(\tilde{x}_j^{(i)}|x_{1:j-1}) - \tilde{\mu}_j^2\right),$$

where $\tilde{x}_j^{(i)}$ denotes the $i$-th generated sample for the $j_{th}$ token of the text $x$, $q_\zeta(\tilde{x}_j^{(i)}|x_{1:j-1})$ is the probability of this sampled token given by the sampling model according to the probability distribution of all possible tokens at position $j$, conditioned on the first $j-1$ tokens of $x$. Besides, $p_\theta(\tilde{x}_j^{(i)}|x_{1:j-1})$ is the conditional probability of $\tilde{x}_j^{(i)}$ evaluated by the scoring model, $s$ represents the total number of possible tokens at each position, corresponding to the size of the entire vocabulary used by the sampling model. We can use the same value at each position in the formula, as the vocabulary size remains consistent across positions. The sample mean and the variance can be considered in practice

$$\tilde{\mu} = \frac{1}{m}\sum_{i=1}^{m}\sum_{j=1}^{n} \log p_\theta(\tilde{x}_j^{(i)}|x_{1:j-1}), \quad \tilde{\sigma}^2 = \frac{1}{m}\sum_{j=1}^{n}\sum_{i=1}^{m}\left(\log p_\theta(\tilde{x}_j^{(i)}|x_{1:j-1}) - \tilde{\mu}_j^2\right),$$

where $\tilde{\mu}_j = \frac{1}{m} \sum_{i=1}^{m} \log p_\theta(\tilde{x}_j^{(i)}|x_{1:j-1})$. In this case, the sampling model is just used to get samples $\tilde{x}$. By sampling a substantial number of texts ($m = 10,000$), we can effectively map out the distribution of their $\log p_\theta(\tilde{x}_j|x_{1:j-1})$ values according to (Bao et al., 2023).

**NPR.** The definition of Normalized Log-Rank Perturbation (NPR) involves the perturbation operation of DetectGPT (Su et al., 2023). The scoring function of NPR is

$$\phi(x) = \frac{\frac{1}{m} \sum_{i=1}^{m} \log r_\theta(\tilde{x}^{(i)})}{\log r_\theta(x)},$$

where $r_\theta(x)$ represents the rank of the original text evaluated by the scoring model, $m$ perturbed samples $\tilde{x}^{(i)}, i \in [1, 2, \cdots, m]$ are generated based on $x$. The $\log r_\theta(x)$ is calculated as $\frac{1}{n} \sum_{j=1}^{n} \log r_\theta(x_j|x_{1:j-1})$, where $n$ denotes the number of tokens of $x$. Similarly, $\log r_\theta(\tilde{x}^{(i)}) = \frac{1}{\tilde{n}^{(i)}} \sum_{j=1}^{\tilde{n}^{(i)}} \log r_\theta(\tilde{x}_j^{(i)}|\tilde{x}_{1:j-1}^{(i)})$, where $\tilde{n}^{(i)}$ is the number of tokens of perturbed sample $\tilde{x}^{(i)}$ generated by the perturbation model $q_\zeta$.

**LRR.** The score function of Log-Likelihood Log-Rank Ratio (LRR) (Su et al., 2023) consider both logarithmic conditional probability and the logarithmic conditional rank evaluated by the scoring model for text

$$\phi(x) = \left| \frac{\frac{1}{n} \sum_{j=1}^{n} \log p_\theta(x_j|x_{1:j-1})}{\frac{1}{n} \sum_{j=1}^{n} \log r_\theta(x_j|x_{1:j-1})} \right| = -\frac{\sum_{j=1}^{n} \log p_\theta(x_j|x_{1:j-1})}{\sum_{j=1}^{n} \log r_\theta(x_j|x_{1:j-1})},$$

where $r_\theta(x_j|x_{1:j-1}) \geq 1$ is the rank of $x_i$, conditioned on its previous $j-1$ tokens. We suppose that the total number of tokens of $x$ is $n$.

**Likelihood.** The Likelihood (Solaiman et al., 2019; Hashimoto et al., 2019; Gehrmann et al., 2019) for a text $x$ which has $n$ tokens can be computed by averaging the log probabilities of each token conditioned on the previous tokens in the text given its preceding context evaluated by the scoring model:

$$\phi(x) = \frac{1}{n} \sum_{j=1}^{n} \log p_\theta(x_j|x_{1:j-1}).$$

**LogRank.** The LogRank (Gehrmann et al., 2019), is defined by firstly using the scoring model to determine the rank of each token's probability (with respect to all possible tokens at that position) and then taking the average of the logarithm of these ranks:

$$\phi(x) = \frac{1}{n} \sum_{j=1}^{n} \log r_\theta(x_j|x_{1:j-1}).$$

**Entropy.** Entropy measures the uncertainty of the predictive distribution for each token (Gehrmann et al., 2019). The score function is defined as:

$$\phi(x) = -\frac{1}{s} \sum_{j=1}^{n} \sum_{i=1}^{s} p_\theta(x_j^{(i)}|x_{1:j-1}) \cdot \log p_\theta(x_j^{(i)}|x_{1:j-1}),$$

where $p_\theta(x_j^{(i)}|x_{1:j-1})$ denotes the probability of each possible token $x^{(i)}$ at position $j$ evaluated by the scoring model, given the preceding context $x_{1:j-1}$. The inner sum computes the entropy for each token's position by summing over all $s$ possible tokens.

**DNA-GPT.** The score function of DNA-GPT is calculated by WScore, which compares the differences between the original and new remaining parts through probability divergence (Yang et al., 2023). Given the truncated context $z$ based on text $x$ and a series of texts sampled by scoring model $p_\theta$ based on $z$, denoted as $\{\tilde{x}^{(1)}, \tilde{x}^{(2)}, \ldots, \tilde{x}^{(m)}\}$. WScore is defined as:

$$\phi(x) = \log p_\theta(x) - \frac{1}{m} \sum_{i=1}^{m} \log p_\theta(\tilde{x}^{(i)}),$$

where we need to note that $x \sim q_s(\cdot)$, $\tilde{x}^{(i)} \sim p_\theta(\cdot|z)$ for $i = \{1, 2, \cdots, m\}$. This formula calculates the score of $x$ by comparing the logarithmic probability differences between the text $x$ and the averaged results of $m$ samples generated by the scoring model under the context $z$ which is actually the truncated $x$. Here, we can write the $\log p_\theta(x)$ as $\sum_{j=1}^{n} \log p_\theta(x_j|x_{1:j-1})$, which is the summation of the logarithm conditional probability of each token conditioned on the previous tokens, assuming the total number of non-padding tokens for $x$ is $n$. The calculation is similar for $\log p_\theta(\tilde{x}^{(i)})$, we just need to replace $x$ in $\log p_\theta(x)$ by $\tilde{x}^{(i)}$.

**RoBERTa-base/large.** The supervised classifiers RoBERTa-base and RoBERTa-large (Liu et al., 2019) use the softmax function to compute a score for text $x$. Two classes are considered: "class = 0" represents text generated by a GPT-2 model, while "class = 1" represents text not generated by a GPT-2 model. The score for text $x$ is defined as the probability that it is classified into class 0 by the classifier, computed as:

$$\phi(\text{class} = 0|x) = \frac{e^{z_0}}{e^{z_0} + e^{z_1}},$$

where $z_j$ is the logits of $x$ corresponding to class $j \in \{0, 1\}$, provided by the output of the pre-trained model.

## C. Proof of Regret Bound of ONS

---

**Algorithm 2** Online Newton Step (Hazan et al., 2016)

---

**Require:** parameter $\gamma$; **Init:** $\theta_1 \leftarrow 0$, $a_0 \leftarrow 1$.

 1: **for** $t = 1$ to $T$ **do**
 2:      Receive loss function $\ell_t : \mathcal{K}_t \to \mathbb{R}$.
 3:      Compute $z_t = \nabla \ell_t(\theta_t)$, $a_t = a_{t-1} + z_t^2$.
 4:      Update via Online Newton Step:

$$\beta_{t+1} = \theta_t - \frac{1}{\gamma} \cdot \frac{z_t}{a_t}.$$

 5:      Get a hint of $d_{t+1}$ to update the decision space $\mathcal{K}_{t+1} \leftarrow [-\frac{1}{2d_{t+1}}, \frac{1}{2d_{t+1}}]$.
 6:      Project $\beta_{t+1}$ to $\mathcal{K}_{t+1}$:

$$\theta_{t+1} = \text{proj}_{\mathcal{K}_{t+1}}(\beta_{t+1}) = \arg \min_{\theta \in \mathcal{K}_{t+1}} (\beta_{t+1} - \theta)^2.$$

 7: **end for**

---

Under $H_0$, i.e., $\mu_x = \mu_y$, the wealth is a P-supermartingale. The value of $g_t$ are constrained to the interval $[-1, 1]$ in previous works (Orabona & Pál, 2016; Cutkosky & Orabona, 2018; Chugg et al., 2023) with the scores $\phi(x_t)$ and $\phi(y_t)$ each range from $[0, 1]$. To ensure that wealth $W_t$ remains nonnegative and to establish the regret bound by ONS, $\theta_t$ is always selected within $[-1/2, 1/2]$. In our setting, however, the actual range of score difference between two texts, denoted as $g_t = \phi(x_t) - \phi(y_t)$, is typically unknown beforehand. If we assume that the ranges for both $\phi(x_t)$ and $\phi(y_t)$ are $[m_t, n_t]$, then the range of their difference $g_t := \phi(y_t) - \phi(x_t)$ is symmetric about zero, and is given by the interval $[-(n_t - m_t), ; n_t - m_t]$. We denote an upper bound $d_t \geq |n_t - m_t|$ and express the range as $g_t \in [-d_t, d_t]$, where $d_t \geq 0$. Then, choosing $\theta_t$ within $[-1/2d_t, 1/2d_t]$ can guarantee that the wealth is a nonnegative P-supermartingale. If we consider the condition of either $W_t \geq 1/\alpha$ or $W_T \geq Z/\alpha$ for any stopping time $T$ as the indication to "reject $H_0$" and apply the randomized Ville's inequality (Ramdas & Manole, 2023), the type-I error can be controlled below the significance level $\alpha$ when $H_0$ is true.

Under $H_1$, i.e., $\mu_x \neq \mu_y$, our goal is to select a proper $\theta_t$ at each round $t$ that can speed up the wealth accumulation. It allows us to declare the detection of an LLM once the wealth reaches the specified threshold $1/\alpha$. We can choose $\theta_t$ recursively following Algorithm 2. This algorithm can guarantee the regret upper bound for exp-concave loss. Following the proof of Theorem 4.6 in (Hazan et al., 2016), we can derive the bound on the regret. The regret of choosing $\theta_t \in \mathcal{K}_t$ after $T$ time steps by Algorithm 2 is defined as

$$\text{Regret}_T(\text{ONS}) := \sum_{t=1}^{T} \ell_t(\theta_t) - \sum_{t=1}^{T} \ell_t(\theta^*),$$

where the loss function for each $t$ is $\ell_t : \mathcal{K}_t \to \mathbb{R}$, and the best decision in hindsight is defined as $\theta^* \in \arg\min_{\theta \in \mathcal{K}_*} \sum_{t=1}^{T} \ell_t(\theta)$, where $\mathcal{K}_* = \bigcap_{t=1}^{T} \mathcal{K}_t$.

**Lemma C.1.** *Let $\ell_t : \mathcal{K}_t \to \mathbb{R}$ be an $\alpha$-exp-concave function for each $t$. Let $D_t$ represent the diameter of $\mathcal{K}_t$, and $G_t$ be a bound on the (sub)gradients of $\ell_t$. Algorithm 2, with parameter $\gamma = \frac{1}{2}\min\left\{\frac{1}{G_t D_t}, \alpha\right\}$ and $a_0 = 1/\gamma^2 D_1^2$, guarantees:*

$$\text{Regret}_T(\text{ONS}) \leq \frac{1}{2\gamma}\left(\sum_{t=1}^{T}\frac{z_t^2}{a_t} + 1\right), \tag{4}$$

*where $G_t D_t$ is a constant for all $t$, and $z_t$, $a_t$, and $\mathcal{K}_t$ are defined in Algorithm 2.*

**Remark 3.** For any $\alpha$-exp-concave function $\ell_t(\cdot)$, if we let a positive number $\gamma$ be $\frac{1}{2}\min\left\{\frac{1}{G_t D_t}, \alpha\right\}$, and initialize $a_0 = 1/\gamma^2 D_1^2$ at the beginning, the above inequality (4) will always hold for choosing the fraction $\theta_t$ by Algorithm 2. That is, the accumulated regret after $T$ time steps, defined as the difference between the cumulative loss from adaptively choosing $\theta_t$ by this ONS algorithm and the minimal cumulative loss achievable by the optimal decision $\theta^*$ at each time step, is bounded by the right-hand side of (4). To prove this lemma, we need to first prove Lemma C.2.

**Lemma C.2.** (Lemma 4.3 in Hazan et al. (2016)) *Let $f : \mathcal{K} \to \mathbb{R}$ be an $\alpha$-exp-concave function, and $D, G$ denote the diameter of $\mathcal{K}$ and a bound on the (sub)gradients of $f$ respectively. The following holds for all $\gamma = \frac{1}{2}\min\left\{\frac{1}{GD}, \alpha\right\}$ and all $\theta, \beta \in \mathcal{K}$:*

$$f(\theta) \geq f(\beta) + \nabla f(\beta)^\top (\theta - \beta) + \frac{\gamma}{2}(\theta - \beta)^\top \nabla f(\beta) \nabla f(\beta)^\top (\theta - \beta). \tag{5}$$

**Remark 4.** For any $\alpha$-exp-concave function $f(\cdot)$, if we let a positive number $\gamma = \frac{1}{2}\min\left\{\frac{1}{GD}, \alpha\right\}$, the above equation (5) will hold for any two points within the domain $\mathcal{K}$ of $f(\cdot)$. This inequality remains valid even if $\gamma > 0$ is set to a smaller value than this minimum, although doing so will result in a looser regret bound. At time $t$ in Algorithm 2, the diameter of the loss function $\ell_t : \mathcal{K}_t \to \mathbb{R}$ is $D = 1/d_t$, since $\mathcal{K}_t = [-1/2d_t, 1/2d_t]$. Additionally, the bound on the gradient of $\ell_t(\theta)$ is $G_t = \max_{\theta \in \mathcal{K}_t} \nabla \ell_t(\theta)$. If $\ell_t(\theta)$ is $\alpha$-exp-concave function and $\gamma = 1/2\min\{d_t/G_t, \alpha\}$, then equation (5) will hold for any $\theta, \beta \in [-1/2d_t, 1/2d_t]$.

**Proof of Lemma C.2.** The composition of a concave and non-decreasing function with another concave function remains concave. Given that for all $\gamma = \frac{1}{2}\min\left\{\frac{1}{GD}, \alpha\right\}$, we have $2\gamma \leq \alpha$, the function $g(\theta) = \theta^{2\gamma/\alpha}$ composed with $f(\theta) = \exp(-\alpha f(\theta))$ is concave. Hence, the function $h(\theta) = \exp(-2\gamma f(\theta))$ is also concave. Then by the definition of concavity,

$$h(\theta) \leq h(\beta) + \nabla h(\beta)^\top (\theta - \beta). \tag{6}$$

We plug $\nabla h(\beta) = -2\gamma \exp(-2\gamma f(\beta)) \nabla f(\beta)$ into equation (6),

$$\exp(-2\gamma f(\theta) \leq \exp(-2\gamma f(\beta))[1 - 2\gamma \nabla f(\beta)^\top (\theta - \beta)].$$

Thus,

$$f(\theta) \geq f(\beta) - \frac{1}{2\gamma} \ln\left(1 - 2\gamma \nabla f(\beta)^\top (\theta - \beta)\right).$$

Since $D, G$ are previously denoted as the diameter of $\mathcal{K}$ and a bound on the (sub)gradients of $f$ respectively, which means that $D \geq |\theta - \beta|$, $G \geq \nabla f(\beta)$. Therefore, we have $|2\gamma \nabla f(\beta)(\theta - \beta)| \leq 2\gamma GD \leq 1 \Rightarrow -1 \leq 2\gamma \nabla f(\beta)(\theta - \beta) \leq 1$. According to the Taylor approximation, we know that $-\ln(1 - a) \geq a + \frac{1}{4}a^2$ holds for $a \geq -1$. The lemma is derived by considering $a = 2\gamma \nabla f(\beta)(\theta - \beta)$.

Since our problem is one-dimensional, then we can use Lemma C.2 to get the regret bound. Here shows the proof of Lemma C.1.

**Proof of Lemma C.1.** The best decision in hindsight is $\theta^* \in \arg\min_{\theta \in \mathcal{K}_*} \sum_{t=1}^{T} \ell_t(\theta)$, where $\mathcal{K}_* = \bigcap_{t=1}^{T} \mathcal{K}_t$. By Lemma C.2, we have the inequality (7) for $\gamma_t = \frac{1}{2}\min\left\{\frac{1}{G_t D_t}, \alpha\right\}$, which is

$$\underbrace{\ell_t(\theta_t) - \ell_t(\theta^*)}_{:=\text{Regret}_t(\text{ONS})} \leq \underbrace{z_t(\theta_t - \theta^*) - \frac{\gamma_t}{2}(\theta_t - \theta^*)^2 z_t^2}_{:=R_t}, \tag{7}$$

where the right hand side of the above inequality is defined as $R_t$, the left hand side is the regret of selecting $\theta_t$ via ONS at time $t$.

We sum both sides of the inequality (7) from $t = 1$ to $T$, then we get

$$\underbrace{\sum_{t=1}^{T} \ell_t(\theta_t) - \sum_{t=1}^{T} \ell_t(\theta^*)}_{:=\text{Regret}_T(\text{ONS})} \leq \sum_{t=1}^{T} R_t. \tag{8}$$

We recall that $D_t$ is defined as the diameter of $\mathcal{K}_t$, i.e., $D_t = \max_{a,b \in \mathcal{K}_t} |a - b|$, and $G_t$ is defined as a bound on the gradients of the loss function $\ell_t(\theta) = -\ln(1 - g_t\theta)$ at time $t$, i.e., $G_t = \max_{\theta_t \in \mathcal{K}_t} |\frac{d}{d\theta_t} \ell_t(\theta_t)|$. In our setting, $\mathcal{K}_t$ is $[-1/2d_t, 1/2d_t]$, thus $D_t = 1/d_t$. The gradient $z_t = \nabla \ell_t(\theta_t) = g_t/(1 - g_t\theta_t)$, so we have $G_t = \max_{\theta_t \in [-\frac{1}{2d_t}, \frac{1}{2d_t}]} \frac{|g_t|}{1 - g_t\theta_t}$. Since $g_t = \phi(x_t) - \phi(y_t) \in [-d_t, d_t]$, and the expression $\frac{|g_t|}{1 - g_t\theta_t}$ increases with the product $g_t\theta_t \in [-1/2, 1/2]$, the maximum is attained when $g_t = d_t$ and $\theta_t = \frac{1}{2d_t}$. Thus, we obtain $G_t = d_t/(1 - d_t \cdot \frac{1}{2d_t}) = 2d_t$. Above all, we get $G_t D_t = 2d_t \cdot \frac{1}{d_t} = 2$ for each $t$, and $\alpha = 1$. The value of $\gamma_t = \frac{1}{2} \min\{1/G_t D_t, \alpha\}$ remains a positive constant for all $t$. Therefore, we simply use $\gamma$ in the following proof, given that $\gamma_t$ takes the same constant value for all $t$.

According to the update rule of the algorithm: $\theta_{t+1} = \text{proj}_{\mathcal{K}_{t+1}}(\beta_{t+1})$, and the definition: $\beta_{t+1} = \theta_t - \frac{1}{\gamma} \cdot \frac{z_t}{a_t}$, we get:

$$\beta_{t+1} - \theta^* = \theta_t - \theta^* - \frac{1}{\gamma} \frac{z_t}{a_t}, \tag{9}$$

and

$$a_t(\beta_{t+1} - \theta^*) = a_t(\theta_t - \theta^*) - \frac{1}{\gamma} z_t. \tag{10}$$

We multiply (9) by (10) to get

$$(\beta_{t+1} - \theta^*)^2 a_t = (\theta_t - \theta^*)^2 a_t - \frac{2}{\gamma} z_t(\theta_t - \theta^*) + \frac{1}{\gamma^2} \frac{z_t^2}{a_t}. \tag{11}$$

Since $\theta_{t+1}$ is the projection of $\beta_{t+1}$ to $\mathcal{K}_{t+1}$, and $\theta_* \in \mathcal{K}_{t+1}$,

$$(\beta_{t+1} - \theta^*)^2 \geq (\theta_{t+1} - \theta^*)^2. \tag{12}$$

Plugging (12) into (11) and rearranging terms yields

$$z_t(\theta_t - \theta^*) \leq \frac{1}{2\gamma} \frac{z_t^2}{a_t} + \frac{\gamma}{2}(\theta_t - \theta^*)^2 a_t - \frac{\gamma}{2}(\theta_{t+1} - \theta^*)^2 a_t.$$

Summing up over $t = 1$ to $T$,

$$
\sum_{t=1}^{T} z_t(\theta_t - \theta^*)
$$

$$
\leq \sum_{t=1}^{T} \left( \frac{1}{2\gamma} \frac{z_t^2}{a_t} + \frac{\gamma}{2}(\theta_t - \theta^*)^2 a_t - \frac{\gamma}{2}(\theta_{t+1} - \theta^*)^2 a_t \right)
$$

$$
= \sum_{t=1}^{T} \frac{1}{2\gamma} \frac{z_t^2}{a_t} + \frac{\gamma}{2}(\theta_1 - \theta^*)^2 a_1 + \sum_{t=2}^{T} \frac{\gamma}{2}(\theta_t - \theta^*)^2 a_t - \sum_{t=1}^{T-1} \frac{\gamma}{2}(\theta_{t+1} - \theta^*)^2 a_t - \frac{\gamma}{2}(\theta_{T+1} - \theta^*)^2 a_T
$$

$$
= \sum_{t=1}^{T} \frac{1}{2\gamma} \frac{z_t^2}{a_t} + \frac{\gamma}{2}(\theta_1 - \theta^*)^2 a_1 + \sum_{t=2}^{T} \frac{\gamma}{2}(\theta_t - \theta^*)^2 (a_t - a_{t-1}) - \frac{\gamma}{2}(\theta_{T+1} - \theta^*)^2 a_T
$$

$$
\leq \sum_{t=1}^{T} \frac{1}{2\gamma} \frac{z_t^2}{a_t} + \frac{\gamma}{2}(\theta_1 - \theta^*)^2 a_1 + \sum_{t=2}^{T} \frac{\gamma}{2}(\theta_t - \theta^*)^2 (a_t - a_{t-1}) \qquad (\text{since } \gamma > 0, a_T > 0)
$$

$$
\leq \sum_{t=1}^{T} \frac{1}{2\gamma} \frac{z_t^2}{a_t} + \frac{\gamma}{2}(\theta_1 - \theta^*)^2 (a_1 - z_1^2) + \sum_{t=1}^{T} \frac{\gamma}{2}(\theta_t - \theta^*)^2 z_t^2 \qquad (\text{by } a_t - a_{t-1} = z_t^2). \tag{13}
$$

According to the definition: $R_t := z_t(\theta_t - \theta^*) - \frac{\gamma}{2}(\theta_t - \theta^*)^2 z_t^2$. We move the last term of the right hand side in (13) to the left hand side and get

$$
\sum_{t=1}^{T} R_t \leq \frac{1}{2\gamma} \sum_{t=1}^{T} \frac{z_t^2}{a_t} + \frac{\gamma}{2}(\theta_1 - \theta^*)^2 (a_1 - z_1^2).
$$

According to our algorithm, $a_1 - z_1^2 = a_0$. Since $\mathcal{K}_* = \bigcap_{t=1}^{T} \mathcal{K}_t \subseteq \mathcal{K}_1$, the diameter $(\theta_1 - \theta^*)^2 \leq D_1^2$. We recall the inequality (8), then

$$
\text{Regret}_T(\text{ONS}) \leq \sum_{t=1}^{T} R_t \leq \frac{1}{2\gamma} \sum_{t=1}^{T} \frac{z_t^2}{a_t} + \frac{\gamma}{2}(\theta_1 - \theta^*)^2 a_0 \leq \frac{1}{2\gamma} \sum_{t=1}^{T} \frac{z_t^2}{a_t} + \frac{\gamma}{2} D_1^2 a_0. \tag{14}
$$

If we let $a_0 = 1/\gamma^2 D_1^2$, it gives Lemma C.2,

$$
\text{Regret}_T(\text{ONS}) \leq \frac{1}{2\gamma} \sum_{t=1}^{T} \frac{z_t^2}{a_t} + \frac{\gamma}{2} D_1^2 a_0 = \frac{1}{2\gamma} \sum_{t=1}^{T} \frac{z_t^2}{a_t} + \frac{\gamma}{2} D_1^2 \cdot \frac{1}{\gamma^2 D_1^2} = \frac{1}{2\gamma} \left( \sum_{t=1}^{T} \frac{z_t^2}{a_t} + 1 \right).
$$

To get the upper bound of regret for our algorithm, we first show that the term $\sum_{t=1}^{T}(z_t^2/a_t)$ is upper bounded by a telescoping sum. For real numbers $a, b \in \mathbb{R}_+$, the first order Taylor expansion of the natural logarithm of $b$ at $a$ implies $\frac{a-b}{a} \leq \ln \frac{a}{b}$, thus

$$
\sum_{t=1}^{T} \frac{z_t^2}{a_t} = \sum_{t=1}^{T} \frac{1}{a_t} \cdot (a_t - a_{t-1}) \leq \sum_{t=1}^{T} \ln \left( \frac{a_t}{a_{t-1}} \right) = \ln \left( \frac{a_T}{a_0} \right).
$$

In our setting, $a_T = a_0 + \sum_{t=1}^{T} z_t^2$, where $a_0 = 1$, $z_t = \frac{g_t}{1 - g_t \theta_t}$. We recall the inequality (14), the upper bound of regret is

$$
\begin{aligned}
\text{Regret}_T(\text{ONS}) &\leq \frac{1}{2\gamma} \cdot \sum_{t=1}^{T} \frac{z_t^2}{a_t} + \frac{\gamma}{2}(\theta_1 - \theta^*)^2 \cdot 1 \\
&\leq \frac{1}{2\gamma} \cdot \ln\left(\frac{a_T}{a_0}\right) + \frac{\gamma}{2} D_1^2 \\
&= \frac{1}{2\gamma} \cdot \ln\left(1 + \sum_{t=1}^{T} \frac{g_t^2}{(1 - g_t \theta_t)^2}\right) + \frac{\gamma}{2} D_1^2 \\
&\leq \frac{1}{2\gamma} \cdot \ln\left(1 + 4 \sum_{t=1}^{T} g_t^2\right) + \frac{\gamma}{2} D_1^2 \\
&\leq \frac{1}{2\gamma} \cdot \ln\left(1 + 4 d_*^2 T\right) + \frac{\gamma}{2} D_1^2.
\end{aligned}
\tag{15}
$$

Since that $\gamma D_1^2$ is a positive constant, $g_t \in [-d_t, d_t]$, $\theta_t \in [-1/2d_t, 1/2d_t]$, it follows that $(1 - g_t \theta_t)^2 \in [\frac{1}{4}, \frac{9}{4}]$. We denote $d_* := \max_{t \geq 1} |d_t|$, which indicates that $g_t^2 \leq d_t^2 \leq d_*^2$. Consequently, we obtain that $\text{Regret}_T(\text{ONS}) = \mathcal{O}\left(\ln T^{\frac{1}{2\gamma}}\right)$. This conclusion will be used to show that the update of $\theta_t$ in the betting game is to play it against the exp-concave loss $\ell_t(\theta) = -\ln(1 - g_t \theta)$ and to get the lower bound of the wealth.

The reason we can obtain the upper bound of regret is that, although the values of $G_t$ and $D_t$ individually unknown, their product is deterministic. Consequently, the value of $\gamma_t = \frac{1}{2} \min\left\{\frac{1}{G_t D_t}, \alpha\right\}$ for all $t$ remains consistent. When we use Lemma C.2 to establish the regret bound, as illustrated by equation (13), the uniform $\gamma$ helps us simplify and combine terms to achieve the final result.

## D. Lower Bound on the Learner's Log-Wealth

**Lemma D.1.** *Assume an online learner receives a loss function $\ell_t(\theta) := -\ln(1 - g_t \theta)$ after committing a point $\theta_t \in \mathcal{K}_t$ in its decision space $\mathcal{K}_t$ at time $t$. Denote $d_* := \max_{t \geq 1} d_t$ with $d_t \geq |g_t|$. Then, if the online learner plays Online Newton Step (Algorithm 2), for any benchmark $u \in K_* := \bigcap_{t=1}^{T} \mathcal{K}_t$, the value of the wealth satisfies*

$$
\begin{aligned}
\ln W_T &= \ln W_T(u) - \text{Regret}_T(\text{ONS}) \\
&\geq \ln W_T(u) - \left(\frac{1}{2\gamma} \cdot \ln\left(1 + 4 d_*^2 T\right) + \frac{\gamma}{2}(\theta_1 - u)^2\right),
\end{aligned}
$$

*where we define the wealth at time $T$ as $W_T(\theta) := \prod_{t=1}^{T}(1 - g_t \theta)$, $\theta_1 \in \mathcal{K}_*$ is the initial point of ONS, and the inverse of the step size $\gamma$ satisfies $\gamma = \frac{1}{2} \min\{\frac{d_t}{G_t}, 1\}$, with $G_t := \max_{\theta \in \mathcal{K}_t} |\nabla \ell_t(\theta)|$ denoting the upper bound of the gradient $\nabla \ell_t(\theta)$.*

*Proof.* Since the update for the wealth is $W_t = W_{t-1} - g_t \theta_t W_{t-1}$, for $t = 1, \cdots, T$,

$$
W_1 = W_0(1 - g_1 \theta_1),
$$

$$
\vdots
$$

$$
W_T = W_{T-1}(1 - g_T \theta_T).
$$

We start with $W_0 = 1$, then by recursion,

$$
W_T = W_0 \cdot \prod_{t=1}^{T}(1 - g_t \theta_t) = \prod_{t=1}^{T}(1 - g_t \theta_t),
$$

thus we can express $\ln(W_T)$ as:

$$
\ln(W_T) = \sum_{t=1}^{T} \ln(1 - g_t \theta_t).
\tag{16}
$$

Similarly, the benchmark $u$ has:

$$\ln(W_T(u)) = \sum_{t=1}^{T} \ln(1 - g_t u). \tag{17}$$

We subtract equation (16) from (17) on both sides to obtain

$$\begin{aligned}
\ln(W_T(u)) - \ln(W_T) &= \sum_{t=1}^{T} \ln(1 - g_t u) - \sum_{t=1}^{T} \ln(1 - g_t \theta_t) \\
&= -\sum_{t=1}^{T} \ln(1 - g_t \theta_t) - \left(-\sum_{t=1}^{T} \ln(1 - g_t u)\right) \\
&= \sum_{t=1}^{T} -\ln(1 - g_t \theta_t) - \sum_{t=1}^{T} -\ln(1 - g_t u).
\end{aligned}$$

The above can be connected to the regret of the online learner, where the loss function is defined as $\ell_t(\theta) := -\ln(1 - g_t \theta)$. Consequently, we have

$$\ln(W_T) = \ln(W_T(u)) - \mathrm{Regret}_T(u). \tag{18}$$

Now applying the regret bound (15) leads to the result. $\qquad\square$

## E. Proof of Proposition 3.1

Some of the ideas in the proof is adapted from Dai et al. (2025), with the key difference that we consider a two-sided test, whereas their analysis focuses on a one-sided test with Bernoulli random variables. Additionally, we consider the range of $g_t$ within a time-varying interval $[-d_t, d_t]$ for each round $t$. We can divide the proof of Proposition 3.1 into three parts as below.

**1. Level-$\alpha$ Sequential Test.** In Algorithm 1, We treat the event $\{W_t \geq 1/\alpha\}$ as rejecting the null hypothesis $H_0$. A level-$\alpha$ sequential test is one that guarantees, under $H_0$, the probability of false rejection is at most $\alpha$, that is.

$$\sup_{P \in H_0} P(\exists t \geq 1 : W_t \geq 1/\alpha) \leq \alpha, \quad \text{or equivalently} \quad \sup_{P \in H_0} P(\tau < \infty) \leq \alpha.$$

When $P \in H_0$, i.e., $\mu_x = \mu_y$, it is true that

$$\mathbb{E}_P[\phi(x_t) - \phi(y_t)] = \mu_x - \mu_y = 0. \tag{19}$$

Wealth process is calculated as $W_t = (1 - g_t \theta_t) \times W_{t-1}$, and the initial wealth $W_0 = 1$, then:

$$W_t = (1 - g_t \theta_t) \times W_{t-1} = \prod_{i=1}^{t}(1 - g_i \theta_i) \times W_0 = \prod_{i=1}^{t}(1 - g_i \theta_i),$$

where $g_i = \phi(x_i) - \phi(y_i)$. Since $\theta_t$ is $\mathcal{F}_{t-1}$-measurable and according to (19), we have

$$\mathbb{E}_P[W_t | \mathcal{F}_{t-1}] = \mathbb{E}_P\left[(1 - g_t \theta_t) \times W_{t-1} \middle| \mathcal{F}_{t-1}\right] = \mathbb{E}_P\left[W_{t-1} \cdot (1 - \theta_t \cdot (\phi(x_t) - \phi(y_t))) \middle| \mathcal{F}_{t-1}\right] = W_{t-1},$$

where the last equality is because $\theta_t$ is fully determined by $\mathcal{F}_{t-1}$, and hence, given $\mathcal{F}_{t-1}$, $\theta_t$ is independent of $(\phi(x_t) - \phi(y_t))$, where the latter has conditional expectation zero under $H_0$.

Thus, $(W_t)_{t \geq 1}$ is a $P$-martingale with $W_0 = 1$. Given that $g_i \in [-d_i, d_i]$ and $\theta_i \in [-1/2d_i, 1/2d_i]$, we have $g_i \theta_i \in [-1/2, 1/2]$ for all $t$, then $W_t = \prod_{i=1}^{t}(1 - g_i \theta_i)$ remains non-negative for all $t$. Therefore, we can apply Ville's inequality (Ville, 1939) to establish that $P(\exists t \geq 1 : W_t \geq 1/\alpha) \leq \alpha$. This inequality shows that the sequential

test: "reject $H_0$ once the wealth $W_t$ reaches $1/\alpha$" controls the type-I error at level $\alpha$. If there exists a time budget $T$, we will verify the final step $W_T \geq Z/\alpha$ of the algorithm, i.e., we treat $\{W_t \geq 1/\alpha$ or $W_T > Z/\alpha\}$ as reject "$H_0$", which is validated by the randomized Ville's inequality of Ramdas & Manole (2023).

**2. Asymptotic Power One.** Test $\phi$ has asymptotic power $\beta = 1$ means that when $H_1$ (i.e., $\mu_x \neq \mu_y$) holds, our algorithm will ensure that wealth $W_t \geq 1/\alpha$ in finite time $t$ to reject $H_0$, that is:

$$\sup_{P \in H_1} P(\tau = \infty) \leq 1 - \beta = 0.$$

We first recall the relation between regret and wealth in the testing-by-betting game (c.f. (18)), i.e.,

$$\ln(W_t) = \ln(W_t(\theta_*)) - \text{Regret}_t(\theta_*), \tag{20}$$

where $W_t$ is the learner's wealth at round $t$, $\theta_*$ is any benchmark which lies in the decision space $\mathcal{K}_* = \left[-\frac{1}{2d_*}, \frac{1}{2d_*}\right]$.

Denote $\omega_* := \mathbb{E}[\ln(1 - g\theta_*)]$ the expected wealth of the benchmark in a single round. Taking the expectation on both sides of (20), we have

$$\mathbb{E}[\ln(W_t)] = \mathbb{E}[\ln(W_t(\theta_*))] - \mathbb{E}\left[\text{Regret}_t(\theta_*)\right]$$

$$= \mathbb{E}\left[\sum_{s=1}^{t} \ln(1 - g_s\theta_*)\right] - \mathbb{E}[\text{Regret}_t(\theta_*)]$$

$$= t\omega_* - \mathbb{E}\left[\text{Regret}_t(\theta_*)\right],$$

where the last equality holds by assuming that the random variables $(g_s)_{s \geq 1}$ are i.i.d.

We now analyze the probability that the null has not been rejected by time $t$, which is when the event $\{W_t < \frac{1}{\alpha}\}$ holds. We have

$$\mathbb{P}\left[W_t < \frac{1}{\alpha}\right] = \mathbb{P}\left[\ln(W_t) < \ln\left(\frac{1}{\alpha}\right)\right]$$

$$= \mathbb{P}\left[\ln(W_t) - \mathbb{E}[\ln(W_t(\theta_*))] < \ln\left(\frac{1}{\alpha}\right) - \mathbb{E}[\ln(W_t(\theta_*))]\right]$$

$$= \mathbb{P}\left[\ln(W_t(\theta_*)) - \text{Regret}_t(\theta_*) - \mathbb{E}[\ln(W_t(\theta_*))] < \ln\left(\frac{1}{\alpha}\right) - \mathbb{E}[\ln(W_t(\theta_*))]\right]$$

$$= \mathbb{P}\left[\ln(W_t(\theta_*)) - \mathbb{E}[\ln(W_t(\theta_*))] < \text{Regret}_t(\theta_*) + \ln\left(\frac{1}{\alpha}\right) - t\omega_*\right]. \tag{21}$$

Now we are going to show that $\text{Regret}_t(\theta_*) + \ln\left(\frac{1}{\alpha}\right) - t\omega_* \leq \frac{-t\omega_*}{2}$ when $t$ is sufficiently large. Then, from (21), we will have $\mathbb{P}\left[\ln(W_t(\theta_*)) - \mathbb{E}[\ln(W_t(\theta_*))] < \text{Regret}_t(\theta_*) + \ln\left(\frac{1}{\alpha}\right) - t\omega_*\right] \leq \mathbb{P}\left[\ln(W_t(\theta_*)) - \mathbb{E}[\ln(W_t(\theta_*))] < -\frac{t}{2}\omega_*\right]$.

From the regret bound of ONS (15), it suffices to show that

$$\frac{1}{2\gamma} \ln\left(1 + \sum_{s=1}^{t} \frac{g_s^2}{(1 - g_s\theta_s)^2}\right) + \frac{\gamma}{2}D_1^2 + \ln\left(\frac{1}{\alpha}\right) \leq \frac{t}{2}\omega_*. \tag{22}$$

We will require the time $t$ to satisfy

$$t \geq \frac{2\gamma D_1^2}{\omega_*}, \tag{23}$$

which leads to $\frac{\gamma}{2}D_1^2 \leq \frac{t\omega_*}{4}$. Therefore, to guarantee (22), it suffices to find $t$ such that

$$\frac{t\omega_*}{4} \geq \ln\left(\frac{1}{\alpha}\right) + \frac{1}{2\gamma} \ln\left(1 + \sum_{s=1}^{t} \frac{g_s^2}{(1 - g_s\theta_s)^2}\right).$$

We further note that $\frac{g_s^2}{(1-g_s\theta_s)^2} \le \frac{(d_*)^2}{(1-1/2)^2} = 4(d_*)^2$, which in turns implies a sufficient condition:

$$\frac{t\omega_*}{4} \ge \ln\left(\frac{1}{\alpha}\right) + \frac{1}{2\gamma}\ln\left(1 + 4t(d_*)^2\right).$$

The above condition together with (23) suggests that $\text{Regret}_t(\theta_*) + \ln\left(\frac{1}{\alpha}\right) - t\omega_* \le \frac{-t\omega_*}{2}$ holds when

$$t \gtrsim \frac{1}{\omega_*}\ln\left(\frac{(d_*)^{1/\gamma}}{\omega_*\alpha}\right). \tag{24}$$

Under (24), we hence have

$$\mathbb{P}\left[W_t < \frac{1}{\alpha}\right] \le \mathbb{P}\left[\ln(W_t(\theta_*)) - \mathbb{E}[\ln(W_t(\theta_*))] < -\frac{t}{2}\omega_*\right]. \tag{25}$$

Denote $\psi_t := \ln(1 - g_t\theta_*) - \mathbb{E}[\ln(1 - g_t\theta_*)] \in [\ln(1/2) - \omega_*, \ln(3/2) - \omega_*]$. We have that $\ln(W_t(\theta_*)) - \mathbb{E}[\ln(W_t(\theta_*))] = \sum_{s=1}^{t}\psi_s$, which is a sum of zero-mean i.i.d bounded random variables. Specifically, $\psi_t$ is a sub-Gaussian with parameter $\sigma := \frac{1}{2}\ln(3)$. By Hoeffding's inequality, we have

$$\mathbb{P}\left[\frac{1}{t}\sum_{s=1}^{t}\psi_s \le -c\right] \le \exp\left(-\frac{tc^2}{2\sigma^2}\right), \tag{26}$$

for any constant $c > 0$. Then,

$$\mathbb{P}\left[\ln(W_t(\theta_*)) - \mathbb{E}[\ln(W_t(\theta_*))] < -\frac{t}{2}\omega_*\right] = \mathbb{P}\left[\frac{1}{t}\sum_{s=1}^{t}\psi_s < -\frac{1}{2}\omega_*\right] \le \exp\left(-\frac{(\omega_*)^2}{2(\ln 3)^2}t\right),$$

where we let $c \leftarrow \frac{\omega_*}{2}$ in (26). Hence, we obtain

$$\mathbb{P}\left[W_t < \frac{1}{\alpha}\right] \le \exp\left(-\frac{(\omega_*)^2}{2(\ln 3)^2}t\right). \tag{27}$$

Let $H_t$ be the event that we stop at time $t$. Then,

$$\begin{aligned}
\mathbb{P}[t = \infty] &= \mathbb{P}\left[\lim_{t\to\infty} \cap_{s\le t}\neg H_s\right] \\
&= \lim_{t\to\infty} \mathbb{P}\left[\cap_{s\le t}\neg H_s\right] \\
&\le \lim_{t\to\infty} \mathbb{P}\left[\neg H_t\right] \\
&= \lim_{t\to\infty} \mathbb{P}\left[W_t < \frac{1}{\alpha}\right] \\
&\le \lim_{t\to\infty} \exp\left(-\frac{(\omega_*)^2}{2(\ln 3)^2}t\right) \\
&= 0,
\end{aligned}$$

where the last equality is by (27). This shows that the test power is one.

**3. Expected Stopping Time.**

Now we upper-bound the expected stopping time as follows. Denote $t_*$ the time when (25) starts to hold at any $t \ge t_*$. We

have

$$\mathbb{E}[\tau] = \sum_{t=1}^{\infty} \mathbb{P}[\tau > t]$$

$$= \sum_{t=1}^{\infty} \mathbb{P}[\cap_{s \leq t} \neg H_s]$$

$$\leq \sum_{t=1}^{\infty} \mathbb{P}[\neg H_t]$$

$$= \sum_{t=1}^{\infty} \mathbb{P}\left[W_t < \frac{1}{\alpha}\right]$$

$$\leq t_* + \sum_{t=1}^{\infty} \exp\left(-\frac{(\omega_*)^2}{2(\ln 3)^2} t\right)$$

$$\leq t_* + \frac{1}{\exp\left(\frac{(\omega_*)^2}{2(\ln 3)^2} t\right) - 1}$$

$$\leq t_* + \frac{2(\ln 3)^2}{\omega_*^2}, \tag{28}$$

where the last inequality uses $\exp(z) \geq 1 + z$ for any $z \geq 0$.

To proceed, we recall the notation $\omega_* := \mathbb{E}[\ln(1 - g\theta_*)]$. We now let the benchmark $\theta_*$ to be the one that maximizes the expected wealth in one round, i.e., $\mathbb{E}[\ln(1 - g\theta)]$, in the decision space $\mathcal{K}_*$.

By the inequality $\ln(1 + c) \geq c - c^2$ for any $|c| \leq \frac{1}{2}$, we have

$$\omega_* = \max_{\theta \in K_*} \mathbb{E}[\ln(1 - g\theta)] \geq \max_{\theta \in K_*} \mathbb{E}[-\theta g - \theta^2 g^2]$$

$$= \max_{\theta \in K_*} -\theta(\mu_x - \mu_y) - \theta^2 \left(\mathrm{Var}[g] + (\mu_x - \mu_y)^2\right),$$

$$\approx \frac{\Delta^2}{\mathrm{Var}[g] + \Delta^2}$$

$$\approx \frac{\Delta^2}{d_*^2 + \Delta^2},$$

where we note $\mathbb{E}[g] = \mathbb{E}[\phi(x) - \phi(y)] = \mu_x - \mu_y$, $\mathrm{Var}[g] = \mathbb{E}[g^2] - (\mathbb{E}[g])^2$, and $\mathrm{Var}[g] \leq d_*^2$.

Combining (24) and (28), together with the expression of $\omega_*$ above, we have

$$\mathbb{E}[\tau] = \mathcal{O}\left(\frac{d_*^2}{\Delta^2} \ln\left(\frac{d_*}{\alpha\Delta}\right) + \frac{d_*^4}{\Delta^4}\right). \tag{29}$$

# F. Proof of Proposition 3.2

Recall we formulated two one-sided hypotheses:

$$H_0^A : \mu_x - \mu_y - \epsilon \leq 0 \text{ vs. } H_1^A : \mu_x - \mu_y - \epsilon > 0,$$

and

$$H_0^B : \mu_y - \mu_x - \epsilon \leq 0 \text{ vs. } H_1^B : \mu_y - \mu_x - \epsilon > 0.$$

**1. Level-$\alpha$ Sequential Test.** In contrast to the betting interval employed in Chugg et al. (2023) for composite hypotheses test, which includes both positive and negative values, we specify a nonpositive decision space $\theta_t \in [-1/2d_t, 0]$ in order to

---

**Algorithm 3** Online Detection of LLMs via Online Optimization and Betting for the Composite Hypotheses Testing

---

**Require:** a score function $\phi(\cdot) : \text{Text} \to \mathbb{R}$.

1: **Init:** $\theta_1^A, \theta_1^B \leftarrow 0$, $a_0^A, a_0^B \leftarrow 1$, wealth $W_0^A, W_0^B \leftarrow 1$, step size $\gamma$, difference parameter $\epsilon$, and significance level parameter $\alpha \in (0, 1)$.

2: **for** $t = 1, 2, \ldots, T$ **do**

3:      *# $T$ is the time budget, which can be $\infty$ if their is no time constraint.*

4:      Observe a text $y_t$ from an unknown source and compute $\phi(y_t)$.

5:      Sample $x_t$ from a dataset of human-written texts and compute $\phi(x_t)$.

6:      Set $g_t^A = \phi(x_t) - \phi(y_t) - \epsilon, g_t^B = \phi(y_t) - \phi(x_t) - \epsilon$.

7:      Update wealth $W_t^A = W_{t-1}^A \cdot (1 - g_t^A \theta_t^A), W_t^B = W_{t-1}^B \cdot (1 - g_t^B \theta_t^B)$.

8:      **if** $W_t^A \geq 2/\alpha$ or $W_t^B \geq 2/\alpha$ **then**

9:          Declare that the source producing the sequence of texts $y_t$ is an LLM.

10:      **end if**

11:      Get a hint $d_{t+1}$ and specify the decision space $\mathcal{K}_{t+1} := [-\frac{1}{2d_{t+1}}, 0]$.

12:      *// Update $\theta_{t+1}^A, \theta_{t+1}^B \in \mathcal{K}_{t+1}$ via ONS on the loss function $\ell_t^A(\theta) := -\ln(1 - g_t^A \theta)$, and $\ell_t^B(\theta) := -\ln(1 - g_t^B \theta)$.*

13:      Compute $z_t^A = \frac{d\ell_t(\theta_t^A)}{d\theta} = \frac{g_t^A}{1 - g_t^A \theta_t^A}, z_t^B = \frac{d\ell_t(\theta_t^B)}{d\theta} = \frac{g_t^B}{1 - g_t^B \theta_t^B}$.

14:      Compute $a_t^A = a_{t-1}^A + (z_t^A)^2, a_t^B = a_{t-1}^B + (z_t^B)^2$.

15:      Compute $\theta_{t+1}^A = \max\left(\min\left(\theta_t^A - \frac{1}{\gamma} \frac{z_t^A}{a_t^A}, 0\right), -\frac{1}{2d_{t+1}}\right), \theta_{t+1}^B = \max\left(\min\left(\theta_t^B - \frac{1}{\gamma} \frac{z_t^B}{a_t^B}, 0\right), -\frac{1}{2d_{t+1}}\right)$.

16: **end for**

17: **if** the source has not been declared as an LLM **then**

18:      Sample $Z \sim \text{Unif}(0, 1)$, declare the sequence of texts $y_t$ is from an LLM if $W_T^A \geq 2Z/\alpha$ or $W_T^B \geq 2Z/\alpha$.

19: **end if**

---

ensure that the wealth process is a supermartingale under $H_0$, while still enabling fast wealth growth under the alternative $H_1$. The resulting wealth process is now given by:

$$W_t^A = W_0 \cdot \prod_{s=1}^{t} (1 - \theta_s(g_s - \epsilon)) = \prod_{s=1}^{t} (1 - \theta_s(g_s - \epsilon))$$

and

$$W_t^B = W_0 \cdot \prod_{s=1}^{t} (1 - \theta_s(-g_s - \epsilon)) = \prod_{s=1}^{t} (1 - \theta_s(-g_s - \epsilon)),$$

where $g_s = \phi(x_s) - \phi(y_s)$. Recall that $\epsilon$ is a nonnegative constant satisfying $\epsilon \leq d_t$ for all $t$. Consequently, the range of $\theta_t(g_t - \epsilon)$ and $\theta_t(-g_t - \epsilon)$, given by $[-(d_t - \epsilon)/2d_t, (d_t + \epsilon)/2d_t]$, always lies within the interval $[-1, 1]$. This guarantees that the wealth process remains nonnegative. Moreover, the supermartingale property of the wealth processes $W_t^A$ and $W_t^B$ can still be preserved. Take $W_t^A$ as an example, under the corresponding null hypothesis $H_0^A : \mu_x - \mu_y \leq \epsilon$, i.e., $\mathbb{E}_P[\phi(x_t) - \phi(y_t)] \leq \epsilon$ for $P \in H_0^A$. Then,

$$\mathbb{E}_P[W_t^A | \mathcal{F}_{t-1}] = \mathbb{E}_P\left[W_{t-1}^A \left(1 - \underbrace{\theta_t}_{\leq 0} \cdot (\phi(x_t) - \phi(y_t) - \epsilon)\right) \bigg| \mathcal{F}_{t-1}\right] \leq W_{t-1}^A,$$

where the inequality is because $\theta_t$ is fully determined by $\mathcal{F}_{t-1}$, and hence, given $\mathcal{F}_{t-1}$, $\theta_t$ is independent of $(\phi(x_t) - \phi(y_t) - \epsilon)$, where the latter has conditional expectation less than or equal to zero under $H_0^A : \mu_x - \mu_y \leq \epsilon$.

As for the other null hypothesis $H_0^B : \mu_y - \mu_x \leq \epsilon$, i.e., $\mathbb{E}_P[\phi(y_t) - \phi(x_t)] \leq \epsilon$, it can be shown that the wealth process $W_t^B$ also forms a nonnegative supermartingale.

Applying the randomized Ville's inequality gives us

$$P(\exists t \geq 1 : W_t^A \geq 2/\alpha) \leq \alpha/2, \tag{30}$$

and

$$P(\exists t \geq 1 : W_t^B \geq 2/\alpha) \leq \alpha/2. \tag{31}$$

Thus we can get the union bound of (30) and (31)

$$P(\exists t \geq 1 : (W_t^A \geq 2/\alpha) \cup (W_t^B \geq 2/\alpha)) \leq \alpha,$$

which indicates that rejecting the null hypothesis when either $W_t^A \geq 2/\alpha$ or $W_t^B \geq 2/\alpha$ is a level-$\alpha$ sequential test. The detection process for the composite hypothesis is shown as Algorithm 3.

**2. Asymptotic Power One.** Now let us switch to analyze the test power of the proposed algorithm. We first recall the relation between regret and wealth in the testing-by-betting game (c.f. (18)), i.e.,

$$\ln(W_t^A) = \ln(W_t^A(\theta_*)) - \text{Regret}_t(\theta_*), \tag{32}$$

where $W_t^A$ is the learner's wealth at round $t$, $\theta_*$ is any benchmark which lies in the decision space $\mathcal{K}'_* = [-\frac{1}{2d'_*}, 0]$ with $d'_* = d_* + \epsilon$.

Denote $\omega_*^A := \mathbb{E}[\ln(1 - (g - \epsilon)\theta_*)]$ the expected wealth of the benchmark in a single round. For the notation simplicity, in the following, we will drop the superscript $A$ in $\omega_*^A$ when it is clear to the context. Taking the expectation on both sides of (32), we have

$$\begin{aligned}
\mathbb{E}[\ln(W_t^A)] &= \mathbb{E}[\ln(W_t^A(\theta_*))] - \mathbb{E}\left[\text{Regret}_t(\theta_*)\right] \\
&= \mathbb{E}\left[\sum_{s=1}^{t} \ln(1 - (g_s - \epsilon)\theta_*)\right] - \mathbb{E}[\text{Regret}_t(\theta_*)] \\
&= t\omega_* - \mathbb{E}\left[\text{Regret}_t(\theta_*)\right],
\end{aligned}$$

where the last equality holds by assuming that the random variables $(g_s)_{s \geq 1}$ are i.i.d.

We now analyze the probability that the null has not been rejected by time $t$, which is when both the events $\{W_t^A < \frac{2}{\alpha}\}$ and $\{W_t^B < \frac{2}{\alpha}\}$ hold. We have

$$\begin{aligned}
\mathbb{P}\left[W_t^A < \frac{2}{\alpha}\right] &= \mathbb{P}\left[\ln(W_t^A) < \ln\left(\frac{2}{\alpha}\right)\right] \\
&= \mathbb{P}\left[\ln(W_t^A) - \mathbb{E}[\ln(W_t^A(\theta_*))] < \ln\left(\frac{2}{\alpha}\right) - \mathbb{E}[\ln(W_t^A(\theta_*))]\right] \\
&= \mathbb{P}\left[\ln(W_t^A(\theta_*)) - \text{Regret}_t(\theta_*) - \mathbb{E}[\ln(W_t^A(\theta_*))] < \ln\left(\frac{2}{\alpha}\right) - \mathbb{E}[\ln(W_t^A(\theta_*))]\right] \\
&= \mathbb{P}\left[\ln(W_t^A(\theta_*)) - \mathbb{E}[\ln(W_t^A(\theta_*))] < \text{Regret}_t(\theta_*) + \ln\left(\frac{2}{\alpha}\right) - t\omega_*\right]. \tag{33}
\end{aligned}$$

Now we are going to show that $\text{Regret}_t(\theta_*) + \ln\left(\frac{2}{\alpha}\right) - t\omega_* \leq \frac{-t\omega_*}{2}$ when $t$ is sufficiently large. Then, from (33), we will have $\mathbb{P}\left[\ln(W_t^A(\theta_*)) - \mathbb{E}[\ln(W_t^A(\theta_*))] < \text{Regret}_t(\theta_*) + \ln\left(\frac{2}{\alpha}\right) - t\omega_*\right] \leq \mathbb{P}\left[\ln(W_t^A(\theta_*)) - \mathbb{E}[\ln(W_t^A(\theta_*))] < -\frac{t}{2}\omega_*\right]$.

From the regret bound of ONS (15), it suffices to show that

$$\frac{1}{2\gamma} \ln\left(1 + \sum_{s=1}^{t} \frac{(g_s - \epsilon)^2}{(1 - (g_s - \epsilon)\theta_s)^2}\right) + \frac{\gamma}{2} D_1^2 + \ln\left(\frac{2}{\alpha}\right) \leq \frac{t}{2}\omega_*. \tag{34}$$

We will require the time $t$ to satisfy

$$t \geq \frac{2\gamma D_1^2}{\omega_*}, \tag{35}$$

which leads to $\frac{\gamma}{2} D_1^2 \leq \frac{t\omega_*}{4}$. Therefore, to guarantee (34), it suffices to find $t$ such that

$$\frac{t\omega_*}{4} \geq \ln\left(\frac{2}{\alpha}\right) + \frac{1}{2\gamma} \ln\left(1 + \sum_{s=1}^{t} \frac{(g_s - \epsilon)^2}{(1 - (g_s - \epsilon)\theta_s)^2}\right).$$

We further note that $\frac{(g_s - \epsilon)^2}{(1 - (g_s - \epsilon)\theta_s)^2} \leq \frac{(d'_*)^2}{(1 - (d_s + \epsilon)/2d_s)^2} \leq c(d'_*)^2$, for some constant $c \geq \left( \frac{2d_s}{d_s - \epsilon} \right)^2$, which in turns implies a sufficient condition:

$$\frac{t\omega_*}{4} \geq \ln \left( \frac{2}{\alpha} \right) + \frac{1}{2\gamma} \ln \left( 1 + ct(d'_*)^2 \right).$$

The above condition together with (35) suggests that $\text{Regret}_t(\theta_*) + \ln \left( \frac{2}{\alpha} \right) - t\omega_* \leq \frac{-t\omega_*}{2}$ holds when

$$t \gtrsim \frac{1}{\omega_*} \ln \left( \frac{(d'_*)^{1/\gamma}}{\omega_* \alpha} \right). \tag{36}$$

Under (36), we hence have

$$\mathbb{P} \left[ W_t^A < \frac{2}{\alpha} \right] \leq \mathbb{P} \left[ \ln(W_t^A(\theta_*)) - \mathbb{E}[\ln(W_t^A(\theta_*))] < -\frac{t}{2}\omega_* \right]. \tag{37}$$

Denote $\psi_t := \ln(1 - (g_t - \epsilon)\theta_*) - \mathbb{E}[\ln(1 - (g_t - \epsilon)\theta_*)] \in [\ln(1/2) - \omega_*, \ln(3/2) - \omega_*]$. We have that $\ln(W_t^A(\theta_*)) - \mathbb{E}[\ln(W_t^A(\theta_*))] = \sum_{s=1}^t \psi_s$, which is a sum of zero-mean i.i.d bounded random variables. Specifically, $\psi_t$ is a sub-Gaussian with parameter $\sigma := \frac{1}{2} \ln(3)$. By Hoeffding's inequality, we have

$$\mathbb{P} \left[ \frac{1}{t} \sum_{s=1}^t \psi_s \leq -c \right] \leq \exp \left( -\frac{tc^2}{2\sigma^2} \right), \tag{38}$$

for any constant $c > 0$. Then,

$$\mathbb{P} \left[ \ln(W_t^A(\theta_*)) - \mathbb{E}[\ln(W_t^A(\theta_*))] < -\frac{t}{2}\omega_* \right] = \mathbb{P} \left[ \frac{1}{t} \sum_{s=1}^t \psi_s < -\frac{1}{2}\omega_* \right] \leq \exp \left( -\frac{(\omega_*)^2}{2(\ln 3)^2}t \right),$$

where we let $c \leftarrow \frac{\omega_*}{2}$ in (38). Combining the above inequality and (37), we obtain

$$\mathbb{P} \left[ W_t^A < \frac{2}{\alpha} \right] \leq \exp \left( -\frac{(\omega_*^A)^2}{2(\ln 3)^2}t \right). \tag{39}$$

Similarly, we have

$$\mathbb{P} \left[ W_t^B < \frac{2}{\alpha} \right] \leq \exp \left( -\frac{(\omega_*^B)^2}{2(\ln 3)^2}t \right), \tag{40}$$

where $\omega_*^B := \mathbb{E} \left[ \ln \left( 1 - (-g - \epsilon)\theta_*^B \right) \right]$ for a benchmark $\theta_*^B$.

Let $H_t$ be the event that we stop at time $t$, i.e., $H_t := \{W_t^A \geq \frac{2}{\alpha} \text{ or } W_t^B \geq \frac{2}{\alpha}\}$. Then, we deduce that

$$\begin{aligned}
\mathbb{P}[t = \infty] &= \mathbb{P} \left[ \lim_{t \to \infty} \cap_{s \leq t} \neg H_s \right] \\
&= \lim_{t \to \infty} \mathbb{P} \left[ \cap_{s \leq t} \neg H_s \right] \\
&\leq \lim_{t \to \infty} \mathbb{P} \left[ \neg H_t \right] \\
&= \lim_{t \to \infty} \mathbb{P} \left[ W_t^A < \frac{2}{\alpha} \text{ and } W_t^B < \frac{2}{\alpha} \right] \\
&\leq \lim_{t \to \infty} \mathbb{P} \left[ W_t^A < \frac{2}{\alpha} \right] + \mathbb{P} \left[ W_t^B < \frac{2}{\alpha} \right] \\
&\leq \lim_{t \to \infty} \exp \left( -\frac{(\omega_*^A)^2}{2(\ln 3)^2}t \right) + \exp \left( -\frac{(\omega_*^B)^2}{2(\ln 3)^2}t \right) \\
&= 0,
\end{aligned}$$

where the last equality is by (39) and (40). This shows that the test power is one.

**3. Expected Rejection Time $\tau$.** Now we upper-bound the expected stopping time as follows. Denote $t_*$ the time when (39) and (40) starts to hold at any $t \geq t_*$. We have

$$
\begin{aligned}
\mathbb{E}[\tau] &= \sum_{t=1}^{\infty} \mathbb{P}\left[\tau > t\right] \\
&= \sum_{t=1}^{\infty} \mathbb{P}\left[\cap_{s \leq t} \neg H_s\right] \\
&\leq \sum_{t=1}^{\infty} \mathbb{P}\left[\neg H_t\right] \\
&\leq \sum_{t=1}^{\infty} \left( \mathbb{P}\left[W_t^A < \frac{2}{\alpha}\right] + \mathbb{P}\left[W_t^B < \frac{2}{\alpha}\right] \right) \\
&\leq t_* + \sum_{t=1}^{\infty} \exp\left(-\frac{(\omega_*^A)^2}{2(\ln 3)^2}t\right) + \exp\left(-\frac{(\omega_*^B)^2}{2(\ln 3)^2}t\right) \\
&\leq t_* + \frac{1}{\exp\left(\frac{(\omega_*^A)^2}{2(\ln 3)^2}t\right) - 1} + \frac{1}{\exp\left(\frac{(\omega_*^B)^2}{2(\ln 3)^2}t\right) - 1} \\
&\leq t_* + \frac{2(\ln 3)^2}{(\omega_*^A)^2} + \frac{2(\ln 3)^2}{(\omega_*^B)^2},
\end{aligned}
\tag{41}
$$

where the last inequality uses $\exp(z) \geq 1 + z$ for any $z \geq 0$.

To proceed, we recall the notation $\omega_*^A := \mathbb{E}\left[\ln(1 - (g - \epsilon)\theta_*)\right]$. We now let the benchmark $\theta_*$ to be the one that maximizes the expected wealth in one round, i.e., $\mathbb{E}\left[\ln(1 - (g - \epsilon)\theta)\right]$, in the decision space $\mathcal{K}_*'$.

By the inequality $\ln(1 + c) \geq c - c^2$ for any $|c| \leq \frac{1}{2}$, we have

$$
\begin{aligned}
\omega_*^A &= \max_{\theta \in K_*'} \mathbb{E}\left[\ln\left(1 - (g - \epsilon)\theta\right)\right] \geq \max_{\theta \in K_*'} \mathbb{E}\left[-\theta(g - \epsilon) - \theta^2(g - \epsilon)^2\right] \\
&= \max_{\theta \in K_*'} -\theta(\mu_x - \mu_y - \epsilon) - \theta^2 \left(\text{Var}[g - \epsilon] + (\mu_x - \mu_y - \epsilon)^2\right), \\
&\approx \frac{(\Delta - \epsilon)^2}{\text{Var}[g - \epsilon] + (\Delta - \epsilon)^2} \\
&\approx \frac{(\Delta - \epsilon)^2}{(d_*')^2 + (\Delta - \epsilon)^2},
\end{aligned}
$$

where we note $\mathbb{E}[g] = \mathbb{E}[\phi(x) - \phi(y)] = \mu_x - \mu_y$, $\text{Var}[g - \epsilon] = \mathbb{E}[(g - \epsilon)^2] - (\mathbb{E}[(g - \epsilon)])^2$, and $\text{Var}[g - \epsilon] \leq (d_*')^2$. Similarly, we have $\omega_*^B \approx \frac{(\Delta - \epsilon)^2}{(d_*')^2 + (\Delta - \epsilon)^2}$.

Combining (36) and (41), together with the expression of $\omega_*^A$ and $\omega_*^B$ above, we have

$$
\mathbb{E}[\tau] = \mathcal{O}\left( \frac{(d_* + \epsilon)^2}{(\Delta - \epsilon)^2} \ln\left(\frac{d_* + \epsilon}{\alpha(\Delta - \epsilon)}\right) + \frac{(d_* + \epsilon)^4}{(\Delta - \epsilon)^4} \right).
\tag{42}
$$

## G. Experimental Results of Detecting 2024 Olympic News or Machine-generated News

Our generation process for fake news is guided by Mitchell et al. (2023); Bao et al. (2023); Su et al. (2023). Specifically, we use the T5 tokenizer to process each human-written news article to retrieve the first 30 tokens as {prefix}. Then, we initiate the generation process by sending the following messages to the model service, such as: "You are a News writer. Please write an article with about 150 words starting exactly with {prefix}."

**Experimental Results for Two Scenarios.** Figure 5 and Figure 6 show the results of detecting real Olympic news and news generated by Gemini-1.5-Flash, Gemini-1.5-Pro, and PaLM 2, when designating human-written text from the XSum dataset as $x_t$ in Scenario 1 and Scenario 2, respectively. In Scenario 1, our algorithm consistently controls the FPRs below

the significance level $\alpha$ for all source models, scoring models, and score functions. This is because we set the true value of $\Delta$ between two sequences of human texts to $\epsilon$, which satisfies the null hypothesis condition, i.e., $|\mu_x - \mu_y| \leq \epsilon$. This ensures that the wealth remains a supermartingale. Texts generated by PaLM 2 are detected almost immediately by most score functions within 100 time steps, as illustrated in Figure 5e and Figure 5f. Conversely, fake Olympic news generated by Gemini-1.5-Pro often fails to be identified as LLM-generated within 500 time steps under certain score functions, as shown in Figure 5c and Figure 5d. Vertical lines in Figure 5a-5d are displayed because the the $\Delta$ values for human texts and fake news, as shown in Table 1, are smaller than the corresponding $\epsilon$ values. This means that the score discrepancies between fake news and XSum texts do not exceed the threshold necessary for rejecting $H_0$. Although under $H_1$, the $\Delta$ for Entropy when using Gemma-2B to score texts generated by Gemini-1.5-Pro is 0.2745, larger than the value of $\epsilon$ (0.2690), the discrepancy is too small to lead to a rejection of $H_0$ within 500 time steps.

According to Figure 6, Scenario 2 exhibits a similar trend to that observed in Scenario 1, where texts generated by PaLM 2 are quickly declared as originating from an LLM, while texts produced by Gemini-1.5-Pro are identified more slowly. In Scenario 2, Fast-DetectGPT consistently outperforms all other score functions when using Neo-2.7 as scoring model, as evidenced by the results in Figure 6a, 6c and 6e. Only the score functions of supervised classifiers have FPRs slightly above the significance level $\alpha$. Although their average estimated values of $\epsilon$ in Table 5 are larger than the actual $\Delta$ value under $H_0$ in Table 1, high FPRs often occur because most $\epsilon$ values estimated in 1000 repeated tests are smaller than the actual $\Delta$ values. When using Gemma-2B as the scoring model in Scenario 2, four score functions: Fast-DetectGPT, LRR, Likelihood, and DNA-GPT consistently maintain FPRs within the expected range $\alpha$. Likelihood is the fastest to reject $H_0$. However, estimated $\epsilon$ values of DetectGPT, NPR, and Entropy are smaller than real $\Delta$ values under $H_0$. Then, even when $H_0$ is true, the discrepancy between human texts exceed the estimated threshold $\epsilon$ for rejecting $H_0$, which result in high FPRs, as shown in Figure 6b, 6d and 6f.

*Table 1.* Values of $\Delta$, which are calculated according to $\Delta = \left| \sum_{i=1}^{500} \phi(x_i)/500 - \sum_{j=1}^{500} \phi(y_j)/500 \right|$, where $\phi(x_i)$ is score of the $i$-th text $x_i$ from XSum, $\phi(y_i)$ is score of the $j$-th text $y_j$ from the source to be detected. Every two columns starting from the third column represent the $\Delta$ values under $H_1$ and $H_0$ for each test scenario. For instance, in calculating $\Delta$ for the third column, $y_j$ represents the $j$-th fake news generated by Gemini-1.5-Flash based on pre-tokens of Olympic 2024 news. For the fourth column, $y_j$ refers to the $j$-th 2024 Olympic news articles. Values in Column "Human, Human" are also used to set $\epsilon$ values for tests in Scenario 1.

| Scoring Model | Score Function | XSum, Olympic | | XSum, Olympic | | XSum, Olympic | |
| --- | --- | --- | --- | --- | --- | --- | --- |
| | | Human, 1.5-Flash | Human, Human | Human, 1.5-Pro | Human, Human | Human, PaLM 2 | Human, Human |
| Neo-2.7 | Fast-DetectGPT | 2.4786 | 0.3634 | 1.2992 | 0.3660 | 3.6338 | 0.4232 |
| | DetectGPT | 0.3917 | 0.0202 | 0.3101 | 0.0274 | 0.6050 | 0.0052 |
| | NPR | 0.0232 | 0.0014 | 0.0155 | 0.0015 | 0.0398 | 0.0005 |
| | LRR | 0.1042 | 0.0324 | 0.0289 | 0.0328 | 0.2606 | 0.0370 |
| | Logrank | 0.2590 | 0.0543 | 0.1312 | 0.0561 | 0.4995 | 0.0743 |
| | Likelihood | 0.3882 | 0.0618 | 0.2170 | 0.0652 | 0.7641 | 0.0948 |
| | Entropy | 0.0481 | 0.0766 | 0.0067 | 0.0728 | 0.1878 | 0.0483 |
| | DNA-GPT | 0.1937 | 0.0968 | 0.0957 | 0.1032 | 0.4086 | 0.1083 |
| | RoBERTa-base | 0.2265 | 0.0461 | 0.0287 | 0.0491 | 0.6343 | 0.0370 |
| | RoBERTa-large | 0.0885 | 0.0240 | 0.0249 | 0.0250 | 0.4197 | 0.0281 |
| Gemma-2B | Fast-DetectGPT | 2.1412 | 0.5889 | 0.9321 | 0.5977 | 3.7314 | 0.6758 |
| | DetectGPT | 0.7146 | 0.3538 | 0.6193 | 0.3530 | 0.8403 | 0.3360 |
| | NPR | 0.0632 | 0.0254 | 0.0477 | 0.0249 | 0.1005 | 0.0232 |
| | LRR | 0.1604 | 0.0129 | 0.0825 | 0.0112 | 0.3810 | 0.0038 |
| | Logrank | 0.3702 | 0.0973 | 0.2527 | 0.0932 | 0.5917 | 0.0687 |
| | Likelihood | 0.6093 | 0.1832 | 0.4276 | 0.1761 | 0.9705 | 0.1358 |
| | Entropy | 0.2668 | 0.2743 | 0.2745 | 0.2690 | 0.4543 | 0.2347 |
| | DNA-GPT | 0.2279 | 0.0353 | 0.1144 | 0.0491 | 0.4072 | 0.0681 |
| | RoBERTa-base | 0.2265 | 0.0461 | 0.0287 | 0.0491 | 0.6343 | 0.0370 |
| | RoBERTa-large | 0.0885 | 0.0240 | 0.0249 | 0.0250 | 0.4197 | 0.0281 |

To summarize, the FPRs can be controlled below the significance level $\alpha$ if the preset $\epsilon$ is greater than or equal to the actual absolute difference in mean scores between two sequences of human texts. However, if $\epsilon$ is greater than or is nearly equal to the $\Delta$ value for human text $x_t$ and machine-generated text $y_t$, it would be challenging for our algorithm to declare the

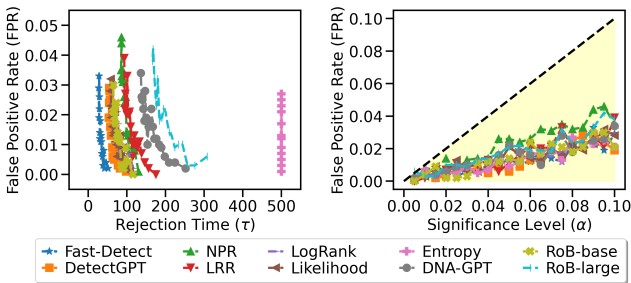

(a) Results for detecting the source of $y_t$ (2024 Olympic news vs. news generated by Gemini-1.5-Flash), with human-written text $x_t$ sampled from XSum. The scoring model used is Neo-2.7.

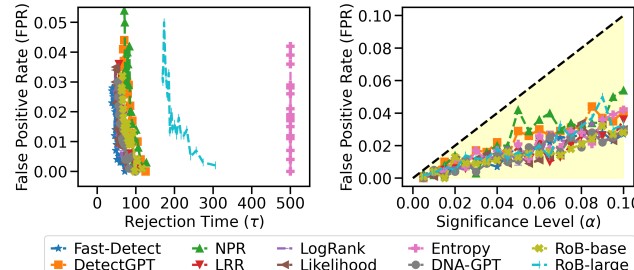

(b) Results for detecting the source $y_t$ (2024 Olympic news vs. news generated by Gemini-1.5-Flash), with human-written text $x_t$ sampled from XSum. The scoring model used is Gemma-2B.

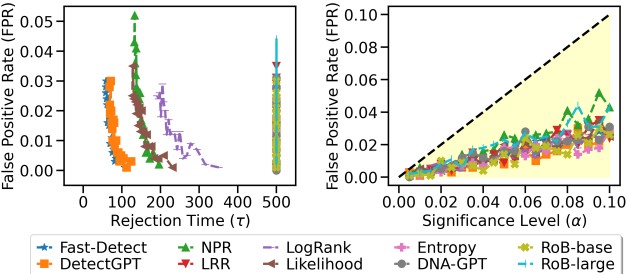

(c) Results for detecting the source of $y_t$ (2024 Olympic news vs. news generated by Gemini-1.5-Pro), with human-written text $x_t$ sampled from XSum. The scoring model used is Neo-2.7.

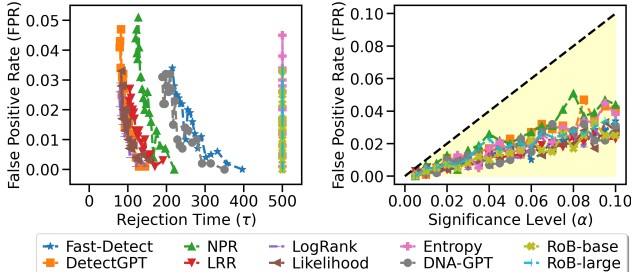

(d) Results for detecting the source of $y_t$ (2024 Olympic news vs. news generated by Gemini-1.5-Pro), with human-written text $x_t$ sampled from XSum. The scoring model used is Gemma-2B.

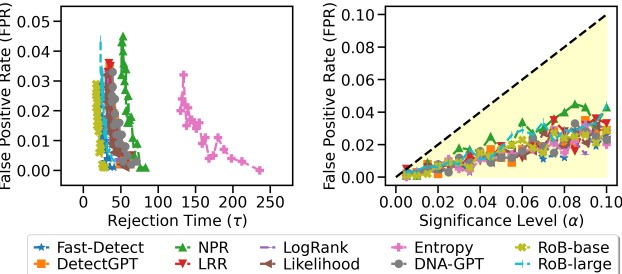

(e) Results for detecting the source of $y_t$ (2024 Olympic news vs. news generated by PaLM 2), with human-written text $x_t$ sampled from XSum. The scoring model used is Neo-2.7.

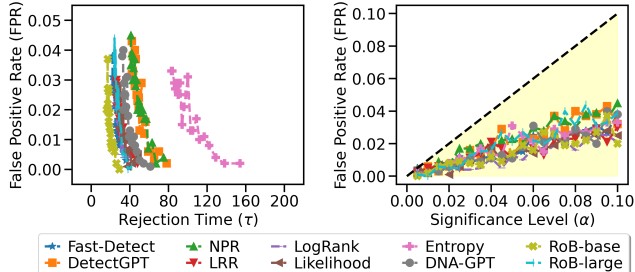

(f) Results for detecting the source of $y_t$ (2024 Olympic news vs. news generated by PaLM 2), with human-written text $x_t$ sampled from XSum. The scoring model used is Gemma-2B.

*Figure 5.* Results for detecting 2024 Olympic news and machine-generated news with Algorithm 3 for Scenario 1. We use 3 source models: Gemini-1.5-Flash, Gemini-1.5-Pro and PaLM 2 to generate fake news and 2 scoring models: Neo-2.7, Gemma-2B. The left column displays results using the Neo-2.7 scoring model, while the right column presents results using the Gemma-2B scoring model. Score functions of supervised classifiers (RoB-base and RoB-large) are independent of scoring models.

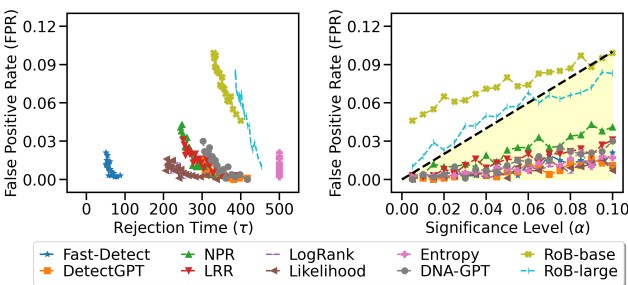

(a) Results for detecting the source of $y_t$ (2024 Olympic news vs. news generated by Gemini-1.5-Flash), with human-written text $x_t$ sampled from XSum. The scoring model used is Neo-2.7.

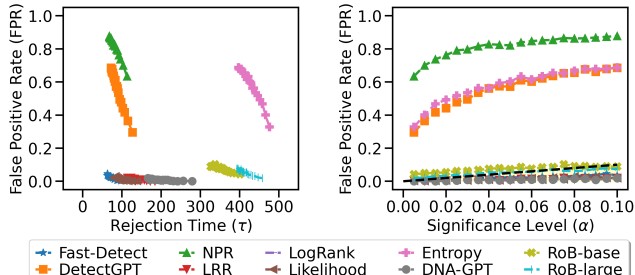

(b) Results for detecting the source of $y_t$ (2024 Olympic news vs. news generated by Gemini-1.5-Flash), with human-written text $x_t$ sampled from XSum. The scoring model used is Gemma-2B.

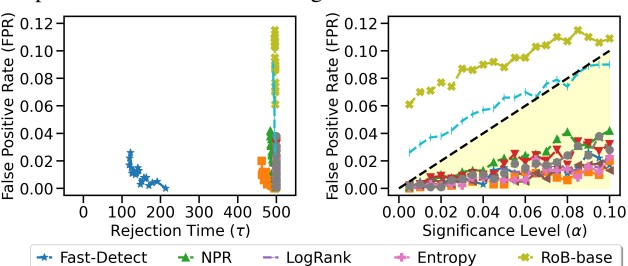

(c) Results for detecting the source of $y_t$ (2024 Olympic news vs. news generated by Gemini-1.5-Pro), with human-written text $x_t$ sampled from XSum. The scoring model used is Neo-2.7.

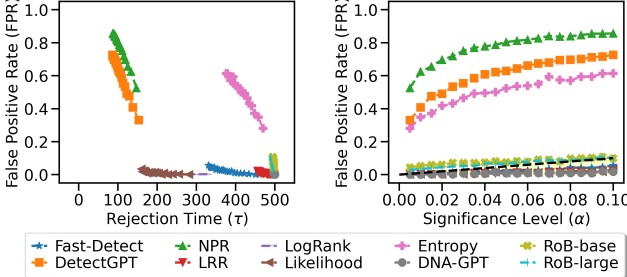

(d) Results for detecting the source of $y_t$ (2024 Olympic news vs. news generated by Gemini-1.5-Pro), with human-written text $x_t$ sampled from XSum. The scoring model used is Gemma-2B.

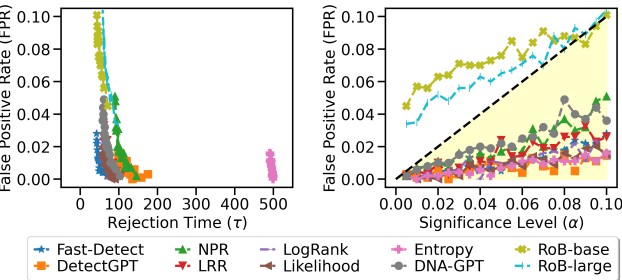

(e) Results for detecting the source of $y_t$ (2024 Olympic news vs. news generated by PaLM 2), with human-written text $x_t$ sampled from XSum. The scoring model used is Neo-2.7.

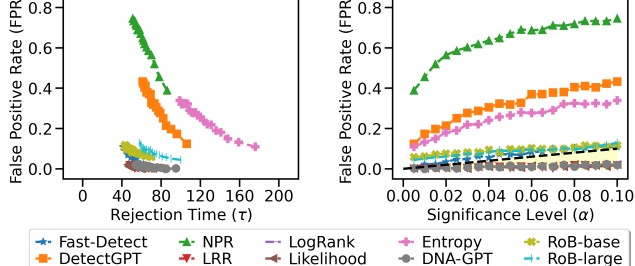

(f) Results for detecting the source of $y_t$ (2024 Olympic news vs. news generated by PaLM 2), with human-written text $x_t$ sampled from XSum. The scoring model used is Gemma-2B.

*Figure 6.* Results for detecting 2024 Olympic news and machine-generated news with our algorithm for Scenario 2. We use 3 source models: Gemini-1.5-Flash, Gemini-1.5-Pro and PaLM 2 to generate fake news and 2 scoring models: Neo-2.7, Gemma-2B. The left column displays results using the Neo-2.7 scoring model, while the right column presents results using the Gemma-2B scoring model. Score functions of supervised classifiers (RoB-base and RoB-large) are independent of scoring models.

*Table 2.* Values of $d_t$ used in Scenario 1, for which we assume that the range of $|g_t| = |\phi(x_t) - \phi(y_t)|$ is known beforehand. Specifically, $d_t$ is calculated as $\max_{i,j \leq 500} |\phi(x_i) - \phi(y_j)|$, where $\phi(x_i)$ is the score of $i$-th XSum text and $\phi(y_j)$ is the score of the $j$-th text generated by Gemini-1.5-Flash, prompted by pre-tokens of 2024 Olympic news. This calculation ensures that $d_t \geq |\phi(x_t) - \phi(y_t)|$ for any time point $1 \leq t \leq 500$. Every two columns starting from the third column represent the $d_t$ values for each $t$ used under $H_1$ and $H_0$ in each test scenario. For instance, the values in the third column Similarly, the fourth column calculates the maximum difference between the scores of all XSum texts and texts sample of 2024 Olympic news.. The derived $d_t$ values is then used to define the domain of $\theta_t$ in our algorithm, where $\theta_t \in [-1/2d_t, 0]$.

| Scoring Model | Score Function | XSum, Olympic | | XSum, Olympic | | XSum, Olympic | |
| | | Human, 1.5-Flash | Human, Human | Human, 1.5-Pro | Human, Human | Human, PaLM 2 | Human, Human |
|---|---|---|---|---|---|---|---|
| **Neo-2.7** | Fast-DetectGPT | 7.6444 | 5.9956 | 6.5104 | 6.1546 | 9.1603 | 5.8870 |
| | DetectGPT | 2.3985 | 2.3102 | 2.1416 | 2.2683 | 2.6095 | 2.7447 |
| | NPR | 0.1500 | 0.1436 | 0.1295 | 0.1465 | 0.1975 | 0.1353 |
| | LRR | 0.8129 | 0.5877 | 0.6400 | 0.5875 | 0.9793 | 0.5421 |
| | Logrank | 1.5861 | 1.6355 | 1.4065 | 1.6355 | 1.7298 | 1.6355 |
| | Likelihood | 2.3004 | 2.4540 | 1.9491 | 2.4540 | 2.6607 | 2.5559 |
| | Entropy | 1.6523 | 1.6630 | 1.5890 | 1.6702 | 1.9538 | 1.6265 |
| | DNA-GPT | 1.5063 | 1.5425 | 1.3621 | 1.5649 | 1.5455 | 1.6348 |
| | RoBERTa-base | 0.9997 | 0.9995 | 0.9995 | 0.9995 | 0.9997 | 0.9996 |
| | RoBERTa-large | 0.9983 | 0.9856 | 0.8945 | 0.8945 | 0.9992 | 0.8608 |
| **Gemma-2B** | Fast-DetectGPT | 7.7651 | 6.5619 | 7.3119 | 6.4343 | 8.6156 | 6.5640 |
| | DetectGPT | 2.9905 | 2.5449 | 2.3846 | 2.4274 | 2.6807 | 2.7878 |
| | NPR | 0.3357 | 0.2196 | 0.3552 | 0.2318 | 0.4118 | 0.2403 |
| | LRR | 1.1189 | 0.7123 | 0.8780 | 0.7109 | 1.2897 | 0.7717 |
| | Logrank | 1.5731 | 1.6467 | 1.4713 | 1.6468 | 1.7538 | 1.6397 |
| | Likelihood | 2.4934 | 2.4229 | 2.3944 | 2.4228 | 2.8379 | 2.4694 |
| | Entropy | 1.9791 | 1.9117 | 1.8572 | 1.8854 | 2.1359 | 1.9210 |
| | DNA-GPT | 1.3214 | 1.4808 | 1.2891 | 1.4607 | 1.5014 | 1.6296 |
| | RoBERTa-base | 0.9997 | 0.9995 | 0.9995 | 0.9995 | 0.9997 | 0.9996 |
| | RoBERTa-large | 0.9983 | 0.9856 | 0.8945 | 0.8945 | 0.9992 | 0.8608 |

source of $y_t$ as an LLM within a limited number of time steps under $H_1$.

Moreover, we found that the rejection time is related to the relative magnitude of $\Delta - \epsilon$ and $d_t - \epsilon$. According to the definition of nonnegative wealth $W_t^A = W_{t-1}^A (1 - \theta_t(g_t - \epsilon))$ or $W_t^B = W_{t-1}^B (1 - \theta_t(-g_t - \epsilon))$, large $-\theta_t$ within the range $[0, 1/2d_t]$ will result in large wealth which allows to quickly reach the threshold for wealth to correctly declare the unknown source as an LLM. Based on the previous proposition of the expected time upper bound for composite hypothesis, we guess that the actual rejection time in our experiment is probably related to the relative magnitude of $\Delta - \epsilon$ and $d_t - \epsilon$, where $d_t$ is a certain value for any $t$ in each test as shown in Table 2. We define the relative magnitude as $\frac{\Delta - \epsilon}{d_t - \epsilon}$ and sort the score functions by this ratio from largest to smallest for Scenario 1, as displayed in the Rank column in Table 3. This ranking roughly corresponds to the chronological order of rejection shown in Figure 5. The quick declaration of an LLM source when $y_t$ is generated by PaLM 2 and the slower rejection of $H_0$ when $y_t$ is generated by Gemini-1.5-Pro in Figure 5 could be attributed to the relatively larger and smaller values of $\frac{\Delta - \epsilon}{d_t - \epsilon}$, respectively. The negative ratios arise because $\Delta < \epsilon$, leading to the vertical lines in the figures. For Scenario 2, the values of $\frac{\Delta - \epsilon}{d_t - \epsilon}$ are provided in Table 4, and the conclusions from Scenario 1 extend to Scenario 2.

Thus, we prefer a larger discrepancy between scores of human-written texts and machine-generated texts, which can increase $\Delta$. Furthermore, a smaller variation among scores of texts from the same source can reduce $\epsilon$. These properties facilitate a shorter rejection time under $H_1$.

Based on the results, when we know the actual value of $d_t$ and $\epsilon$, our algorithm is very effective. When we estimate their values based on the previous samples that we get, the algorithm can still exhibit a good performance for most score functions. It can be inferred that the rejection time and FPRs are effected by the score function $\phi(\cdot)$ and the scoring model that we select. If the configuration can further amplify the score discrepancy between human-written texts and machine-generated texts, the rejection time can be shortened. If the the score discrepancy between human texts is small, the value FPR will be low.

*Table 3.* Values of the ratio $\frac{\Delta - \epsilon}{d_t - \epsilon}$ for Scenario 1, where $\Delta$ and $\epsilon$ are listed in Table 1, $d_t$ are shown in Table 2. We sort the score function according to the ratio from largest to smallest, as shown in the Rank column. This ranking roughly corresponds to the chronological order of rejection in in Figure 5.

| Scoring Model | Score Function | Human, 1.5-Flash | | Human, 1.5-Pro | | Human, PaLM 2 | |
|---|---|---|---|---|---|---|---|
| | | Ratio | Rank | Ratio | Rank | Ratio | Rank |
| Neo-2.7 | Fast-DetectGPT | 0.2905 | 1 | 0.1519 | 1 | 0.3675 | 3 |
| | DetectGPT | 0.1562 | 3 | 0.1337 | 2 | 0.2303 | 7 |
| | NPR | 0.1467 | 4 | 0.1093 | 3 | 0.1995 | 9 |
| | LRR | 0.0920 | 7 | -0.0064 | 8 | 0.2373 | 6 |
| | Logrank | 0.1336 | 6 | 0.0556 | 5 | 0.2568 | 5 |
| | Likelihood | 0.1458 | 5 | 0.0806 | 4 | 0.2608 | 4 |
| | Entropy | -0.0181 | 10 | -0.0436 | 10 | 0.0732 | 10 |
| | DNA-GPT | 0.0687 | 8 | -0.0060 | 7 | 0.2089 | 8 |
| | RoBERTa-base | 0.1892 | 2 | -0.0215 | 9 | 0.6205 | 1 |
| | RoBERTa-large | 0.0662 | 9 | -0.0001 | 6 | 0.4032 | 2 |
| Gemma-2B | Fast-DetectGPT | 0.2163 | 1 | 0.0498 | 7 | 0.3848 | 3 |
| | DetectGPT | 0.1368 | 6 | 0.1311 | 1 | 0.2151 | 8 |
| | NPR | 0.1218 | 8 | 0.0690 | 5 | 0.1989 | 9 |
| | LRR | 0.1334 | 7 | 0.0823 | 4 | 0.2933 | 6 |
| | Logrank | 0.1849 | 3 | 0.1157 | 2 | 0.3104 | 4 |
| | Likelihood | 0.1844 | 4 | 0.1134 | 3 | 0.3089 | 5 |
| | Entropy | -0.0044 | 10 | 0.0035 | 8 | 0.1155 | 10 |
| | DNA-GPT | 0.1498 | 5 | 0.0527 | 6 | 0.2366 | 7 |
| | RoBERTa-base | 0.1892 | 2 | -0.0215 | 10 | 0.6205 | 1 |
| | RoBERTa-large | 0.0662 | 9 | -0.0001 | 9 | 0.4032 | 2 |

*Table 4.* Values of the ratio $\frac{\Delta - \epsilon}{d_t - \epsilon}$ for Scenario 2, where $\Delta$ and $\epsilon$ are listed in Table 1 and Table 5 respectively, $d_t$ are shown in Table 6. We sort the scoring function according to the ratio from largest to smallest, as shown in the Rank column. This ranking roughly corresponds to the chronological order of rejection in in Figure 6.

| Scoring Model | Score Function | Human, 1.5-Flash | | Human, 1.5-Pro | | Human, PaLM 2 | |
|---|---|---|---|---|---|---|---|
| | | Ratio | Rank | Ratio | Rank | Ratio | Rank |
| Neo-2.7 | Fast-DetectGPT | 0.1895 | 1 | 0.0874 | 1 | 0.2420 | 2 |
| | DetectGPT | 0.0427 | 7 | 0.0103 | 2 | 0.1024 | 9 |
| | NPR | 0.0592 | 2 | 0.0076 | 3 | 0.1268 | 8 |
| | LRR | 0.0474 | 5 | -0.0653 | 7 | 0.1568 | 7 |
| | Logrank | 0.0576 | 4 | -0.0218 | 5 | 0.1687 | 4 |
| | Likelihood | 0.0582 | 3 | -0.0132 | 4 | 0.1680 | 5 |
| | Entropy | -0.0883 | 10 | -0.1162 | 10 | -0.0051 | 10 |
| | DNA-GPT | 0.0405 | 8 | -0.0298 | 6 | 0.1636 | 6 |
| | RoBERTa-base | 0.0461 | 6 | -0.1042 | 9 | 0.2713 | 1 |
| | RoBERTa-large | 0.0092 | 9 | -0.0942 | 8 | 0.1814 | 3 |
| Gemma-2B | Fast-DetectGPT | 0.1562 | 1 | 0.0368 | 5 | 0.2444 | 2 |
| | DetectGPT | 0.1351 | 3 | 0.1147 | 2 | 0.1552 | 8 |
| | NPR | 0.1492 | 2 | 0.1167 | 1 | 0.1961 | 6 |
| | LRR | 0.0867 | 6 | 0.0117 | 7 | 0.1959 | 5 |
| | Logrank | 0.1205 | 5 | 0.0607 | 4 | 0.2086 | 4 |
| | Likelihood | 0.1267 | 4 | 0.0702 | 3 | 0.2144 | 3 |
| | Entropy | 0.0270 | 9 | 0.0293 | 6 | 0.1017 | 10 |
| | DNA-GPT | 0.0713 | 7 | -0.0144 | 8 | 0.1747 | 9 |
| | RoBERTa-base | 0.0462 | 8 | -0.1007 | 9 | 0.2737 | 1 |
| | RoBERTa-large | 0.0060 | 10 | -0.1013 | 10 | 0.1803 | 7 |

**Comparisons with Baselines.** Permutation test is a fixed-time test. The goal is to test whether the source of text $y_t$ is the same as that of the human-written text $x_t$, i.e., whether their scores are from the same distribution. If we choose the mean

*Table 5.* Average values of $\epsilon$ used in Scenario 2 estimated by 20 texts in sequence of human-written text $x_t$. Every two columns starting from the third column represent the $\epsilon$ values used for $H_1$ and $H_0$ for each test scenario. For instance, the third column calculates $\epsilon$ for tests between XSum text and Gemini-1.5-Flash-generated text sequences by scoring 20 XSum texts, dividing them into two equal groups, and then doubling the average absolute mean difference between these groups across 1000 random shuffles. The fourth column follows the same method to determine the $\epsilon$ value for tests between XSum texts and 2024 Olympic news. This is calculated as $\epsilon = 2 \cdot \frac{1}{1000} \sum_{n=1}^{1000} \left| \sum_{i=1}^{10} \phi(x_i^{(n)})/10 - \sum_{i=11}^{20} \phi(x_i^{(n)})/10 \right|$, where $\phi(x_i^{(n)})$ denotes the score of the $i$-th text after the $n$-th random shuffling of 20 text scores.

| Scoring Model | Score Function | XSum, Olympic | | XSum, Olympic | | XSum, Olympic | |
| --- | --- | --- | --- | --- | --- | --- | --- |
| | | Human, 1.5-Flash | Human, Human | Human, 1.5-Pro | Human, Human | Human, PaLM 2 | Human, Human |
| **Neo-2.7** | Fast-DetectGPT | 0.6357 | 0.6371 | 0.6395 | 0.6417 | 0.6415 | 0.6426 |
| | DetectGPT | 0.2781 | 0.2807 | 0.2840 | 0.2870 | 0.2851 | 0.2847 |
| | NPR | 0.0141 | 0.0141 | 0.0145 | 0.0146 | 0.0140 | 0.0141 |
| | LRR | 0.0665 | 0.0666 | 0.0674 | 0.0676 | 0.0652 | 0.0649 |
| | Logrank | 0.1634 | 0.1632 | 0.1624 | 0.1624 | 0.1597 | 0.1590 |
| | Likelihood | 0.2448 | 0.2450 | 0.2455 | 0.2426 | 0.2419 | 0.2416 |
| | Entropy | 0.1994 | 0.1978 | 0.2022 | 0.2000 | 0.1973 | 0.2004 |
| | DNA-GPT | 0.1404 | 0.1394 | 0.1321 | 0.1314 | 0.1345 | 0.1320 |
| | RoBERTa-base | 0.1438 | 0.1459 | 0.1463 | 0.1496 | 0.1261 | 0.1206 |
| | RoBERTa-large | 0.0768 | 0.0787 | 0.0732 | 0.0740 | 0.0821 | 0.0853 |
| **Gemma-2B** | Fast-DetectGPT | 0.6806 | 0.6785 | 0.6749 | 0.6726 | 0.6781 | 0.6825 |
| | DetectGPT | 0.2731 | 0.2725 | 0.2685 | 0.2664 | 0.2822 | 0.2840 |
| | NPR | 0.0170 | 0.0168 | 0.0171 | 0.0171 | 0.0169 | 0.0169 |
| | LRR | 0.0724 | 0.0728 | 0.0732 | 0.0735 | 0.0725 | 0.0719 |
| | Logrank | 0.1547 | 0.1542 | 0.1567 | 0.1550 | 0.1556 | 0.1521 |
| | Likelihood | 0.2438 | 0.2414 | 0.2472 | 0.2473 | 0.2427 | 0.2429 |
| | Entropy | 0.2082 | 0.2088 | 0.2120 | 0.2092 | 0.2076 | 0.2078 |
| | DNA-GPT | 0.1354 | 0.1355 | 0.1317 | 0.1325 | 0.1278 | 0.1286 |
| | RoBERTa-base | 0.1438 | 0.1433 | 0.1437 | 0.1448 | 0.1198 | 0.1208 |
| | RoBERTa-large | 0.0807 | 0.0800 | 0.0750 | 0.0743 | 0.0831 | 0.0823 |

value as the test statistic, the null hypothesis is that the means are equal ($H_0 : \mu_x = \mu_y$) with a batch size of $k$. Once we have generated $k$ samples from these two sources, we conduct the test. When $H_0$ is true, the samples are drawn from the same distribution, which means that the observed discrepancy between the two batches of scores is supposed to be minimal. The test determines whether this difference between the sample means is large enough to reject $H_0$ at a significance level. Specifically, we compare the p-value with the significance level $\alpha$ for each batch in an uncorrected test, and $\alpha/2^j$ for the $j$-th batch in a corrected test. The permutation test is conducted as below:

(1) Calculate the observed mean difference $\Delta = \left| \sum_{i=1}^{k} \phi(x_i)/k - \sum_{j=1}^{k} \phi(y_j)/k \right|$ of these two batches, and assume that $H_0$ is true;

(2) Combine these two sequences into one dataset, reshuffle the data, and divide it into two new groups. This is the permutation operation. Calculate the sampled absolute mean difference for the $n$-th permutation, $\tilde{\Delta}^{(n)} = |\sum_{i=1}^{k} \phi(\tilde{x}_i^{(n)})/k - \sum_{j=1}^{k} \phi(\tilde{y}_j^{(n)})/k|$;

(5) Repeat step (2) for a sufficient number of permutations ($n = 2,000$ in our test);

(6) Calculate the p-value, which is the proportion of permutations where $\tilde{\Delta}^{(n)} > \Delta$, relative to the total number of permutations ($2,000$). If the p-value is greater than the significance level $\alpha$, we retain $H_0$; otherwise we reject $H_0$.

Proceed to the next batch if $H_0$ is retained in this batch test, and continue the above process until $H_0$ is rejected or all data are tested.

In the experiment, we consider the composite hypothesis testing, which means the null hypothesis is $H_0 : |\mu_x - \mu_y| \leq \epsilon$. If we still use the above permutation test, it will become much easier for $\Delta \geq \tilde{\Delta}^{(n)}$ to hold, even when $H_0$ is actually true. This would result in significantly higher FPRs. Thus, we only check p-values when the observed $\Delta$ exceeds the estimated $\epsilon$.

*Table 6.* Average values of $d_t$ used in Scenario 2 estimated by using the first 10 texts of each sequence. Every two columns starting from the third column represent the $d_t$ values used for $H_1$ and $H_0$ for each test scenario. Specifically, for the third column, we get the first 10 samples from XSum and the first 10 observed texts generated by Gemini-1.5-Flash. We then calculate the maximum difference between $\phi(x_i)$ and $\phi(y_j)$ for any $1 \le i \le 10$, $1 \le j \le 10$. We double this maximum value to estimate $d_t$ value, i.e., $d_t = 2 \cdot \max_{i,j \le 10} |\phi(x_i) - \phi(y_j)|$. The fourth column follows a similar calculation for detecting 2024 Olympic news with XSum texts, where $\phi(x_i)$ represents score of $i$-th text from XSum, $\phi(y_j)$ denotes the score of the $j$-th text of 2024 Olympic news.

| Scoring Model | Score Function | XSum, Olympic | | XSum, Olympic | | XSum, Olympic | |
| --- | --- | --- | --- | --- | --- | --- | --- |
| | | Human, 1.5-Flash | Human, Human | Human, 1.5-Pro | Human, Human | Human, PaLM 2 | Human, Human |
| **Neo-2.7** | Fast-DetectGPT | 10.3586 | 6.6788 | 8.1840 | 6.7517 | 13.0086 | 6.8456 |
| | DetectGPT | 2.9396 | 2.6967 | 2.8232 | 2.7183 | 3.4076 | 2.7313 |
| | NPR | 0.1680 | 0.1358 | 0.1464 | 0.1397 | 0.2178 | 0.1337 |
| | LRR | 0.8620 | 0.6418 | 0.6572 | 0.6298 | 1.3115 | 0.6294 |
| | Logrank | 1.8230 | 1.6746 | 1.5966 | 1.6676 | 2.1740 | 1.6807 |
| | Likelihood | 2.7089 | 2.5370 | 2.4016 | 2.4975 | 3.3493 | 2.5529 |
| | Entropy | 1.9123 | 1.9560 | 1.8840 | 1.9841 | 2.0712 | 1.9249 |
| | DNA-GPT | 1.4570 | 1.5696 | 1.3530 | 1.5374 | 1.8097 | 1.6042 |
| | RoBERTa-base | 1.9404 | 1.1136 | 1.2745 | 1.1795 | 1.9993 | 1.0390 |
| | RoBERTa-large | 1.3443 | 0.6548 | 0.5857 | 0.5713 | 1.9435 | 0.6014 |
| **Gemma-2B** | Fast-DetectGPT | 10.0337 | 7.4171 | 7.6691 | 7.4983 | 13.1693 | 7.6051 |
| | DetectGPT | 3.5422 | 3.0998 | 3.3271 | 3.0398 | 3.8780 | 3.1375 |
| | NPR | 0.3264 | 0.2326 | 0.2795 | 0.2346 | 0.4432 | 0.2270 |
| | LRR | 1.0874 | 0.7253 | 0.8615 | 0.7283 | 1.6474 | 0.7234 |
| | Logrank | 1.9435 | 1.6850 | 1.7377 | 1.6759 | 2.2468 | 1.6162 |
| | Likelihood | 3.1277 | 2.6890 | 2.8156 | 2.6686 | 3.6366 | 2.5868 |
| | Entropy | 2.3749 | 2.4105 | 2.3417 | 2.3869 | 2.6341 | 2.3166 |
| | DNA-GPT | 1.4319 | 1.3889 | 1.3308 | 1.3798 | 1.7274 | 1.4109 |
| | RoBERTa-base | 1.9343 | 1.1536 | 1.2867 | 1.1800 | 1.9993 | 0.9913 |
| | RoBERTa-large | 1.3832 | 0.6636 | 0.5695 | 0.5501 | 1.9499 | 0.6488 |

The rejection time and FPRs that we plot are the average values across 1000 repeated runs under each significance level $\alpha$.

Permutation test is very time-consuming, because if we have m samples for each group, then we will get $m/k$ batches, each batch need to conduct the above steps.

As observed in Figure 7, the permutation tests without correction always have higher FPRs under $H_0$ than that with corrected significance levels. This phenomenon aligns with the fact that without significance level adjustments, it is impossible to control the type-I error. Permutation tests with large batch sizes demonstrate relatively low FPRs, which are approximately equal to 0. However, this seemingly excellent performance is due to the fact that the preset $\epsilon$ values are much larger than the actual absolute difference in mean scores between sequences of XSum texts and 2024 Olympic news. This discrepancy results in fewer or no p-value checks, thus sustaining $H_0$. Consequently, the FPRs are nearly identical across each significance level $\alpha$. Permutation tests are sensitive to the discrepancies between two sequences, and the $\Delta$ value tends to change more with smaller batch sizes due to variation among scores of texts from the same source. Although a permutation test can reject $H_0$ after the first batch test, it consistently exhibit an FPR greater than $\alpha$. Moreover, even if we set the value of $\epsilon$ based on a much larger sample size, rather than from estimates derived from a few points, there will still be variance between the $\Delta$ value calculated in batches and the preset $\epsilon$. As long as $\Delta$ calculated by a batch of samples is greater than the preset $\epsilon$, the permutation test is likely to have a large FPR under $H_0$. Compared to fixed-time methods, our method can use parameter estimates based on just a few points to ensure faster rejection and lower FPRs. It can save time with high acuuracy especially when there is no prior knowledge of the threshold $\epsilon$ for composite hypotheses.

## H. Experimental Results of Detecting Texts from Three Domains

We also test on the dataset of Bao et al. (2023) to explore the influence of text domains on the detection result of our method. In this experiment, we only consider Scenario 1. We let $x_t$ be human-written text from three datasets following Mitchell et al. (2023), each chosen to represent a typical LLMs application scenario. Specifically, we incorporate news articles sourced

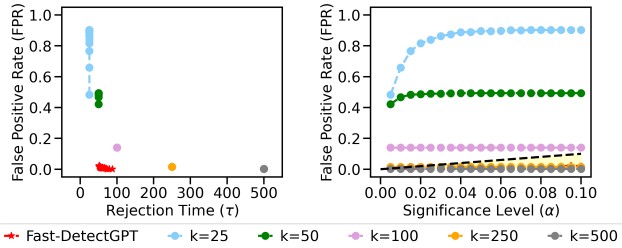

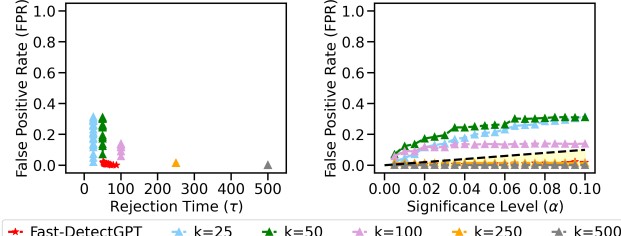

(a) Comparisons between our method and the permutation test without correction, $y_t$ are 2024 Olympic news or news generated by Gemini-1.5-Flash, with human-written text $x_t$ sampled from XSum. The scoring function used is Neo-2.7.

(b) Comparisons between our method and the permutation test with correction, $y_t$ are 2024 Olympic news or news generated by Gemini-1.5-Flash, with human-written text $x_t$ sampled from XSum. The scoring function used is Neo-2.7.

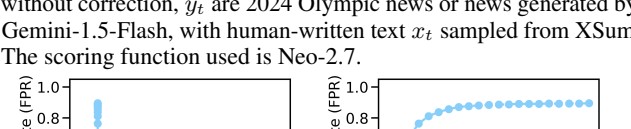

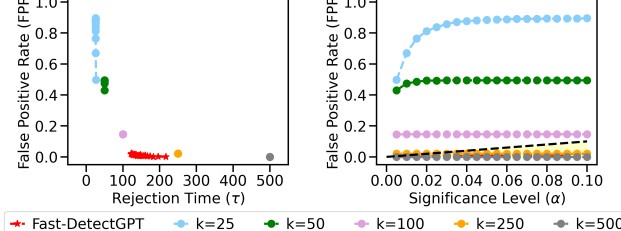

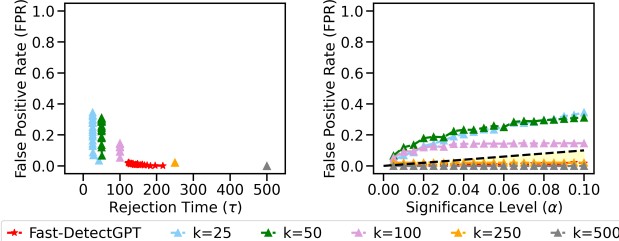

(c) Comparisons between our method and the permutation test without correction, $y_t$ are 2024 Olympic news or news generated by Gemini-1.5-Pro, with human-written text $x_t$ sampled from XSum. The scoring function used is Neo-2.7.

(d) Comparisons between our method and the permutation test with correction, $y_t$ are 2024 Olympic news or news generated by Gemini-1.5-Pro, with human-written text $x_t$ sampled from XSum. The scoring function used is Neo-2.7.

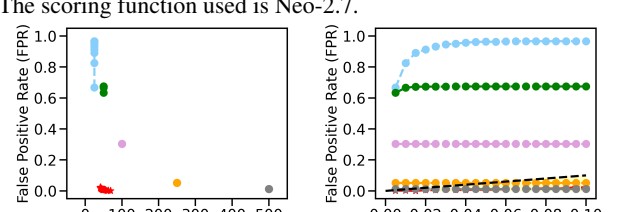

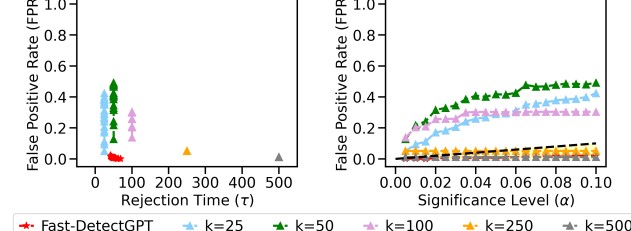

(e) Comparisons between our method and the permutation test without correction, $y_t$ are 2024 Olympic news or news generated by PaLM 2 with human-written text $x_t$ sampled from XSum. The scoring function used is Neo-2.7.

(f) Comparisons between our method and the permutation test with correction, $y_t$ are 2024 Olympic news or news generated by PaLM 2 with human-written text $x_t$ sampled from XSum. The scoring function used is Neo-2.7.

Figure 7. Comparisons between our method and two baselines for detecting 2024 Olympic news and machine-generated news. Fake news are generated by 3 source models: Gemini-1.5-Flash, Gemini-1.5-Pro and PaLM 2. The scoring model used is Neo-2.7 with the score function of Fast-DetectGPT. We consider five batch sizes: $k = 25, 50, 100, 250, 500$. The left column displays results from the permutation test without correction, while the right column presents results of permutation test with corrected significance levels $\alpha$ for each batch test.

from the XSum dataset (Narayan et al., 2018), stories from Reddit WritingPrompts dataset (Fan et al., 2018) and long-form answers written by human experts from the PubMedQA dataset (Jin et al., 2019). Then, the capability of our algorithm is evaluated by detecting the source of texts $y_t$ originated from the above source models or human datasets. Source models involved in this experiment are GPT-3 (Brown, 2020), ChatGPT (OpenAI, 2022), and GPT-4 (Achiam et al., 2023) while the scoring model is Neo-2.7 (Black et al., 2021). The perturbation function for DetectGPT and NPR is T5-11B, and the sampling model for Fast-DetectGPT is GPT-J-6B.

Figure 8a and Figure 8b present the averaged results when the two text streams are from the same domain and different domains, respectively—that is, when both sequences are sampled from or prompted using the same dataset, or different datasets. Specifically, we compute the average rejection time and false positive rate (FPR) under each significance level $\alpha$, aggregated over three LLMs (GPT-3, ChatGPT, GPT-4) and three domain configurations (three same-domain or three different-domain). A comparison of the two figures reveals that detection is faster when the prepared texts and the texts under evaluation originate from the same domain. For all score functions, our algorithm consistently controls FPRs below the target significance level $\alpha$, and the corresponding average rejection times remain below 150. Among all functions, Fast-DetectGPT is the fastest to reject $H_0$, with average value of $\tau$ being around 40 for same-domain texts and about 80 for different-domain texts.

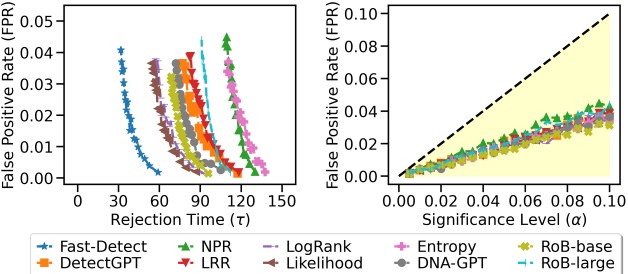 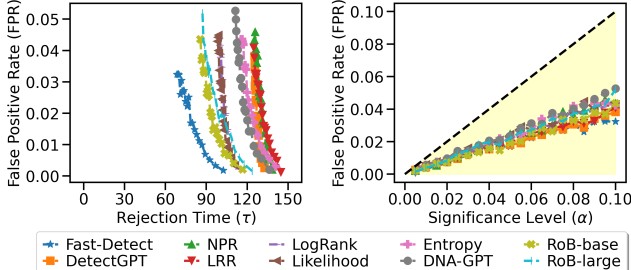

(a) Averaged test results with texts $x_t$ and $y_t$ from the same domain across three source models and three text domains. The scoring model used is Neo-2.7.

(b) Averaged test results with texts $x_t$ and $y_t$ from different domains across three source models and three text domains. The scoring model used is Neo-2.7.

*Figure 8.* Average results of Scenario 1. There are three source models: GPT-3, ChatGPT, and GPT-4, and three domains. For (a), two text sequences are both of XSum, Writing or PubMed dataset. For (b), two sequences are of XSum and Writing, XSum and PubMed, Writing and PubMed. Each sequence has 150 samples, which means the time budget is $T = 150$. The left subfigure in (a) and (b) shows the average rejection time under $H_1$ versus. the averaged FPRs under $H_0$ under each significance level $\alpha$. Thus, plots closer to the left bottom corner are preferred, which indicate correct detection of an LLM with shorter rejection times and lower FPRs. In the right subfigure of each panel, the black dashed line along with the shaded area illustrates the expected FPR, consistently maintained below the significance level $\alpha$.

Specifically, according to Figure 10 , when texts are both sampled from PubMedQA datasets, the behaviour of most score functions are worse than that for texts from XSum and WritingPrompts. Specifically, it costs more time for them to reject $H_0$. There are more vertical lines in Figure 9c, 10i and 9i, which means texts generated by GPT-4 are more challenging for our algorithm when using certain score functions such as RoBERTa-base/large to detect before $T = 150$.

Figure 8b illustrates the performance of our algorithm with different score functions when detecting two streams of different-domain texts. All functions can guarantee FPRs below the $\alpha$-significance level. When $x_t$ is from XSum and $y_t$ is generated by GPT-4 based on Writing, only Fast-DetectGPT can declare the source of $y_t$ as an LLM before $T = 150$. Another interesting phenomenon is that when $y_t$ is generated by GPT-3 based on PubMed, tests with most score functions fail to successfully identify its source as an LLM before the time budget expires, regardless of the domain of the human text used as $x_t$ for detection. Only the score functions of two supervised classifiers consistently reject $H_0$ before 150, which is shown as Figure 9g, 10d and 10g.

The parameter $\theta_t$ is chosen from the range $[-1/2d_t, 0]$. The value of $d_t$ for any $t$ is equal to the maximum absolute difference between two sequences of scores for each test, as can be seen in Table 9 for the same-domain texts and Table 9 for the different-domain texts. We get the absolute difference $\Delta$ between the scores $\phi(x_t)$ and $\phi(y_t)$ for texts from the same domain and different domains involved in our experiments, as shown in the Table 11 and Table 12 respectively. Averaged values of $\Delta$ across three source models (GPT-3, ChatGPT and GPT-4) when texts $x_t$ and $y_t$ are from the same domain and different domains are presented in Table 7 and 8, respectively. In practice, we can also select the value of $d_t$ and $\epsilon$ based on the hint

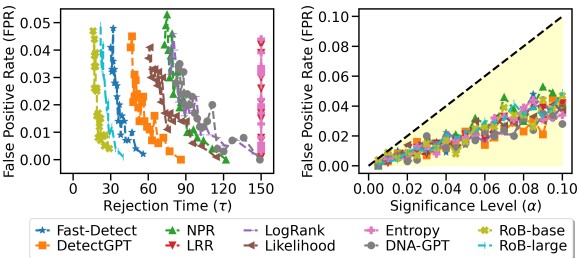

(a) Same domain: $x_t$ is sampled from XSum, $y_t$ is from XSum or generated by GPT-3 based on XSum.

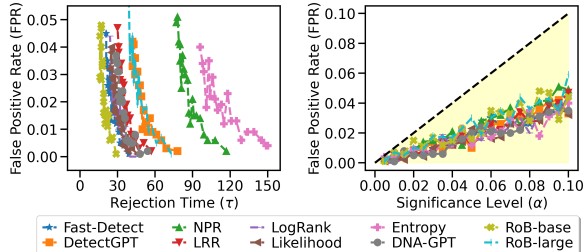

(b) Same domain: $x_t$ is sampled from XSum, $y_t$ is from XSum or generated by ChatGPT based on XSum.

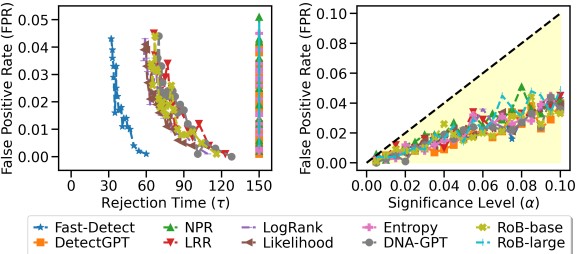

(c) Same domain: $x_t$ is sampled from XSum, $y_t$ is from XSum or generated by GPT-4 based on XSum.

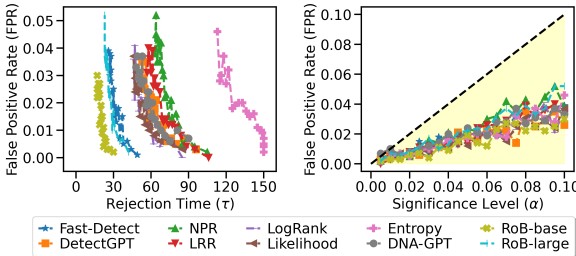

(d) Same domain: $x_t$ is sampled from Writing, $y_t$ is from Writing or generated by GPT-3 based on Writing.

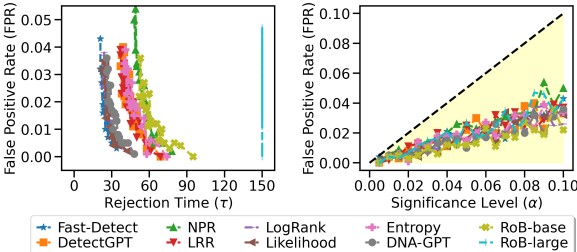

(e) Same domain: $x_t$ is sampled from Writing, $y_t$ is from Writing or generated by ChatGPT based on Writing.

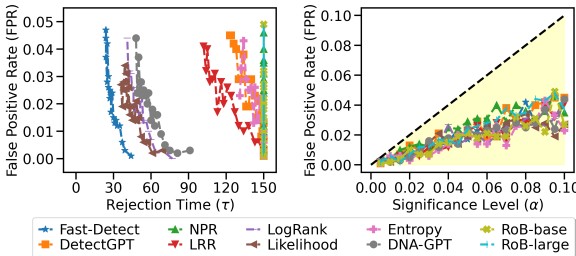

(f) Same domain: $x_t$ is sampled from Writing, $y_t$ is from Writing or generated by GPT-4 based on Writing.

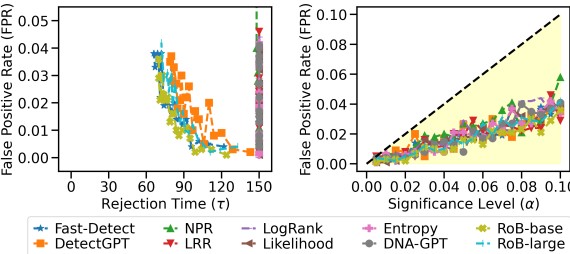

(g) Same domain: $x_t$ is sampled from PubMed, $y_t$ is from PubMed or generated by GPT-3 based on PubMed.

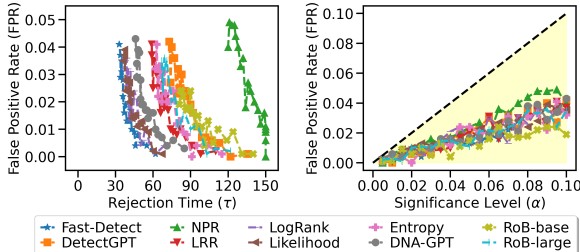

(h) Same domain: $x_t$ is sampled from PubMed, $y_t$ is from PubMed or generated by ChatGPT based on PubMed.

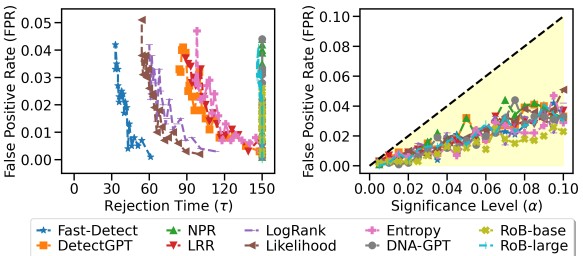

(i) Same domain: $x_t$ is sampled from PubMed, $y_t$ is from PubMed or generated by GPT-4 based on PubMed.

*Figure 9.* Test results: mean rejection times (under $H_1$) and FPRs (under $H_0$) under each significance level $\alpha$ using 10 score functions, with texts $x_t$ and $y_t$ from the same domain. There are 3 source models: GPT-3, ChatGPT and GPT-4. Scoring model: Neo-2.7.

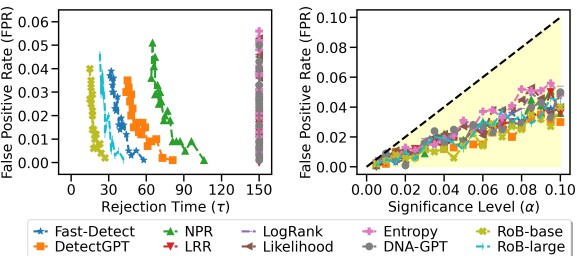

(a) Different domains: $x_t$ is sampled from XSum, $y_t$ is from Writing or generated by GPT-3 based on Writing.

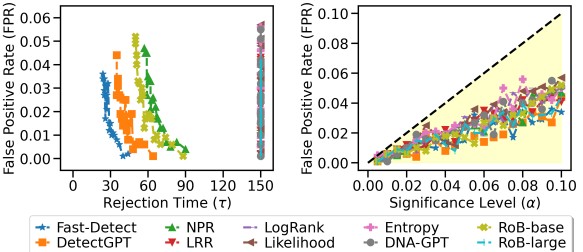

(b) Different domains: $x_t$ is sampled from XSum, $y_t$ is from Writing or generated by ChatGPT based on Writing.

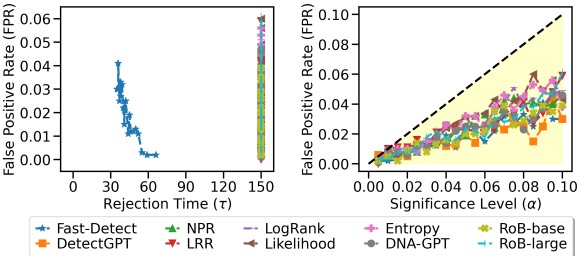

(c) Different domains: $x_t$ is sampled from XSum, $y_t$ is from Writing or generated by GPT-4 based on Writing.

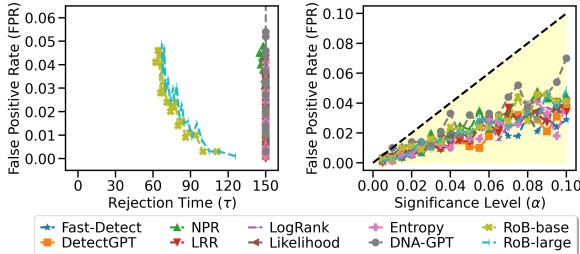

(d) Different domains: $x_t$ is sampled from XSum, $y_t$ is from PubMed or generated by GPT-3 based on PubMed.

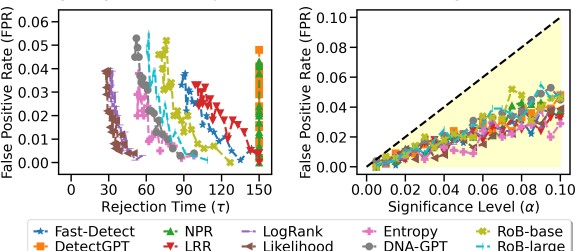

(e) Different domains: $x_t$ is sampled from XSum, $y_t$ is from PubMed or generated by ChatGPT based on PubMed.

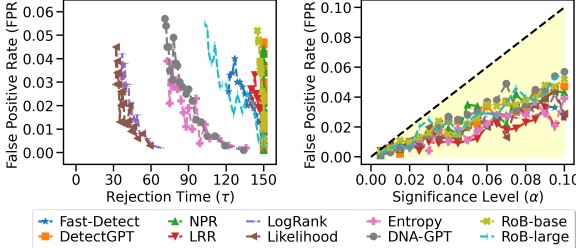

(f) Different domains: $x_t$ is sampled from XSum, $y_t$ is from PubMed or generated by GPT-4 based on PubMed.

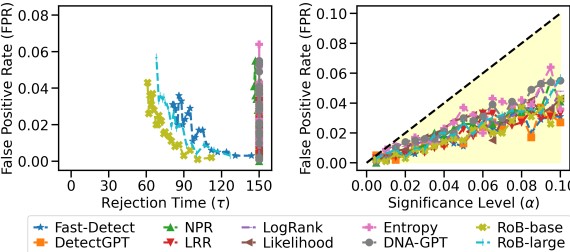

(g) Different domains: $x_t$ is sampled from Writing, $y_t$ is from PubMed or generated by GPT-3 based on PubMed.

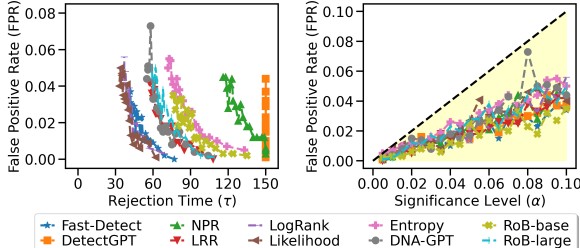

(h) Different domains: $x_t$ is sampled from Writing, $y_t$ is from PubMed or generated by ChatGPT based on PubMed.

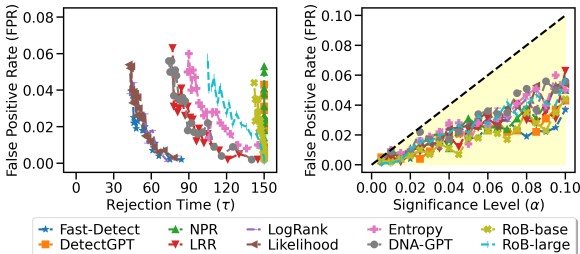

(i) Different domains: $x_t$ is sampled from Writing, $y_t$ is from PubMed or generated by GPT-4 based on PubMed.

*Figure 10.* Test results: mean rejection times (under $H_1$) and FPRs (under $H_0$) under each significance level $\alpha$ using 10 score functions, with texts $x_t$ and $y_t$ from different domains. There are 3 source models: GPT-3, ChatGPT and GPT-4. Scoring model: Neo-2.7.

of the bound $d_t$ or by the previous observed samples.

Since we let $\epsilon$ equal to the actual $\Delta$ value of two sequences of human texts, which ensures FPRs of all score functions for each significance level $\alpha$ remain below $\alpha$. As we have mentioned previously, our algorithm can ensure a nonnegative supermartingale wealth under $H_0$ and thus control the type-I error. Besides, the relative magnitude of $\Delta - \epsilon$ and $d_t - \epsilon$ would influence the rejection time under $H_1$.

*Table 7.* Average Values of $\Delta$ derived by using Neo-2.7 as the scoring model and various score functions to texts from the same domain across three source models (GPT-3, ChatGPT and GPT-4). Every two columns starting from the third column represent the average $\Delta$ values across three LLMs under $H_1$ and $H_0$ in each test scenario. For instance, the third column presents the average absolute difference in mean scores between 150 Xsum texts and 150 texts generated by LLMs based on XSum texts. The fourth column illustrates the average $\Delta$ value between 150 XSum texts and 150 XSum texts. Values in Column "Human, Human" are also used to set $\epsilon$ value for tests.

| Scoring Model | Score Function | XSum, XSum | | Writing, Writing | | PubMed, PubMed | |
|---|---|---|---|---|---|---|---|
| | | Human, LLMs | Human, Human | Human, LLMs | Human, Human | Human, LLMs | Human, Human |
| **Neo-2.7** | Fast-DetectGPT | 2.2235 | 0.0513 | 2.5690 | 0.0138 | 1.0891 | 0.0179 |
| | DetectGPT | 0.4048 | 0.0562 | 0.4985 | 0.0121 | 0.2206 | 0.0061 |
| | NPR | 0.0200 | 0.0025 | 0.0276 | 0.0001 | 0.0135 | 0.0018 |
| | LRR | 0.0919 | 0.0106 | 0.1002 | 0.0047 | 0.0714 | 0.0075 |
| | Logrank | 0.2710 | 0.0303 | 0.4082 | 0.0037 | 0.2707 | 0.0257 |
| | Likelihood | 0.4384 | 0.0420 | 0.6518 | 0.0145 | 0.4604 | 0.0354 |
| | Entropy | 0.1017 | 0.0344 | 0.2393 | 0.0086 | 0.2135 | 0.0243 |
| | DNA-GPT | 0.1917 | 0.0317 | 0.2852 | 0.0232 | 0.5531 | 0.1456 |
| | RoBERTa-base | 0.4585 | 0.0142 | 0.2825 | 0.0186 | 0.1330 | 0.0143 |
| | RoBERTa-large | 0.2165 | 0.0090 | 0.1387 | 0.0057 | 0.1088 | 0.0072 |

*Table 8.* Average Values of $\Delta$ derived by using different score functions to texts from the defferent domains across three source models (GPT-3, ChatGPT and GPT-4). Neo-2.7 is the scoring model used for the first eight score functions. Every two columns starting from the third column represent the average $\Delta$ values across three LLMs under $H_1$ and $H_0$ in each test scenario. For instance, the third column presents the average absolute difference in mean scores between 150 Xsum texts and 150 texts generated by LLMs based on Writing texts. The fourth column illustrates the average $\Delta$ value between 150 XSum texts and 150 Writing texts.

| Scoring Model | Score Function | XSum, Writing | | XSum, PubMed | | Writing, PubMed | |
|---|---|---|---|---|---|---|---|
| | | Human, LLMs | Human, Human | Human, LLMs | Human, Human | Human, LLMs | Human, Human |
| **Neo-2.7** | Fast-DetectGPT | 2.2848 | 0.2841 | 0.7267 | 0.3624 | 1.0108 | 0.0783 |
| | DetectGPT | 0.4828 | 0.0342 | 0.0786 | 0.2992 | 0.0630 | 0.2835 |
| | NPR | 0.0246 | 0.0030 | 0.0115 | 0.0031 | 0.0145 | 0.0012 |
| | LRR | 0.0428 | 0.0665 | 0.0572 | 0.0219 | 0.1094 | 0.0447 |
| | Logrank | 0.1050 | 0.3149 | 0.2429 | 0.0278 | 0.5579 | 0.2872 |
| | Likelihood | 0.1603 | 0.5025 | 0.4548 | 0.0081 | 0.9573 | 0.4969 |
| | Entropy | 0.2071 | 0.4464 | 0.2778 | 0.1154 | 0.7242 | 0.5617 |
| | DNA-GPT | 0.1301 | 0.1551 | 3.0447 | 2.4916 | 3.1997 | 2.6467 |
| | RoBERTa-base | 0.3284 | 0.0459 | 0.3304 | 0.1974 | 0.2844 | 0.1514 |
| | RoBERTa-large | 0.1351 | 0.0073 | 0.1962 | 0.0874 | 0.1889 | 0.0801 |

*Table 9.* Values of $d_t$ with the assumption that we know the range of $|g_t| = |\phi(x_t) - \phi(y_t)|$ beforehand. Every two columns starting from the third column represent the $d_t$ values for each $t$ used under $H_1$ and $H_0$ in each test scenario.

| Text Domain | Score Functions | Test1 | | Test2 | | Test3 | |
|---|---|---|---|---|---|---|---|
| | | Human, GPT-3 | Human, Human | Human, ChatGPT | Human, Human | Human, GPT-4 | Human, Human |
| **XSum, Xsum** | Fast-DetectGPT | 7.4812 | 6.0978 | 7.9321 | 5.8211 | 7.1870 | 5.9146 |
| | DetectGPT | 2.7997 | 2.3369 | 2.4485 | 2.4753 | 2.0792 | 2.1686 |
| | NPR | 0.2025 | 0.1469 | 0.1418 | 0.1449 | 0.1184 | 0.1248 |
| | LRR | 0.5798 | 0.5798 | 0.6088 | 0.4623 | 0.5913 | 0.5412 |
| | Logrank | 1.3841 | 1.4601 | 1.1775 | 1.0302 | 1.4912 | 1.4399 |
| | Likelihood | 2.1195 | 2.1755 | 1.8610 | 1.5937 | 2.2947 | 2.1624 |
| | Entropy | 2.0369 | 1.9155 | 1.4552 | 1.3976 | 1.9421 | 1.8986 |
| | DNA-GPT | 1.0847 | 0.9399 | 0.9219 | 0.8006 | 1.0386 | 0.9783 |
| | RoBERTa-base | 0.9997 | 0.9977 | 0.9997 | 0.9977 | 0.9997 | 0.9977 |
| | RoBERTa-large | 0.9970 | 0.8191 | 0.9944 | 0.3471 | 0.9862 | 0.8191 |
| **Writing, Writing** | Fast-DetectGPT | 8.0598 | 6.5768 | 8.4128 | 5.9812 | 6.6396 | 6.5999 |
| | DetectGPT | 3.5032 | 3.1609 | 3.2897 | 3.1502 | 2.7128 | 3.3487 |
| | NPR | 0.2103 | 0.1500 | 0.1920 | 0.1356 | 0.1286 | 0.1502 |
| | LRR | 0.6487 | 0.5483 | 0.7428 | 0.5115 | 0.5417 | 0.4754 |
| | Logrank | 2.1419 | 1.4950 | 1.5963 | 1.4485 | 1.6212 | 1.4024 |
| | Likelihood | 3.3565 | 2.0134 | 2.4280 | 1.8810 | 2.3618 | 2.0046 |
| | Entropy | 2.8300 | 1.6976 | 1.6677 | 1.4732 | 2.0601 | 1.5292 |
| | DNA-GPT | 1.2223 | 1.2879 | 1.1989 | 1.0239 | 1.3639 | 1.2719 |
| | RoBERTa-base | 0.9997 | 0.9997 | 0.9997 | 0.9997 | 0.9996 | 0.9997 |
| | RoBERTa-large | 0.9992 | 0.9172 | 0.9456 | 0.5028 | 0.9172 | 0.9172 |
| **Pubmed, Pubmed** | Fast-DetectGPT | 5.6200 | 4.7132 | 5.8150 | 4.8065 | 4.6136 | 4.6400 |
| | DetectGPT | 2.0692 | 1.8394 | 2.1635 | 2.3610 | 1.5185 | 2.0677 |
| | NPR | 0.1888 | 0.2020 | 0.1867 | 0.2185 | 0.2028 | 0.2173 |
| | LRR | 0.6811 | 0.7180 | 0.7433 | 0.6000 | 0.8885 | 0.7180 |
| | Logrank | 1.8434 | 2.4121 | 1.9131 | 1.7634 | 2.6000 | 2.3949 |
| | Likelihood | 2.8480 | 3.4780 | 3.1426 | 2.8756 | 3.8419 | 3.5055 |
| | Entropy | 2.0549 | 2.2002 | 2.0570 | 1.8376 | 2.4877 | 2.3156 |
| | DNA-GPT | 5.1276 | 4.7710 | 4.3683 | 4.7319 | 4.9899 | 4.8031 |
| | RoBERTa-base | 0.9982 | 0.9962 | 0.9984 | 0.9953 | 0.9995 | 0.9963 |
| | RoBERTa-large | 0.9688 | 0.8863 | 0.8860 | 0.8863 | 0.8702 | 0.8863 |

*Table 10.* Values of $d_t$ with the assumption that we know the range of $|g_t| = |\phi(x_t) - \phi(y_t)|$ beforehand. Every two columns starting from the third column represent the $d_t$ values for each $t$ used under $H_1$ and $H_0$ in each test scenario.

| Text Domain | Score Function | Test1 | | Test2 | | Test3 | |
| --- | --- | --- | --- | --- | --- | --- | --- |
| | | Human, GPT-3 | Human, Human | Human, ChatGPT | Human, Human | Human, GPT-4 | Human, Human |
| XSum, Writing | Fast-DetectGPT | 7.7260 | 6.2430 | 8.0558 | 6.0578 | 7.3268 | 6.1780 |
| | DetectGPT | 3.1711 | 2.8288 | 2.8371 | 2.6976 | 2.5561 | 2.6884 |
| | NPR | 0.2001 | 0.1398 | 0.1736 | 0.1571 | 0.1249 | 0.1352 |
| | LRR | 0.5817 | 0.5355 | 0.6395 | 0.6469 | 0.4837 | 0.5354 |
| | Logrank | 1.8576 | 1.6371 | 1.4423 | 1.7444 | 1.0620 | 1.5484 |
| | Likelihood | 3.1515 | 2.2793 | 2.0097 | 2.3806 | 1.5890 | 2.3587 |
| | Entropy | 2.6910 | 2.0533 | 1.8358 | 2.0545 | 1.5863 | 1.7798 |
| | DNA-GPT | 1.0351 | 1.3620 | 0.8960 | 1.1271 | 0.7700 | 1.1035 |
| | RoBERTa-base | 0.9997 | 0.9920 | 0.9997 | 0.9997 | 0.9996 | 0.9997 |
| | RoBERTa-large | 0.9992 | 0.9172 | 0.9455 | 0.3471 | 0.2952 | 0.5028 |
| XSum, Pubmed | Fast-DetectGPT | 5.7796 | 5.2418 | 6.0477 | 5.5643 | 5.2864 | 5.5884 |
| | DetectGPT | 1.9280 | 2.1249 | 2.2991 | 2.7521 | 2.0123 | 2.8121 |
| | NPR | 0.1529 | 0.1813 | 0.1607 | 0.1745 | 0.1608 | 0.1609 |
| | LRR | 0.7144 | 0.6422 | 0.6765 | 0.5376 | 0.6952 | 0.5247 |
| | Logrank | 1.6718 | 2.2405 | 1.3408 | 1.6702 | 1.3285 | 1.5756 |
| | Likelihood | 2.4632 | 3.1031 | 2.3094 | 2.5604 | 2.2453 | 2.3964 |
| | Entropy | 2.2528 | 2.3452 | 1.9397 | 1.6543 | 1.9994 | 1.8273 |
| | DNA-GPT | 6.6172 | 6.2606 | 5.6482 | 6.0117 | 6.3382 | 6.0307 |
| | RoBERTa-base | 0.9983 | 0.9964 | 0.9984 | 0.9976 | 0.9995 | 0.9976 |
| | RoBERTa-large | 0.9688 | 0.8191 | 0.8610 | 0.8863 | 0.8703 | 0.8863 |
| Writing, Pubmed | Fast-DetectGPT | 4.9091 | 5.3870 | 6.3816 | 5.4647 | 5.8265 | 5.3231 |
| | DetectGPT | 2.7326 | 2.9296 | 2.5926 | 3.0456 | 2.3285 | 3.1283 |
| | NPR | 0.1546 | 0.1829 | 0.1609 | 0.1927 | 0.1733 | 0.1588 |
| | LRR | 0.7289 | 0.6567 | 0.8526 | 0.7093 | 0.8070 | 0.6364 |
| | Logrank | 1.7772 | 2.0524 | 2.0617 | 1.8547 | 1.8669 | 1.6617 |
| | Likelihood | 2.6541 | 2.8230 | 3.1136 | 2.6684 | 3.0234 | 2.6871 |
| | Entropy | 2.4478 | 2.5402 | 2.6542 | 2.3688 | 2.3984 | 2.2263 |
| | DNA-GPT | 7.0524 | 6.6958 | 6.0130 | 6.3766 | 6.6221 | 6.3146 |
| | RoBERTa-base | 0.9983 | 0.9964 | 0.9996 | 0.9995 | 0.9996 | 0.9996 |
| | RoBERTa-large | 0.9688 | 0.9172 | 0.8610 | 0.8863 | 0.8703 | 0.8863 |

*Table 11.* Values of $\Delta$ derived by using Neo-2.7 as the scoring model for the first eight score functions to score texts from the same domain. There are three source models: GPT-3, ChatGPT and GPT-4. Every two columns starting from the third column represent the $\Delta$ values under $H_1$ and $H_0$ in each test scenario.

| Text Domain | Score Function | Test1 | | Test2 | | Test3 | |
|---|---|---|---|---|---|---|---|
| | | Human, GPT-3 | Human, Human | Human, ChatGPT | Human, Human | Human, GPT-4 | Human, Human |
| XSum, Xsum | Fast-DetectGPT | 1.9598 | 0.0770 | 2.9471 | 0.0106 | 1.7638 | 0.0664 |
| | DetectGPT | 0.5656 | 0.0729 | 0.4816 | 0.0114 | 0.1672 | 0.0844 |
| | NPR | 0.0305 | 0.0031 | 0.0248 | 0.0006 | 0.0047 | 0.0037 |
| | LRR | 0.0349 | 0.0159 | 0.1568 | 0.0032 | 0.0839 | 0.0127 |
| | Logrank | 0.1938 | 0.0455 | 0.3794 | 0.0077 | 0.2397 | 0.0378 |
| | Likelihood | 0.3488 | 0.0630 | 0.5960 | 0.0109 | 0.3704 | 0.0521 |
| | Entropy | 0.0586 | 0.0517 | 0.1456 | 0.0117 | 0.1009 | 0.0400 |
| | DNA-GPT | 0.1579 | 0.0476 | 0.2742 | 0.0287 | 0.1432 | 0.0188 |
| | RoBERTa-base | 0.5939 | 0.0210 | 0.5946 | 0.0002 | 0.1871 | 0.0212 |
| | RoBERTa-large | 0.3941 | 0.0133 | 0.2037 | 0.0001 | 0.0516 | 0.0135 |
| Writing, Writing | Fast-DetectGPT | 2.3432 | 0.0204 | 3.1805 | 0.0206 | 2.1831 | 0.0002 |
| | DetectGPT | 0.5752 | 0.0181 | 0.6980 | 0.0093 | 0.2223 | 0.0088 |
| | NPR | 0.0330 | 0.0001 | 0.0415 | 0.0001 | 0.0083 | 0.0001 |
| | LRR | 0.0979 | 0.0062 | 0.1512 | 0.0009 | 0.0516 | 0.0070 |
| | Logrank | 0.3737 | 0.0055 | 0.5407 | 0.0026 | 0.3103 | 0.0028 |
| | Likelihood | 0.5915 | 0.0217 | 0.8511 | 0.0046 | 0.5126 | 0.0171 |
| | Entropy | 0.2260 | 0.0130 | 0.3428 | 0.0036 | 0.1490 | 0.0093 |
| | DNA-GPT | 0.2441 | 0.0348 | 0.3745 | 0.0155 | 0.2370 | 0.0193 |
| | RoBERTa-base | 0.6070 | 0.0279 | 0.2184 | 0.0093 | 0.0220 | 0.0186 |
| | RoBERTa-large | 0.3845 | 0.0085 | 0.0143 | 0.0033 | 0.0175 | 0.0052 |
| Pubmed, Pubmed | Fast-DetectGPT | 0.7397 | 0.0025 | 1.3683 | 0.0243 | 1.1594 | 0.0268 |
| | DetectGPT | 0.2417 | 0.0092 | 0.2470 | 0.0051 | 0.1729 | 0.0041 |
| | NPR | 0.0146 | 0.0015 | 0.0175 | 0.0012 | 0.0084 | 0.0028 |
| | LRR | 0.0101 | 0.0070 | 0.1099 | 0.0042 | 0.0944 | 0.0112 |
| | Logrank | 0.0282 | 0.0293 | 0.4115 | 0.0092 | 0.3725 | 0.0385 |
| | Likelihood | 0.0771 | 0.0411 | 0.6922 | 0.0121 | 0.6119 | 0.0532 |
| | Entropy | 0.0766 | 0.0316 | 0.2939 | 0.0048 | 0.2701 | 0.0364 |
| | DNA-GPT | 0.3518 | 0.1309 | 0.8721 | 0.0875 | 0.4354 | 0.2184 |
| | RoBERTa-base | 0.1859 | 0.0078 | 0.1666 | 0.0137 | 0.0466 | 0.0215 |
| | RoBERTa-large | 0.1318 | 0.0077 | 0.1294 | 0.0031 | 0.0652 | 0.0108 |

*Table 12.* Values of $\Delta$ derived by using Neo-2.7 as the scoring model for the first eight score functions to score texts from different domains. There are three source models: GPT-3, ChatGPT and GPT-4. Every two columns starting from the third column represent the $\Delta$ values under $H_1$ and $H_0$ in each test scenario.

| Text Domain | Score Functions | Test1 | | Test2 | | Test3 | |
|---|---|---|---|---|---|---|---|
| | | Human, GPT-3 | Human, Human | Human, ChatGPT | Human, Human | Human, GPT-4 | Human, Human |
| XSum, Writing | Fast-DetectGPT | 2.1206 | 0.2430 | 2.8602 | 0.2996 | 1.8737 | 0.3096 |
| | DetectGPT | 0.6211 | 0.0278 | 0.6617 | 0.027 | 0.1657 | 0.0478 |
| | NPR | 0.0323 | 0.0008 | 0.0377 | 0.0038 | 0.0039 | 0.0044 |
| | LRR | 0.0391 | 0.0526 | 0.0757 | 0.0747 | 0.0137 | 0.0723 |
| | Logrank | 0.0893 | 0.2899 | 0.2081 | 0.33 | 0.0175 | 0.3249 |
| | Likelihood | 0.1362 | 0.4771 | 0.3281 | 0.5184 | 0.0166 | 0.5121 |
| | Entropy | 0.1843 | 0.4233 | 0.1228 | 0.462 | 0.3142 | 0.4539 |
| | DNA-GPT | 0.1279 | 0.1509 | 0.1953 | 0.1638 | 0.0672 | 0.1505 |
| | RoBERTa-base | 0.6513 | 0.0163 | 0.2744 | 0.0653 | 0.0596 | 0.0562 |
| | RoBERTa-large | 0.3789 | 0.0029 | 0.0253 | 0.0078 | 0.0011 | 0.0112 |
| XSum, Pubmed | Fast-DetectGPT | 0.4179 | 0.3244 | 0.9937 | 0.3989 | 0.7686 | 0.3639 |
| | DetectGPT | 0.0098 | 0.2424 | 0.0723 | 0.3245 | 0.1538 | 0.3308 |
| | NPR | 0.0148 | 0.0017 | 0.0133 | 0.0029 | 0.0064 | 0.0048 |
| | LRR | 0.0214 | 0.0184 | 0.0868 | 0.0273 | 0.0633 | 0.0199 |
| | Logrank | 0.0349 | 0.0226 | 0.3819 | 0.0388 | 0.312 | 0.0219 |
| | Likelihood | 0.1195 | 0.0014 | 0.6838 | 0.0205 | 0.5612 | 0.0024 |
| | Entropy | 0.0782 | 0.1232 | 0.4019 | 0.1032 | 0.3533 | 0.1197 |
| | DNA-GPT | 2.8511 | 2.6302 | 3.2363 | 2.4517 | 3.0468 | 2.3930 |
| | RoBERTa-base | 0.3711 | 0.193 | 0.3591 | 0.2063 | 0.2609 | 0.1928 |
| | RoBERTa-large | 0.2087 | 0.0846 | 0.2165 | 0.0902 | 0.1633 | 0.0873 |
| Writing, Pubmed | Fast-DetectGPT | 0.6609 | 0.0813 | 1.2933 | 0.0993 | 1.0782 | 0.0543 |
| | DetectGPT | 0.0376 | 0.2702 | 0.0453 | 0.2974 | 0.1060 | 0.2830 |
| | NPR | 0.0156 | 0.0025 | 0.0170 | 0.0008 | 0.0108 | 0.0004 |
| | LRR | 0.0312 | 0.0342 | 0.1614 | 0.0474 | 0.1356 | 0.0525 |
| | Logrank | 0.3248 | 0.2673 | 0.7119 | 0.2912 | 0.6369 | 0.3030 |
| | Likelihood | 0.5966 | 0.4785 | 1.2021 | 0.4978 | 1.0733 | 0.5145 |
| | Entropy | 0.5015 | 0.5465 | 0.8638 | 0.5651 | 0.8072 | 0.5735 |
| | DNA-GPT | 3.002 | 2.7812 | 3.4000 | 2.6155 | 3.1972 | 2.5435 |
| | RoBERTa-base | 0.3548 | 0.1767 | 0.2939 | 0.1410 | 0.2046 | 0.1366 |
| | RoBERTa-large | 0.2057 | 0.0817 | 0.2088 | 0.0825 | 0.1521 | 0.0761 |

# I. Addressing Parameter Estimation Challenges and Mixed LLMs or Sources Tasks

**Estimate Parameters Based on More Samples.** Better parameter estimation can enhance the performance of our algorithm. For instance, in the previous experiments, we used the first 10 samples of each sequence to estimate parameters $d_t$ and $\epsilon$. Here, we give an example to show that using a larger sample size for estimation could possibly yield better results. Specifically, using 20 samples from each sequence for estimation, with test begins at the 21-th time step, could lead to improved algorithm performance.

Figure 11 demonstrates that when parameters are estimated with more samples from the initial time steps, all score functions maintain False Positive Rates (FPRs) below the specified significance levels. Additionally, almost all functions identify the LLM source more quickly compared to when fewer samples are used for estimation. This example indicates the potential benefits of using more extensive data for parameter estimation in enhancing the effectiveness of our approach.

**Tasks for a Mixture of LLMs or Sources.** Our method can be extended to additional tasks. Its fundamental goal in sequential hypothesis testing is to determine whether texts from an unknown source originate from the same distribution as those in a prepared human text dataset, where we consider mean value as the statistical metric.

Even if texts from the LLM source are produced by various LLMs, they still satisfy the alternative hypothesis, which means that our statistical guarantees remain valid and the algorithm could continue to perform effectively. The results are illustrated in Figure 12a.

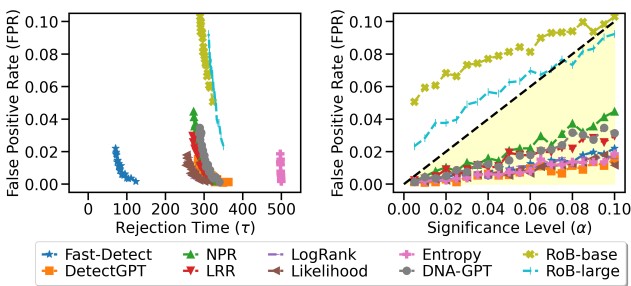

(a) Averaged test results for estimating parameters based on first 10 samples.

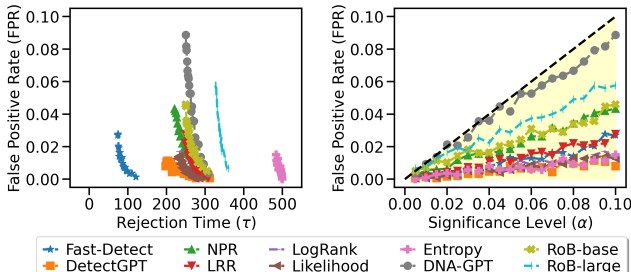

(b) Averaged test results for estimating parameters based on first 20 samples.

*Figure 11.* Comparison of different durations in the initial stage for parameter estimation in Scenario 2. Here, text $x_t$ is sampled from XSum and $y_t$ is from 2024 Olympic news or machine-generated news, across three source models. The scoring model is Neo-2.7. Subfigure (b) suggests that a longer duration for parameter estimation may lead to improved results. We emphasize that the test begins at $t = 21$.

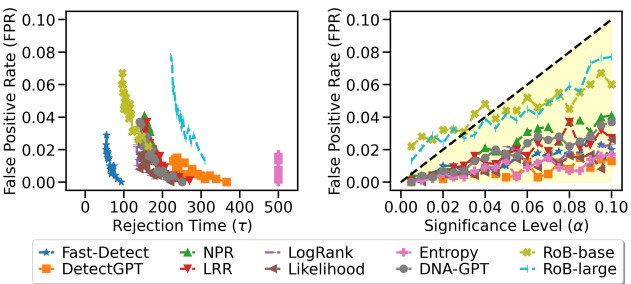

(a) Task 1: When the LLM source posts texts generated by different LLMs (under $H_1$). Specifically, the sequence consists of 100 texts generated by Gemini-1.5-Pro, 200 texts generated by Gemini-1.5-Flash, and 200 texts generated by PaLM 2.

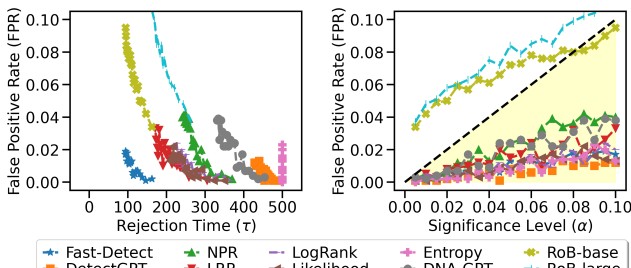

(b) Task 2: When the unknown source posts a mixture of human-written texts and LLM-generated texts (under $H_1$). Specifically, the sequence consists of 200 texts written by human, and 300 texts generated by PaLM 2.

*Figure 12.* (**Extension to other settings**) (a) Results when the sequence of texts $y_t$ are produced by various LLMs instead of a single one. (b) Results under the setting that the null hypothesis corresponds to the case that all the texts from the unknown source are human-written, while the alternative hypothesis $H_1$ corresponds to the one that not all $y_t$ are human-written.

When the unknown source publishes both human-written and LLM-generated texts, our method can effectively address this scenario. Here, the null hypothesis assumes that all texts from the unknown source are human-written. In contrast, the alternative hypothesis posits that not all texts are human-written, which indicates the presence of texts generated by LLMs. Figure 12b demonstrates that our algorithm, equipped with nearly all score functions, consistently performs well in this new context.

