# OpenReview forum: "Online Detection of LLM-Generated Texts via Sequential Hypothesis Testing by Betting"
_ICML.cc/2025/Conference — ICML 2025 poster_

### Official Review · Reviewer_DHzx · 2025-02-27

**Overall Recommendation:** 3

**Summary:**

The paper focuses on developing an algorithm for online detection of texts generated by
large language models (LLMs). The main contribution is an algorithm based on sequential
hypothesis testing techniques, which allows for quick and accurate identification of LLMgenerated texts in a streaming setting. The algorithm leverages score functions from existing
offline detection methods and uses a betting framework to accumulate evidence for or
against the null hypothesis that the text source is human-written. The authors conduct
comprehensive experiments using various score functions and datasets to demonstrate the
effectiveness of their method.

**Claims And Evidence:**

The claims made in the paper, such as the ability to control the false positive rate and
provide an upper bound on the expected detection time, are supported by clear and
convincing evidence. The authors present theoretical propositions and empirical results from
experiments that validate these claims.

**Essential References Not Discussed:**

None

**Experimental Designs Or Analyses:**

The authors use a variety of score functions and datasets to test the algorithm's
performance, which provides a comprehensive evaluation of its effectiveness. The
experiments are repeated multiple times to account for randomness, and the results are
averaged to provide reliable estimates of the algorithm's performance. The analysis of the
results is thorough, and the authors provide clear explanations for the observed trends.

**Methods And Evaluation Criteria:**

Methods: the use of sequential hypothesis testing and betting techniques is a novel
approach to the online detection of LLM-generated texts.
The evaluation criteria, including the false positive rate and rejection time, are relevant
metrics for assessing the performance of the algorithm in a streaming setting. The authors
also consider composite hypotheses and provide a detailed analysis of the algorithm's
performance under different scenarios.

**Other Comments Or Suggestions:**

The paper discusses using pseudo base-to-new partitions to train the detector, but it's not
clear whether the performance of the model is sensitive to the number of partitions. While
the ablation study shows performance increases with more partitions, there might be an
optimal number beyond which performance plateaus or even deteriorates. A more detailed
analysis of this aspect could provide better guidance on how to choose the number of
partitions (K) for practical applications.

**Other Strengths And Weaknesses:**

Strengths:
The proposed method has practical applications in areas such as content moderation,
academic integrity, and social media analysis.
Weaknesses:
The online optimization and betting framework may have higher computational complexity
compared to traditional methods, especially when dealing with large-scale datasets.

**Questions For Authors:**

1. How does the algorithm handle cases where the score function outputs are highly variable or noisy? Would this affect the algorithm's ability to control the false positive rate?
2. The experiments in the paper are conducted on specific datasets and LLMs. How well does the method generalize to other types of texts or different LLMs?
3. The paper suggests that the online optimization and betting framework may have higher computational complexity compared to traditional methods, particularly when handling large-scale datasets. Could the authors provide a more detailed discussion on the computational complexity of the proposed framework?
4. Specifically, how does it scale with larger datasets, and are there any strategies for mitigating this complexity in practical applications?
5. The paper mentions that performance improves with an increasing number of pseudo base-to-new partitions in the detector training. However, it remains unclear whether there exists an optimal number of partitions. Specifically, is there a point beyond which adding more partitions leads to diminishing returns, or could the performance even start to deteriorate? A more thorough analysis of this relationship would be helpful in understanding the balance between the number of partitions and the model’s effectiveness.
6. Additionally, could the authors provide more detailed guidance on how to select the number of partitions (K) for practical applications? Understanding how to choose K in real-world scenarios would be beneficial, particularly in optimizing performance while avoiding unnecessary complexity or overfitting.

**Relation To Broader Scientific Literature:**

None.

**Theoretical Claims:**

The paper includes several theoretical claims, including the control of the false positive rate
and the upper bound on the expected detection time. The proofs for these claims are
provided in the appendix. The proofs appear to be correct and well-reasoned, with clear
explanations of the assumptions and the logical steps involved

---

> ### Author Rebuttal · Authors · 2025-03-31
>
> We first would like to thank the reviewer for the positive feedback and good suggestions. Here are our responses.
>
> **(Supplementary material.)**
> We would like to clarify that the supplementary material includes a detailed README.md file, which provides a clear overview of the codebase, experiment organization, score functions used, and testing scenarios.
>
> **(Computational complexity.)**
> Thank you for raising this point. It is unclear which specific “traditional methods” the reviewer is referring to, so we address both possible interpretations below.
> - If the reviewer is referring to offline detection methods (e.g., Fast-DetectGPT, LRR, etc.). We would like to clarify that our method is built on top of existing offline detectors, and is designed to complement rather than compete with them.
> Our focus is to build an online framework that adapts existing offline detectors to an online setting where texts are observed in a streaming fashion. In this case, the only additional computation introduced by our method is a 1D Online Newton Step (ONS) update. For example, if we use the score function of Fast-DetectGPT to compute the text score, **the computational time will be that of the underlying offline detector plus a 1D online Newton update, which incurs only light overhead.**
> - On the other hand, if the reviewer is referring to the fixed-time permutation test baseline that we include for comparison, we note that permutation-based methods typically involve multiple resampling and recomputation of the test statistic (details in Appendix G), which is often computationally heavier than our lightweight online updates. In our case, each round of detection only involves computing a single score and performing a 1D update, making it much more efficient in streaming settings.
>
> **(Partitions.)**
> We would like to clarify that our method is built on top of existing offline detectors, and is designed to complement rather than compete with them. Thus, our method does not involve base-to-new partitioning, nor does it assume or require any particular structure in the score function.
> Our contribution lies in providing a general online detection framework that takes text scores evaluated by a chosen score function of an existing offline detector as input, and performs sequential hypothesis testing with rigorous statistical guarantees. Specifically, our method ensures type-I error control at level-$\alpha$ and power of 1.
> While some of the offline detectors we incorporate, such as Fast-DetectGPT, may internally use base-to-new partitioning as part of their score computation, this is entirely orthogonal to our approach, which operates solely on the text scores.
> Therefore, the choice of how the score function is computed, including the number of partitions or any use of base-to-new structure, is not within the scope of our framework.
>
> **(FPR control for noisy score inputs.)**
> Our theoretical guarantee on controlling the type-I error (false positive rate) relies on **Ville's inequality** applied to a **non-negative supermartingale** wealth process. This guarantee holds **regardless of the variance or noisiness** of the score function outputs, as long as the null hypothesis holds, i.e., the expected difference in scores satisfies $\mu_x=\mu_y$ or $|\mu_x - \mu_y| \leq \epsilon$ under the composite null.
>
> Specifically, as shown in Appendix E of our paper, the wealth process
> $W_t = \prod_{i=1}^t (1 - g_i \theta_i)$
> is a supermartingale under $H_0$ where the score dufference $ g_t = \phi(x_t) - \phi(y_t) $ and $ \theta_t $ is the adaptive betting fraction. Applying randomized Ville’s inequality [1, 2] to this process ensures that
> $\mathbb{P}(\exists t: W_t \geq 1/\alpha \text{ or } W_T \geq Z/\alpha) \leq \alpha,$
> where $Z\sim\text{Unif}(0,1).$
>
> The above property of wealth process guarantees control of type-I error at level $\alpha$. While noisy scores may increase the rejection time under $H_1$, they do not compromise our theoretical type-I error guarantee under $H_0$.
>
> [1] Ville, J. Etude critique de la notion de collectif. Gauthier- Villars Paris, 1939.
>
> [2] Ramdas, A. and Manole, T. Randomized and exchangeable improvements of markov’s, chebyshev’s and chernoff’s inequalities. arXiv preprint arXiv:2304.02611, 2023.
>
>  **(Generalization to other types of texts or different LLMs.)**
> We would like to clarify that we also have more experimental results for additional datasets such as WritingPrompts (stories) and PubMed (long-form answers written by human experts) in Appendix H. Furthermore, we have tested scenarios where the target sequence of texts is of a different domain/topic from that of the prepared human-written texts (e.g., Figure 8). Besides, We extend our method to scenarios where texts from the unknown source are produced by various LLMs (see Figure12(a)), and when the unknown source posts a mixture of human-written texts and LLM-generated texts (see Figure12(b)). Our method consistently perform well in the above scenarios.

---

### Official Review · Reviewer_2DZ6 · 2025-03-14

**Overall Recommendation:** 3

**Summary:**

The paper studies the problem of detecting whether a series of texts is LLM/machine-generated sequentially. To this end, they build on existing work on offline detection of LLM-generated text which proposes a variety of scoare functions, as well on recent work in sequential hypothesis testing. More concretely, they frame the problem as an sequential hypothesis testing problem where the null is that the expected score over the source’s distribution is equal to that over a human-written distribution (which they assume access to). They particularize the algorithm and analysis from Chugg et. al. 2023 to their setting, and subsequently enjoy a bound on the probability of false rejection (over all time) as well as a an upper bound on the expected stopping time of the method. They run experiments with a variety of score function specifications from prior literature and three source models that show promising empirical results in terms of detection speed and false discovery control for the proposed method, showing that testing by betting can be useful for this application of recent interest and relevance.

**Claims And Evidence:**

Yes the claims are generally supported by evidence. One note is that the assumption on the score function is repeatedly referred to as “mild” yet in my opinion it is not really obviously that mild, so maybe this point could be discussed more at length or be given more nuance. Additionally, maybe more care could be taken to analyze (or at least discuss) the predictiveness/usefulness of the upper bound on stopping time. For example, if one uses the actual, true constants of the bound and plugs in reasonable values for the parameters, I am worried the resulting bound would be quite large.

**Essential References Not Discussed:**

Not to my knowledge.

**Experimental Designs Or Analyses:**

I feel like the experimental design and analyses are intuitive and fitting for the application at hand. I appreciate the inclusion of both the alpha-spending baseline as well as the invalid baseline.

**Methods And Evaluation Criteria:**

I think the experiments conducted are a good first step and useful for illustrating the abilities of testing by betting approaches on the application at hand. The authors consider relevant baselines, as well as multiple score specifications and different LLM models generating the text. Of course by only using two particular datasets of human text we only get very preliminary insight and much additional work would be needed to properly evaluate machine-generated detection methods in a real world, live setting.

**Other Comments Or Suggestions:**

One suggestion I have is considering one-sided hypotheses tests. To me, it would make sense to consider scores for which the higher they are, the more likely it is for the source to be an LLM. In that case, considering the null hypothesis be $H_0: \mu_y > \mu_x + \epsilon$ would be more fitting. I find the testing for equality or near equality to be a bit unrealistic, as, whatever the score, I would expect different means for different subpopulations of humans, and it may be easier to hope for a score that maintains ordering (i.e. is able to cluster humans to one side and machines to the other). It would also possibly require a bit of additional modification of the techniques from Chugg et. al. 2023 and the analysis.

Typos and other small comments:
- “is” should be “are” on line 73 right column I think
- “guarantees” should be “guarantee” on line 229 left column
- “perform” should be “performs” on line 369 left column
- Figure 1 appears as too low quality for me so maybe that can be fixed?

**Other Strengths And Weaknesses:**

The core strength of this paper is that it applies the testing by betting framework to the very suitable and relevant application (detecting machine-generated text). This application in my opinion makes much more sense as an online decision problem rather than offline and therefore I think the paper is a good addition to the existing literature on detecting llm-generated text.

The assumption that the machine and human generated texts come i.i.d. from their respective distributions is quite significant and potentially too unrealistic for the scenario at hand. One other possible weakness is that it may make more sense to consider one-sided hypotheses tests and test for ‘>’ or ‘<‘ rather than $H_0: \mu_x = \mu_y$.

**Questions For Authors:**

My main question is related to the suggestion above — why would we have a nulll test for ‘=‘ rather than ‘>’/‘<‘? What are your thoughts on doing one-sided tests given the nature of this application?

**Relation To Broader Scientific Literature:**

The paper employs recent theoretical advancements in sequential hypothesis testing, most notably that of Chugg et. al. 2023 (but this is part of a flourishing line of work, for which the authors give a good list of references). The paper also ties to existing work on detecting machine-generating texts, by proposing a sequential approach, as well as by employing different scores proposed in various prior works.

**Theoretical Claims:**

The theoretical claims follow almost directly from prior work in sequential hypothesis testing by betting.

---

> ### Author Rebuttal · Authors · 2025-03-31
>
> We first thank the reviewer for your positive feedback and helpful suggestions. Below are our responses.
>
> **(Assumption on the score function.)**
> We agree that the assumption of the existence of a score function that produces distinguishable means for human-written texts and LLM-generated texts is critical. We describe this assumption as “mild” based on empirical observations. As discussed in Appendix G (Table 1) and Appendix H (Table11-12), we provide evidence showing that the empirical mean score differences produced by most of the adopted score functions are significant across multiple human-written text datasets and LLMs. Additionally, to address the case where the mean difference under $H_0$ is small but nonzero (i.e., when both sequences are human-written by different individuals), we have extended our theoretical analysis to the composite hypothesis setting (see Proposition 3.2). This formulation allows for a tolerance parameter $\epsilon$ and leads to a more realistic criterion. Our experiments are all conducted in this setting and show good performance, which validates the effectiveness of our method under our assumption.
>
> **(Expected stopping time bound)** Regarding the upper bound on the expected stopping time, we acknowledge the reviewer’s concern that the bound might be loose when directly plugging in actual parameter values. We emphasize that this bound serves primarily as a worst-case guarantee. Nevertheless, our empirical results demonstrate that the algorithm often detects LLM-generated texts much earlier than the bound suggests in practice. Additionally, the bound also provides valuable insights into the stopping time by revealing which factors affect it and in what way, such as the mean score gap $\Delta$, the bound estimate $d_*$, and the user-specified significance level $\alpha$.
>
> **(Real world setting.)**
> We would like to clarify that we have more experimental results considering the real-world applications in Appendix H for additional datasets such as WritingPrompts and PubMed. Furthermore, we have tested scenarios where the target sequence of texts is of a different domain/topic from that of the prepared human-written texts (e.g., Figure 8). Besides, We extend our method to scenarios where texts from the unknown source are produced by various LLMs (see Figure12(a)), and when the unknown source posts a mixture of human-written texts and LLM-generated texts (see Figure12(b)). The experimental results of the above scenarios show the effectiveness of our method.
>
> **(Overwhelming quantity.)** Thanks for your helpful suggestion! we will revise and streamline the appendix in future versions.
>
> **(One-sided hypotheses tests.)**
> We thank the reviewer for this insightful comment. We think it is relatively straightforward to modify the underlying method for
> the one-sided test that the reviewer kindly points out. For example, if the null hypothesis is $H_0: \mu_x < \mu_y$,  then we can specify the wealth dynamic as $W_t = W_{t-1} \left( 1 - \theta_t (\phi(y_t) - \phi(x_t)) \right).$ Under the null $H_0$, we have
> $E[ W_t | F_{t-1} ] = W_{t-1} \left( 1 - \theta_t E[ \phi(y_t) - \phi(x_t)] \right) = W_{t-1} \left( 1 + \theta_t (\mu_x - \mu_y ) \right) \leq W_{t-1},$ where we used the fact that $\theta_t$ is $F_{t-1}$-measurable. This shows that the wealth process can be a non-negative supermartingale for this case. Therefore, we can apply Ville's inequality to show that this test is a valid-$\alpha$ test. Furthermore, when the alternative $H_1$ is true, we can apply the online Newton method can help increase the wealth.
>
> On the other hand, we would like to point out that using this one-sided test requires the user knows that the score of human-written text is higher (or lower) than the score of machine-generated text beforehand. The two-sided test does not require this assumption. Specifically, if the score function $\phi(\cdot)$ is not necessarily an affinity score (e.g., perplexity [1]), then the formulation in our paper might be more suitable.
>
> We agree that different means for different sub-populations of humans are more realistic. While we are primarily concerned with the sequential testing scenario of humans vs. LLMs, we believe that enabling different sub-populations of humans and exploring potential application scenario is a valuable future direction.
>
> [1] Yongqiang Ma, Jiawei Liu, Fan Yi, Qikai Cheng, Yong Huang, Wei Lu, and Xiaozhong Liu. Ai vs. human–differentiation analysis of scientific content generation. arXiv preprint arXiv:2301.10416, 2023.
>
> **(Typos and other comments.)**
> We thank the reviewer for carefully pointing out these typos and the figure issue. We have corrected these. Figure 1 will be replaced with a higher-resolution version to improve its visual quality. We appreciate your attention to detail.

---

### Official Review · Reviewer_bKsA · 2025-03-15

**Overall Recommendation:** 2

**Summary:**

This work has studied an online detection method for AI-generated texts, and it can identify texts from unknown source models. The proposed method mainly makes use of sequential hypothesis testing and has an advantage of non-parametric property. Comparision experiments with several baseline methods (e.g. DetectGPT, Fast-Detect, LRR) shows the merits of the proposed method.

## update after rebuttal. I appreciate the authors for the rebuttal response which partially addressed my concerns. However, I still feel it is not essential to the motivation/necessity of such sequential detection scenarios. Besides, I am not convinced by the comparison with some offline detectors, e.g. Binoculars, which has already performed well in terms of detection speed and accuracy. I would therefore maintain my rating.

**Claims And Evidence:**

Yes.

**Essential References Not Discussed:**

NA.

**Experimental Designs Or Analyses:**

Yes, I have checked the experimental part which includes baselines and comparison results.

**Methods And Evaluation Criteria:**

Yes, the idea of using hypothesis testing by betting makes sense for the online detection scenario.

**Other Comments Or Suggestions:**

No.

**Other Strengths And Weaknesses:**

Strengths:

The authors propose an online LLM-generated text detection method based on sequential hypothesis testing, capable of identifying texts generated by unknown-source LLMs. Additionally, this method is non-parametric and does not require assuming different prior distributions for human and LLM-generated texts.

The proposed algorithm is rigorously justified through extensive theoretical and experimental analyses. The content is comprehensive (though carefully reading nearly 50 pages is unrealistic, such thoroughness is necessary). The writing is well-structured, notation usage is correct, figures are visually appealing, and the construction process is easy to understand.

From the perspective of evaluating the effectiveness of different detection algorithms, this paper is undoubtedly a well-founded and rigorous innovation. It parallels the Performance Profiles framework used in optimization research (https://arxiv.org/abs/cs/0102001), but currently, no theoretically guaranteed and objective evaluation methodology exists in AI-generated text detection. For example, evaluating detection algorithms based on rejection counts is an interesting idea.

Weaknesses:

In the Introduction, the authors emphasize that existing score-based detection methods are sensitive to threshold selection (lines 036-038, right column). However, considering representative real-time methods like Binoculars and Fast-DetectGPT, threshold adjustments do not seem to be frequently required for achieving robust detection across different domains and source models. This is even more evident for trained zero-shot methods like RoBERTa-Base/Large and ReMoDetect (NeurIPS 2024). Therefore, this may not be a significant weakness of prior detection methods, and the authors might need to further justify the necessity of an online detection approach.

The paper appears to primarily discuss sequential hypothesis testing in the context of existing detection methods rather than developing a fundamentally new detection approach (i.e., the adaptation seems too direct). Many of the claimed advantages are inherent to sequential hypothesis testing (e.g., error control and ONS strategies). Considering that some existing detection methods have also applied hypothesis testing concepts (such as Raidar (ICLR 2024) and certain watermarking techniques), albeit with differences (e.g., sequential testing does not require a fixed sample size in advance), the novelty of this work seems somewhat limited.

The study lacks evaluation in traditional attack scenarios, such as text rewriting, paraphrasing, style transfer, and multilingual settings—an essential step for validating a new detector.

In the Introduction, the authors state that their method does not assume any underlying distribution for human or machine-generated texts (lines 104-108, left column). However, later in the paper (lines 141-148, left column), human and LLM-generated texts are assumed to originate from some distribution. This inconsistency may lead to misunderstandings.

**Questions For Authors:**

I have some questions regarding this work:

i) The authors categorize detection methods as either offline or online, grouping all real-time, training-free detection methods under offline detection. Could the authors clarify the distinction between real-time detection methods (e.g., likelihood-based methods and Fast-DetectGPT) and their concept of online detection? Based on the authors’ analysis of the null hypothesis, the performance of online detection heavily relies on offline detection. If the score function from offline detection fails to effectively distinguish between human and LLM-generated texts, the effectiveness of hypothesis testing will also degrade. Given that offline detection methods can already be highly effective, why is online detection necessary? This issue is particularly relevant to the attack scenarios mentioned in Weaknesses point 3.

ii) What is the strategy for selecting the human reference sample x_t? The choice of reference samples directly affects the hypothesis testing results. The paper mainly considers news texts, but in real-world applications, how can we locate or curate suitable human text datasets to ensure that the test remains consistently effective?

iii) Binoculars (ICML 2024) achieves high AUROC while maintaining a low false positive rate (FPR), which is also one of the goals of the proposed method. How does this method compare against Binoculars to further highlight its advantages?

iv) The paper refers to \textit{time}, which seems to indicate the number of detection steps. Could the authors also report the actual computational time required for detection?

**Relation To Broader Scientific Literature:**

Yes, this work proposes a relatively novel way to conduct online detection of AI-generated texts. It enjoys faster detection speed, and might be more suitable for detecting streaming data(e.g. from social media).

**Theoretical Claims:**

Yes, I've checked two propositions (3.1 and 3.2).

---

> ### Author Rebuttal · Authors · 2025-03-31
>
> We thank the reviewer for your careful reading and useful comments. Here are our responses.
>
> **(Necessity.)** Our emphasis is not on the frequency of adjustments, but rather on the fact that **most** offline detectors detect by comparing the text score with a **pre-determined** threshold, chosen via training and directly impacting accuracy. This applies to detectors like RoBERTa-Base/Large and ReMoDetect, which require supervised training and threshold-based classification (e.g., probability). While our method only uses an offline detector to score texts, **the threshold (i.e., the significance level $\alpha$) is specified according to the user’s needs (the type-1 error allowed)**.
>
> **(Novelty.)** While prior work such as Raidar leverages hypothesis testing with log-probability scores in an offline setting, our approach is, to our knowledge, **the first to provide a general framework for adapting existing score-based detectors to the online setting**. Our key contribution is introducing an **any-time valid, non-parametric sequential testing framework** with **statistical guarantees** and **bounded expected detection time**, which is not available in existing offline detectors.
>
> **(Attack scenarios.)** Our work does not propose a new score function, which requires to be evaluated under attacks. Rather, we focus on how to use existing detectors in an online framework. Some have already addressed attacks: DetectGPT and Fast-DetectGPT consider paraphrasing attacks (Mitchell et al., 2023, Bao et al., 2023), and DNA-GPT studies revision attacks (Yang et al., 2023). While such attacks may reduce the gap between human and LLM scores, they typically do not eliminate the difference between them. Therefore, our basic hypothesis, that their population means are different, is reasonable.
>
> **(Distribution assumption.)** Lines 104–108: *"... our approach is non-parametric, and hence ..."*. That is, we make no assumption on the specific form of underlying distributions (e.g., Gaussian), which is standard for non-parametric methods (Balsubramani et al., 2015). Like most hypothesis testing methods, including non-parametric ones, we assume samples are drawn i.i.d. from unknown distributions (lines 141–148), without assuming their specific forms.
>
> **(Online vs. Real-time)** Scenarios are different. Some detectors predict **individual** samples efficiently, but they are still applied offline: each sample is scored and classified independently. In contrast, our online framework leverages sequential hypothesis testing, where samples are observed in a **streaming** fashion. Our goal is to obtain an anytime-valid level-$\alpha$ test (i.e., controlling type-1 error when the source is human) while controlling the time to reject $H_0$ when the alternative is true (i.e., controlling the time to correctly identify the source as an LLM). To the best of our knowledge, this is **a novel online detection scenario of detecting LLMs, and therefore existing works for offline classification cannot be directly applied to our online scenario.** For example, a naive way to adapt the offline detector in our scenario is that if at a certain point, the offline detector predicts that a sample is written by LLM, then it declares the source is LLM (and hence stops). However, this naive method will have a false positive rate of 1 when the number of rounds becomes sufficiently large, unless the offline detector never makes an error in recognizing human text.
>
> **(Human samples.)** The reference human texts need not match the domain of the observed samples, because the hypothesis testing framework does not impose any assumptions on their distribution. As shown in Appendix H, our method works well even when the domains of the reference and target texts differ (XSum, WritingPrompts, PubMedQA).
>
> **(vs. Binoculars.)** Our method is not an offline detector, but rather a general online detection framework with strong theoretical guarantees. While offline detectors optimize metrics like AUROC, we focus on making sequential decisions with any-time valid statistical guarantees. In principle, our method could operate on top of Binoculars’ score function in an online manner, providing early stopping and statistical guarantees that Binoculars alone does not offer. Therefore, the goals of our method are complementary to existing detectors.
>
> **(Computational time.)** Our method is designed to offer an online framework to offline detectors. **The computational time will be that of the underlying offline detector plus a 1D online Newton update, which incurs very light overhead.**  The number of detection steps reflects the statistical notion of sample complexity, a key metric in sequential hypothesis testing. Our focus is on the statistical efficiency (i.e., how much texts are needed before making a decision with statistical guarantees), rather than raw computational time, which mainly depends on the score function used.

---

### Decision · Program_Chairs · 2025-05-01

**Decision:**

Accept (poster)

**Comment:**

The authors present a method for online detection of LLM-generated text using a sequential hypothesis testing framework based on betting. The authors build offline score functions and apply a nonparametric sequential testing procedure to provide statistical guarantees. The authors also conduct extensive theoretical analysis and empirical evaluation across multiple datasets, score functions, and source models.

The methodology is interesting and the application to the streaming detection settings is compelling. The reviewers and I generally agree that the authors provide a rigorous theoretical framing, and a comprehensive empirical setup and experiments. Reviewers also appreciated the novelty of the online detection scenario and the robustness of the statistical guarantees.

However, reviews and I share concerns about the overall novelty relative to existing sequential testing frameworks, as well as the limited evaluation against recent offline detectors like Binoculars. One reviewer also questioned the necessity of the proposed framework given the effectiveness of existing real-time detectors, though the rebuttal by the authors clarified that the proposed method offers anytime-valid statistical guarantees not present in those baselines.

Overall, while not all concerns were fully resolved, the paper presents a sound and well-executed framework that extends existing offline detectors into a meaningful online setting with rigorous guarantees. On this basis I recommend a weak accept for this paper, however I do recommend the authors address the issues raised by the reviewers and noted above in their revised manuscript.